# Exact and Linear Convergence for Federated Learning under Arbitrary Client Participation is Attainable

**Bicheng Ying**[1]    **Zhe Li**[2]    **Haibo Yang**[2]

[1]Google    [2]Rochester Institute of Technology
ybc@google.com, {zl4063,hbycis}@rit.edu

## Abstract

This work tackles the fundamental challenges in Federated Learning (FL) posed by arbitrary client participation and data heterogeneity, prevalent characteristics in practical FL settings. It is well-established that popular FedAvg-style algorithms struggle with exact convergence and can suffer from slow convergence rates since a decaying learning rate is required to mitigate these scenarios. To address these issues, we introduce the concept of stochastic matrix and the corresponding time-varying graphs as a novel modeling tool to accurately capture the dynamics of arbitrary client participation and the local update procedure. Leveraging this approach, we offer a fresh decentralized perspective on designing FL algorithms and present `FOCUS`, Federated Optimization with Exact Convergence via Push-pull Strategy, a provably convergent algorithm designed to effectively overcome the previously mentioned two challenges. More specifically, we provide a rigorous proof demonstrating that `FOCUS` achieves exact convergence with a linear rate regardless of the arbitrary client participation, establishing it as the first work to demonstrate this significant result.

## 1 Introduction

Federated Learning (FL) has emerged as a powerful paradigm for distributed learning, enabling multiple clients to collaboratively train models without sharing raw data. Yet, a central challenge in FL is the *arbitrary and unpredictable* nature of client participation. In real-world FL, clients may join or leave at will, participate intermittently, or drop out due to connectivity or resource constraints.

Recall the goal of the FL problem is to minimize the following sum-of-loss function:

$$F(x) := \frac{1}{N} \sum_{n=1}^{N} f_n(x), \quad f_n(x) := \mathbb{E}_{\xi \sim \mathcal{D}_n} \hat{f}_n(x; \xi), \tag{1}$$

where $x \in \mathbb{R}^d$ represents the $d$-dimensional model parameter and $f_n$ stands for the local cost function. It is well established that when clients perform multiple local updates on non-i.i.d. data, their local models tend to diverge. This leads to client drift from the optimal solution of problem (1), a phenomenon that persists even under the often impractical uniform client sampling assumption [Karimireddy et al., 2020, Li et al., 2020]. Moreover, arbitrary client participation introduces another objective bias: instead of converging to the true global optimum, the global model converges to a stationary point of a distorted, participation-weighted objective [Wang et al., 2020, Wang and Ji, 2022]. To mitigate this persistent error, existing methods typically require decaying the learning rate asymptotically to zero, at least in theory. While this strategy can reduce the bias in the limit, it often leads to slower convergence. Hence, a key question naturally arises:

> **Question**: *Is it possible to achieve exact convergence under both arbitrary client participation and multiple local updates without decaying the learning rate?*

We will provide an affirmative answer to this question in this paper. We begin by introducing a novel analytical framework that reformulates the core operations of FL - client participation, local updates,

39th Conference on Neural Information Processing Systems (NeurIPS 2025).

and model aggregation over time-varying graphs - as a sequence of stochastic matrix multiplications [Horn and Johnson, 2012]. Next, with this tool, we develop a new algorithm FOCUS, Federated Optimization with Exact Convergence via Push-pull Strategy, which is inspired by decentralized optimization algorithms [Nedic and Ozdaglar, 2009, Sayed et al., 2014, Lian et al., 2017, Lan et al., 2020]. More specifically, we leverage the push-pull technique [Xin and Khan, 2018, Pu et al., 2020] with the time-varying graphs [Nedic et al., 2017, Ying et al., 2021, Nguyen et al., 2025] instead of commonly used static or strongly connected communication graphs, since FOCUS is designed for the FL setting. Compared to the variance reduction technique [Johnson and Zhang, 2013, Defazio et al., 2014] or the adaptively learning participation probabilities, the push-pull approach handles the unknown client participation scenario much better both empirically and theoretically.

Our main contributions are summarized as follows:

- We provide a systematic approach to reformulate all core processes of FL – client participation, local updating, and model aggregation through the stochastic matrix multiplication.

- We proposed Federated Optimization with Exact Convergence via Push-pull Strategy (FOCUS), which is designed based on the optimization principle instead of heuristic design.

- Even under arbitrary client participation, FOCUS exhibits linear convergence (exponential decay) for both strongly convex and non-convex (with PL condition) scenarios without assuming the bounded heterogeneity or decaying the learning rates.

- We also introduce a stochastic gradient variant, SG-FOCUS, which demonstrates faster convergence and higher accuracy, both theoretically and empirically.

| Algorithm | Exact Converg.[1] | Strongly-Convex Complexity[2] | Non-Convex Complexity | Assumptions[5] | | Extra Comment |
|---|---|---|---|---|---|---|
| | | | | Participation | Hetero. Grad. | |
| FedAvg [Li et al., 2020] | ✘ | $O(\frac{1}{\epsilon})$ | $O(\frac{1}{\epsilon^2})$ | Uniform | Bounded | Bounded gradient assumption |
| LocalSGD [Koloskova et al., 2020] | ✘ | $O(\frac{1}{\sqrt{\epsilon}})$ | $O(\frac{1}{\epsilon^{3/2}})$ | Uniform | Bounded | Doubly stochastic matrix |
| FedAU [Wang and Ji, 2024] | ✘ | – | $O(\frac{1}{\epsilon^2})$ | Arbitrary | Bounded | Bounded global gradient |
| FedAWE [Xiang et al., 2024] | ✘ | – | $O(\frac{1}{\epsilon^2})$ | Arbitrary | Bounded | Doubly stochastic matrix |
| SCAFFOLD [Karimireddy et al., 2020] | ✘[3] | $O\big(\log(\frac{1}{\epsilon})\big)$ | $O(\frac{1}{\epsilon})$ | Uniform | None | Comm. $2d$ vector per round[6] |
| ProxSkip/ScaffNew [Mishchenko et al., 2022] | ✘ | $O\big(\log(\frac{1}{\epsilon})\big)$ | – | Full | None | Comm. $2d$ vector per round |
| MIFA [Gu et al., 2021] | ✘ | – | $O(\frac{1}{\epsilon^2})$ | Arbitrary | Bounded | Bounded delay Assump. + Server stores each client model |
| FOCUS (This paper) | ✔ | $O\big(\log(\frac{1}{\epsilon})\big)$ | $O\big(\log(\frac{1}{\epsilon})\big)^{[4]}$ | Arbitrary | None | No need to learn partici. prob. |

Table 1: Comparison of multiple algorithms. [1] Exact convergence refers to the algorithm's ability to converge to the exact solution under arbitrary sampling, without requiring a decaying learning rate. [2] Complexity refers to the number of iterations required for the algorithm to achieve an error within $\epsilon$ of the optimal solution. We have removed the impact of the stochastic gradient variance in all rates. [3] There is no convergence proof of SCAFFOLD under arbitrary client participation scenario. Empirically, we observed it may be possible. [4] This rate is established with PL condition. [5] Arbitrary participation refers to Assumption 1 and the bounded heterogeneous gradient are the assumptions that $\|f_i(x) - F(x)\| \leq \sigma_G$. [6] It is possible to reduce the uplink communication into $d$ while downlink one is still $2d$ [Huang et al., 2024].

## 2 Related Work

FedAvg [McMahan et al., 2017] is the most widely adopted algorithm in FL. It roughly consists of three steps: 1) the server activates a subset of clients, which then retrieves the server's current model. 2) Each activated client independently updates the model by training on its local dataset. 3) Finally, the server aggregates the updated models received from the clients, computing their average. This process can be represented mathematically as:

$$x_{0,i}^{(r)} \Leftarrow x_r, \quad \forall i \in S_r \qquad \text{(Pull Model)} \tag{2a}$$

$$\text{For } t = 0, 1, \cdots, \tau - 1 : \qquad \text{(Local Update)}$$

$$x_{t+1,i}^{(r)} = x_{t,i}^{(r)} - \eta \nabla f_i(x_{t,i}^{(r)}) \qquad \forall i \in S_r \text{ in parallel} \tag{2b}$$

$$x_{r+1} \Leftarrow \frac{1}{|S_r|} \sum_{i \in S_r} x_{\tau,i}^{(r)} \qquad \text{(Aggregate Model)} \tag{2c}$$

where the set $S_r$ represents the indices of the sampled clients at the communication round $r$. The notation $x_r \in \mathbb{R}^d$ stands for the server's model parameters at $r$-th round, while $x_{t,i}^{(r)}$ stands for the client $i$'s model at the $t$-th local update step in the $r$-th round. We use $\Leftarrow$ to indicate that communication has happened between clients and the server.

Because of the data heterogeneity and multiple local update steps, Li et al. [2020] has shown that the fixed point of FedAvg is not the same as the minimizer of (1) in the convex scenario. More specifically, they quantified that

$$\|x^o - x^\star\|^2 = \Omega\big((\tau - 1)\eta\big)\|x^\star\|^2, \tag{3}$$

where $x^o$ is the fixed point of the FedAvg algorithm and $x^\star$ is the optimal point. This phenomenon, commonly referred to as client drift [Karimireddy et al., 2020], can be mitigated by introducing a control variate during the local update step, an approach inspired by variance reduction techniques [Johnson and Zhang, 2013]. Prominent examples of this strategy, including SCAFFOLD [Karimireddy et al., 2020] and ProxSkip [Mishchenko et al., 2022], can further circumvent the need for a bounded heterogeneity assumption. Yet, this approach incurs increased communication costs, doubling them due to the transmission of a control variate with the same dimensionality as the model parameters.

Many analytical studies on FL assume that the sampled clients are drawn from a uniform distribution, an assumption shared by the literature cited in the preceding paragraph, but this is almost impractical in reality [Kairouz et al., 2021, Xiang et al., 2024, Li et al., 2025]. Wang and Ji [2022] shows that FedAvg might fail to converge to $x^\star$ under non-uniform sampling distributions, even with a decreasing learning rate $\eta$. To address the challenges posed by non-uniformity, a common approach involves either explicitly knowing or adaptively learning the client participation probabilities during the iterative process and subsequently modifying the averaging weights accordingly [Wang and Ji, 2024, Wang et al., 2024, Xiang et al., 2024]. Yet, neither of them can achieve exact convergence, and the learning process may slow down the convergence. An alternative approach is to use Variance Reduction (VR) techniques, as seen in methods like MIFA Gu et al. [2021] and FedVARP Jhunjhunwala et al. [2022]. Yet, the heuristic integration of VR with FL often fails to jointly address the client drift issue. This results, once again, in inexact convergence when a constant learning rate is used.

It is known that FL and decentralized optimization are closely related [Lalitha et al., 2018, Koloskova et al., 2020, Kairouz et al., 2021], and this work is closely related to the tools introduced in the decentralized optimization society. We leave a detailed decentralized literature review in Appendix A.

## 3    Graph, Stochastic Matrix, and Arbitrary Client Participation

FL algorithms are commonly expressed in a per-client style, as exemplified by the previously highlighted FedAvg formulation (2a)-(2c). While this representation offers ease of understanding and facilitates straightforward programming implementation, a stacked vector-matrix representation can unlock more powerful mathematical tools for the design and analysis of FL algorithms.

To illustrate the concept, let us consider two toy examples of vector-matrix multiplication:

$$W_{\text{assign}}x = \begin{bmatrix} 1 & 0 & 0 \\ 1 & 0 & 0 \\ 0 & 0 & 1 \end{bmatrix} \begin{bmatrix} x_0 \\ x_1 \\ x_2 \end{bmatrix} = \begin{bmatrix} x_0 \\ x_0 \\ x_2 \end{bmatrix}, \quad W_{\text{avg}}x = \begin{bmatrix} 0 & 0.5 & 0.5 \\ 0 & 1 & 0 \\ 0 & 0 & 1 \end{bmatrix} \begin{bmatrix} x_0 \\ x_1 \\ x_2 \end{bmatrix} = \begin{bmatrix} (x_1 + x_2)/2 \\ x_1 \\ x_2 \end{bmatrix}$$

While the calculations themselves are straightforward, their significance lies in the appropriate interpretation of the matrices and vectors within the FL context. We interpret $x_i \in \mathbb{R}^{1 \times d}$ as the model parameter stored in the worker $i$. Index 0 is assigned for the server, and the result indices are for clients. Then, the first $W_{\text{assign}}$ can be viewed as the server assigning its value $x_0$ to client 1 while the value of client 2 is unchanged as the same as not participated scenario. The second $W_{\text{avg}}$ can be viewed as the server setting its own value as the average of the value of worker 1 and worker 2. These two toy matrices reflect the pull and aggregate model – two key steps in the FedAvg algorithm.

More formally, given a sampled client indices set $S_r$, subscript $r$ for the $r$-th round, we define the model-assign matrix $R(S_r)$ and the model-average matrix $A(S_r)$ as

$$R(S_r)[i,j] = \begin{cases} 1 & \text{if } i \in S_r \text{ and } j = 0 \\ 1 & \text{if } i \notin S_r \text{ and } j = i \\ 0 & \text{otherwise} \end{cases}, \quad A(S_r)[i,j] = \begin{cases} 1 & \text{if } i = j \neq 0 \\ 1/|S_r| & \text{if } i \in S_r \text{ and } j = 0 \\ 0 & \text{otherswise} \end{cases} \tag{4}$$

While the mathematical notation of the matrix may not be immediately apparent, its structure should be clear to see the illustration provided in Figure 1. In the figure, we utilize the graph language to

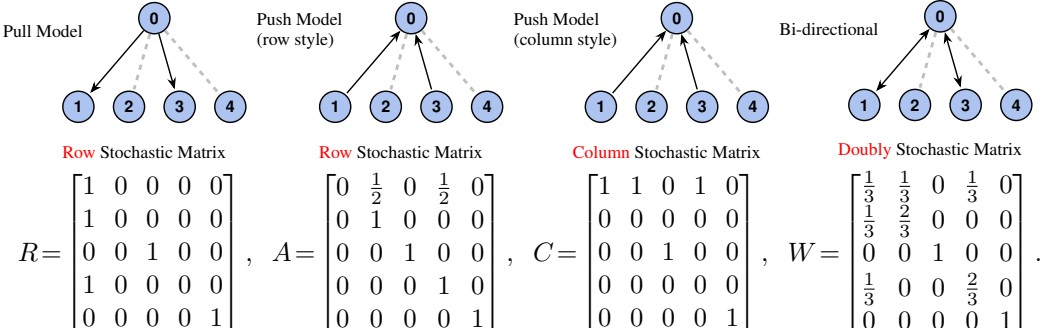

$$R=\begin{bmatrix}1&0&0&0&0\\1&0&0&0&0\\0&0&1&0&0\\1&0&0&0&0\\0&0&0&0&1\end{bmatrix}, \quad A=\begin{bmatrix}0&\frac{1}{2}&0&\frac{1}{2}&0\\0&1&0&0&0\\0&0&1&0&0\\0&0&0&1&0\\0&0&0&0&1\end{bmatrix}, \quad C=\begin{bmatrix}1&1&0&1&0\\0&0&0&0&0\\0&0&1&0&0\\0&0&0&0&0\\0&0&0&0&1\end{bmatrix}, \quad W=\begin{bmatrix}\frac{1}{3}&\frac{1}{3}&0&\frac{1}{3}&0\\\frac{1}{3}&\frac{2}{3}&0&0&0\\0&0&1&0&0\\\frac{1}{3}&0&0&\frac{2}{3}&0\\0&0&0&0&1\end{bmatrix}.$$

Figure 1: The graph representation of the communication pattern of 5 nodes and its possible corresponding stochastic matrices. For clearness, the self-loop is not drawn. If the node 0 is treated as server and node 1 to 4 as clients, the leftmost is a typical pull-model step, i.e. client 1 and 3 are participated; the second left graph depicts the model average step in the FedAvg; the third graph is a same graph but using column-stochastic matrix, which is uncommon in the FL literature; The last one is a typical (symmetric) doubly stochastic matrix case used in the decentralized optimization algorithm.

visualize the matrix $W$. We can treat $W$ as a weighted adjacency matrix; the non-zero value entry $W[i,j]$ implies a link from node $j$ to node $i$. Hence, $W$ is also commonly referred as the mixing matrix. Suppose $S_r = \{1,3\}$, then the matrices $R(S_r)$ and $A(S_r)$ correspond to the leftmost and second leftmost matrices and graphs depicted in the figure, respectively.

The weights are selected to ensure the resulting matrix is a stochastic matrix. Specifically, a matrix $W$ is called row stochastic if $W\mathbb{1} = \mathbb{1}$, where $\mathbb{1}$ is a all-one vector; it is called column stochastic if $\mathbb{1}^{\mathsf{T}}W = \mathbb{1}^{\mathsf{T}}$; and it is doubly stochastic if it satisfies both row and column stochastic properties [Horn and Johnson, 2012, Meyer, 2023]. It is straightforward to verify that the above two matrices both are row-stochastic matrices. Analogously, for the participation set, we can define a corresponding column stochastic matrix $C(S_r)$ and a doubly stochastic matrix $W(S_r)$.

$$C(S_r)[i,j]=\begin{cases}1 & \text{if } j \in S_r \text{ and } i=0\\1 & \text{if } j \notin S_r \text{ and } i=j\\0 & \text{otherswise}\end{cases}, \quad W(S_r)[i,j]=\begin{cases}1/|S_r| & \text{if } i \in S_r \text{ and } j=0\\1/|S_r| & \text{if } j \in S_r \text{ and } i=0\\1-\sum_i W[i,j] & \text{if } i=j\\0 & \text{otherswise}\end{cases}$$

Suppose $S_r = \{1,3\}$, then the matrices $W(S_r)$ and $C(S_r)$ correspond to the rightmost and second rightmost ones depicted in the figure. These four matrices will play the critical role in the following algorithm design and convergence proof section. In contrast to decentralized algorithms, where assumptions are directly imposed on the mixing matrix, **we do not make any assumption about them in this paper** since we utilize them to model the client participation process. For completeness, a brief review of stochastic matrices and their properties is provided in the Appendix C.

### 3.1 Arbitrary Client Participation Modeling

FL focuses on the process of generating the arbitrary client participation set $S_r$. Inspired by Wang and Ji [2022], in this paper, we model the arbitrary client participation by the following assumption.

**Assumption 1** (Arbitrary Client Participation). *In each communication round, the participation of the $i$-th worker is indicated by the event $\mathbb{1}_i$, which occurs with a **unknown** probability $p_i \in (0,1]$. $\mathbb{1}_i = 1$ indicates that the $i$-th worker is activated while $\mathbb{1}_i = 0$ indicates not. The corresponding averaging weights are denoted by $q_i$, where $q_i = \mathbb{E}\left[\mathbb{1}_i/(\sum_{j=1}^{N}\mathbb{1}_j)\right]$.*

Assumption 1 is a general one covering multiple cases:

**Case 1: Full Client Participation.** This is simply as $p_i \equiv 1$ and $q_i \equiv \frac{1}{N}$ for all client indices $i$.

**Case 2: Active Arbitrary Participation.** Each client $i$ independently determines if they will participate in the communication round. The event $\mathbb{1}_i$ follows the Bernoulli distribution $p_i$, where $p_i \in (0,1]$. (Note $\sum_i p_i \neq 1$.) If $\{p_i\}_{i=1}^{N}$ are close to each other, then $q_i \approx p_i/(\sum_j p_j)$.

**Case 3: Passive Arbitrary Participation.** The server randomly samples $m$ clients in each round. Each client is randomly selected without replacement according to the category distribution with the

normalized weights $q_1, q_2, \cdots, q_N$, where $\sum_i q_i = 1, q_i > 0$. $p_i$ does not have a simple closed form. But if it is sampled with replacement, then $p_i = 1 - (1 - q_i)^m$.

**Case 3a: Uniform Sampling.** This is a special case of case 3, where $p_i \equiv m/N$ and $q_i \equiv 1/m$.

Passive arbitrary participation is often referred to as arbitrary client sampling. We also use "sampling" and "client participation" interchangeably throughout this paper. Now, considering that $S_r$ is generated according to Assumption 1, it can be readily verified that the corresponding assigning matrix and averaging matrix possess the following property:

$$\bar{R} = \mathbb{E}\, R(S_r) = \begin{bmatrix} 1 & 0 & \cdots & 0 \\ q_1 & 1 - q_1 & \cdots & 0 \\ \vdots & \vdots & \ddots & \vdots \\ q_N & 0 & \cdots & 1 - q_N \end{bmatrix}, \quad \bar{A} = \mathbb{E}\, A(S_r) = \begin{bmatrix} 0 & q_1 & \cdots & q_N \\ 0 & 1 & \cdots & 0 \\ \vdots & \vdots & \ddots & \vdots \\ 0 & 0 & \cdots & 1 \end{bmatrix} \quad (5)$$

The column-stochastic matrix equals ($\bar{C} = \mathbb{E}[C(S_r)] = \bar{R}^{\mathsf{T}}$) by definition. While doubly stochastic matrices are prevalent in the decentralized literature, they are not often applicable to FL-style algorithms, and therefore we do not discuss them further. With this approach, we effectively transform the problem of arbitrary client participation probabilities into an analysis of the matrix properties of $R(S_r)$, $A(S_r)$, and $C(S_r)$ as which we will exploit in the subsequent section.

# 4 From Interpretation to Correction: A New Federated Optimization with Exact Convergence via Push-pull Strategy – FOCUS

In this section, we demonstrate how leveraging the graph and stochastic matrix can facilitate the development of more powerful FL algorithms.

## 4.1 Interpret FedAvg as Decentralized Algorithm with Time-Varying Graphs

A direct application of the above mixing matrix is that we can concisely represent the FL algorithm in vector-matrix form, similar to decentralized algorithms [Li et al., 2020, Koloskova et al., 2020].

First, let $\boldsymbol{x}_k = \text{vstack}[x_{k,0}; x_{k,1}; \cdots; x_{k,N}] \in \mathbb{R}^{(N+1) \times d}$ denote the state at iteration $k$. This matrix is formed by vertically stacking the server's model parameters $x_{k,0} \in \mathbb{R}^d$ and the model parameters $x_{k,i} \in \mathbb{R}^d$ from the $N$ workers. Similarly, let $\nabla \boldsymbol{f}(\boldsymbol{x}_k) = \text{vstack}[\mathbb{0}; \nabla f_1(x_{k,1}); \cdots; \nabla f_N(x_{k,N})] \in \mathbb{R}^{(N+1) \times d}$ represent the corresponding stacked vector of local gradients at iteration $k$.[1] Note that the first component of the stacked gradient is $\mathbb{0}$ because the server holds no data. This implies that the server's local loss function is identically zero, $f_0(x) \equiv 0$, and consequently, $\nabla f_0(x_{k,0}) = \mathbb{0}$. This also ensures that including the server's term $f_0$ does not alter the original loss function defined in (1).

Next, observe that during the local update phase of FedAvg, nodes compute updates independently without communication. In the context of our stochastic matrix, this corresponds to using the identity matrix, $I$. To represent the algorithm with a single iteration index $k$, we map the $t$-th local update in the $r$-th communication into the $k$-th iteration, where $k = r\tau + t$. Using the tools previously introduced, we can now reformulate FedAvg (2a)-(2c) as the following one-index iterative form:

$$\begin{aligned} \boldsymbol{y}_k &= R_k \boldsymbol{x}_k & \text{(Pull model)} & \quad (6a) \\ \boldsymbol{y}_k^+ &= \boldsymbol{y}_k - \eta D_k \nabla \boldsymbol{f}(\boldsymbol{y}_k) & \text{(Local update)} & \quad (6b) \\ \boldsymbol{x}_{k+1} &= A_k \boldsymbol{y}_k^+ & \text{(Agg. model)} & \quad (6c) \end{aligned}$$

where the **time-varying** matrices $R_k$, $A_k$, and $D_k$ are defined as

$$R_k = \begin{cases} R(S_r) & k = r\tau + 1 \\ I & \text{otherwise} \end{cases}, \quad A_k = \begin{cases} A(S_r) & k = (r+1)\tau \\ I & \text{otherwise} \end{cases}, \quad D_k[i, j] = \begin{cases} 1 & \text{if } i = j \in S_r \\ 0 & \text{otherswise} \end{cases}. \quad (7)$$

This diagonal matrix $D_k$ serves to deactivate unparticipated workers and the server during local updates. $S_r$ is the set of participated clients' indices at round $r$, which can be determined by the iteration $k$, i.e., $r\tau \leq k < (r+1)\tau$. An illustration of this process using graphs is shown in Figure 2.

---

[1] In the decentralized optimization literature, it is common to represent parameters $x$ and gradients $\nabla f(x)$ as row vectors (dimension $1 \times d$). This allows the graph mixing operation, defined by a matrix $W \in \mathbb{R}^{(N+1) \times (N+1)}$, to be concisely expressed as $W \boldsymbol{x}_k$.

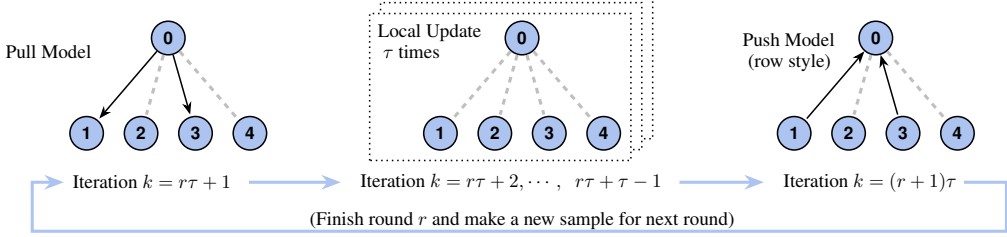

Figure 2: Represent FedAvg using graphs. The dashed line means no communication.

**Mixing Matrices in FedAvg.** It is feasible to further condense (6a)-(6c) into a single-line form

$$\boldsymbol{x}_{k+1} = W_k(\boldsymbol{x}_k - \eta \nabla \boldsymbol{f}(\boldsymbol{x}_k)) \tag{8}$$

The specific selection of $W_k$ is detailed in the Appendix. But $W_k$ cannot be a doubly stochastic matrix unless it is a full client participation case. Consequently, the theorem presented in [Koloskova et al., 2020] is not directly applicable to FedAvg in this context.

**Convergence Result of FedAvg with Arbitrary Participations.** In the appendix D, we provide a new proof of FedAvg under the arbitrary participation scenario through this decentralized optimization formulation. When the algorithm $k \to \infty$, the limiting point of FedAvg is around an irreducible neighborhood depending on the local update steps $\tau$, data heterogeneity $\sigma_g^2$, and the extra bias $\delta_g^2$ introduced due to non-uniform participation probabilities. This motivates us to develop a new FL algorithm capable of addressing and eliminating all aforementioned errors and biases.

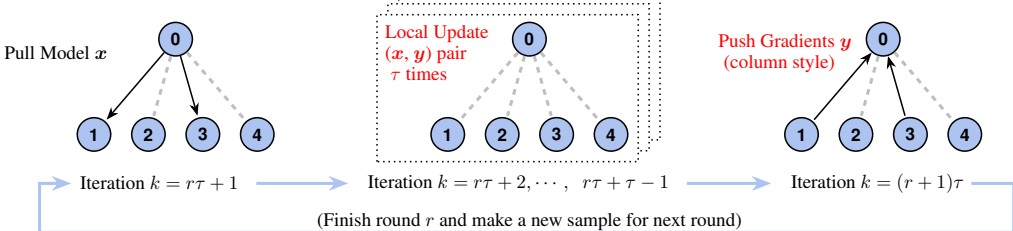

Figure 3: Illustration of our new `FOCUS` algorithm. There are two key differences from FedAvg style algorithm. One is it pulls the model variable $\boldsymbol{x}$ but pushes the gradient variable $\boldsymbol{y}$, and another is the push matrix is the column stochastic matrix instead of the row stochastic.

### 4.2 `FOCUS` Corrects Arbitrary Client Participation and Local-Update Bias

#### 4.2.1 Push-Pull Strategy for FL Settings

To eliminate the biases introduced by arbitrary client participation, we move beyond heuristic designs and adopt a formal optimization framework. This involves reformulating the FL problem as a constrained optimization task, a structure commonly employed in decentralized algorithms:

$$\min_{\{x_0, x_1, \cdots, x_N\}} \quad F(\boldsymbol{x}) = \frac{1}{N} \sum_{i=0}^{N} f_i(x_i) \tag{9}$$

$$\text{s.t.} \quad R(S_r)\boldsymbol{x} = \boldsymbol{x}, \ \forall \ S_r \tag{10}$$

Note a minor but critical difference from the formulation (1) is that there are $N + 1$ model parameters $x_i$ applied in each local cost function $f_i$ instead of a single $x$. To see the equivalence between this and (1), notice $R(S_r)\boldsymbol{x} = \boldsymbol{x}$ implies $x_i = x_0, \forall i \in S_r$. Consequently, if the union of all sampled client sets $\{S_r\}$ covers the entire client population, then all individual client models and the server model are constrained to converge to the same state.

This formulation motivated us to explore a primal-dual approach to solve this constrained problem. Among the various primal-dual-based decentralized algorithms, the push-pull algorithm aligns particularly well with the FL setting. It is characterized by the following formulation:

$$\boldsymbol{x}_{k+1} = R(\boldsymbol{x}_k - \eta_k \boldsymbol{y}_k) \tag{11}$$

$$\boldsymbol{y}_{k+1} = C\big(\boldsymbol{y}_k + \nabla \boldsymbol{f}(\boldsymbol{x}_{k+1}) - \nabla \boldsymbol{f}(\boldsymbol{x}_k)\big), \tag{12}$$

**Algorithm 1** `FOCUS`: Federated Optimization with Exact Convergence via Push-pull Strategy [2]

---

1: **Notation**: $x_r$ and $y_r$ are the server's values while $x_{t,i}^{(r)}$ and $y_{t,i}^{(r)}$ are clients' values.
2: **Initialize**: Choose learning rate $\eta$ and local update $\tau$. Server randomly chooses $x_0$ and sets $y_0 = 0$. All clients initiate with $\nabla f_i(x_{-1,i}^{(0)}) = y_{0,i}^{(0)}$.
3: **for** $r = 0, 1, ..., R - 1$ **do**
4:     Get $S_r$ an arbitrary client participation index set
5:     **for** $i$ in $S_r$ **parallel do**
6:         $x_{0,i}^{(r)} \Leftarrow x_r,\ y_{0,i}^{(r)} \Leftarrow 0$          ▷ Pull $x_r$ (No need to pull $y_r$)
7:         **for** $t = 0, \cdots, \tau - 1$ **do**
8:             $y_{t+1,i}^{(r)} = y_{t,i}^{(r)} + \nabla f_i(x_{t,i}^{(r)}) - \nabla f_i(x_{t-1,i}^{(r)})$          ▷ See $\nabla f_i(x_{-1,i}^{(r)})$ around (22).
9:             $x_{t+1,i}^{(r)} = x_{k,i}^{(r)} - \eta y_{t+1,i}^{(r)}$
10:         **end for**
11:     **end for**
12:     $y_{r+1} \Leftarrow y_r + \sum_{i \in S_r} y_{\tau,i}^{(r)}$          ▷ Push $y_{\tau,i}^{(r)}$ (Not Averaged)
13:     $x_{r+1} = x_r - \eta y_{r+1}$
14: **end for**

---

where $\boldsymbol{y}_0 = \nabla \boldsymbol{f}(\boldsymbol{x}_0)$, and $R$ and $C$ represent row-stochastic and column-stochastic matrices, respectively. The algorithm name "push-pull" arises from the intuitive interpretation of these matrices. The row-stochastic matrix $R$ can be interpreted as governing the "pull" operation, where each node aggregates information from its neighbors. Conversely, the column-stochastic matrix $C$ governs the "push" operation, where each node disseminates its local gradient information to its neighbors. Moreover, recalling the definition of row and column stochastic matrices $R\mathbb{1} = \mathbb{1}$ and $\mathbb{1}^\mathsf{T} C = \mathbb{1}^\mathsf{T}$, push-pull algorithm has the following interesting properties:

$$\boldsymbol{x}^\star = R\boldsymbol{x}^\star, \qquad \text{(consensus property)}$$

$$\mathbb{1}^\mathsf{T} \boldsymbol{y}_k = \mathbb{1}^\mathsf{T} \nabla \boldsymbol{f}(\boldsymbol{x}_k),\ \forall k \qquad \text{(tracking property)}$$

where $x^\star$ is the fixed point of the algorithm under some mild conditions on the static graph $R$ and $C$. The first property, consensus, implies that all workers' model parameters eventually converge to a common value. The second property, tracking, indicates that the sum of the variables $y$ (aggregated across workers) approximates the global gradient, ensuring the algorithm's iterates move in a direction that minimizes the global loss function. It is worth pointing out that when consensus is reached such that all relevant local models in $\boldsymbol{x}_k$ equal some $\bar{x}$, the sum of the local gradients $\mathbb{1}^\mathsf{T} \nabla \boldsymbol{f}(\boldsymbol{x}_k)$ becomes exactly $N\nabla F(\bar{x})$. For more details, we refer the readers to Pu et al. [2020], Xin and Khan [2018].

We are interested in solving the optimization problem with multiple constraints problem (9)-(10). The original push-pull algorithm is not sufficient. Analogous to the approach taken in the FedAvg section, Here, we extend it to the time-varying matrices $R_k$ and $C_k$ to model the client sampling and local update processes, respectively. These modifications lead to the following algorithmic formulation:

$$\boldsymbol{x}_{k+1} = R_k(\boldsymbol{x}_k - \eta D_k \boldsymbol{y}_k) \qquad (13)$$

$$\boldsymbol{y}_{k+1} = C_k\big(\boldsymbol{y}_k + \nabla \boldsymbol{f}(\boldsymbol{x}_{k+1}) - \nabla \boldsymbol{f}(\boldsymbol{x}_k)\big), \qquad (14)$$

where the definition of $R_k$ is the same as the one in FedAvg and $C_k = R_k^\mathsf{T}$ while $D_k$ is slightly different from (7) about the server's entry. $D_k[0,0] = 1$ if $k = r\tau + 1$ otherwise 0. The graph representation of this algorithm is shown in Figure 3.

#### 4.2.2 Convert Vector-Matrix Form Back To FL-Style Algorithm

Substituting the definition of the mixing matrix into (13) and (14), we will get a concrete FL algorithm as listed in Algorithm 1 with non-trivial transformations. **The steps to establish this new FL algorithm effectively reverses the process outlined in the previous subsection**. Here we provide a few key steps. First, it is straightforward to verify that $x_{k,i}$ and $y_{k,i}$ are not moved if the client $i$ is not participating in the corresponding round, so we will ignore them in the next derivation. At the beginning of the $r$-th round, i.e. $k = r\tau + 1$, (13) becomes

$$x_{k+1,0} = x_{k,0} - \eta y_{k,0} \qquad \text{(server updates)} \qquad (15)$$

---

[2]The code is available at `https://github.com/BichengYing/FedASL`. The algorithm was originally named Federated Learning for Arbitrary Sampling and Local Update (FedASL). The acronym ASL also stands for the Adaptation System Laboratory at UCLA and EPFL, where Dr. Ying completed his Ph.D.

$$x_{k+1,i} \Leftarrow x_{k+1,0}, \quad \forall i \in S_r \qquad \text{(client pulls model)} \qquad (16)$$

While at the end of the $r$-th round, i.e. $k = (r+1)\tau$, (14) becomes

$$y'_{k+1,i} = y_{k,i} + \nabla f_i(x_{k+1,i}) - \nabla f_i(x_{k,i}), \qquad \forall i \in S_r \qquad (17)$$

$$y_{k+1,0} \Leftarrow y_{k,0} + \sum_{i \in S_r} y'_{k+1,i} \qquad \text{(server collects info)} \qquad (18)$$

$$y_{k+1,i} \Leftarrow 0, \quad \forall i \in S_r \qquad \text{(client resets } y_k) \qquad (19)$$

Note that we introduce a temporary variable $y'_{k+1,i}$ because the matrix multiplication $C_k$ is applied on the updated value $y'_k$ instead of $y_k$ directly. During local updates, the server does not update the value while the client executes the local update in the gradient tracking style:

$$x_{k+1,i} = x_{k,i} - \eta y_{k,i} \qquad (20)$$
$$y_{k+1,i} = y_{k,i} + \nabla f_i(x_{k+1,i}) - \nabla f_i(x_{k,i}) \qquad (21)$$

Next, we revert to the standard two-level indexing used in FL by mapping the single iteration index $k = r\tau + t$ to the $r$-th iteration and $t$-th local update step and replacing $x_{k+1,i}$ by $x_{t,i}^{(r)}$.

Finally, assembling all the above equations together and switching the order of $x$ and $y$, we arrive at the `FOCUS` shown in Algorithm 1. Because of the switched order, at the beginning of each round, i.e. $k = r\tau$, the $y$-update becomes

$$y_{1,i}^{(r)} = y_{0,i}^{(r)} + \nabla f_i(x_{0,i}^{(r)}) - \nabla f_i(x_{-1,i}^{(r)}) \qquad (22)$$

Note that in the original update rule (21), the gradient $\nabla f_i(x_{k,i})$ is computed in the preceding step and then reused, thus avoiding redundant computation at the current step. This principle of gradient reuse carries over directly to the two-level index notation. Recall that $x_{t,i}^{(r)}$ will not change if the worker $i$ does not participate. Hence, we can establish, by induction, that $\nabla f_i(x_{-1,i}^{(r)})$ corresponds to the stored gradient from the end of the most recent round in which the worker participated.

## 5 Performance Analysis

Now we are ready to present the necessary assumptions and convergence property for `FOCUS`. Due to limited space, all proofs are deferred to Appendix E.

**Assumption 2** ($L-$Smoothness). *All local cost functions $f_i$ are $L-$smooth, i.e., $f_i(x) \le f_i(y) + \langle x - y, \nabla f_i(y) \rangle + \frac{L}{2}\|x - y\|^2$.*

**Assumption 3** ($\mu-$Strong Convexity). *All local cost functions $f_i$ are $\mu-$strongly convex, that is, $f_i(x) \ge f_i(y) + \langle x - y, \nabla f_i(y) \rangle + \frac{\mu}{2}\|x - y\|^2$.*

**Assumption 4** (PL Condition). *The global loss function $F$ satisfies the Polyak-Lojasiewicz condition $\|\nabla F(x)\|^2 \ge 2\beta\big(F(x) - F^\star\big), \quad \forall x$, where $\beta > 0$ and $F^\star$ is the optimal function value.*

**Theorem 1.** *Under arbitrary participation assumption 1 and $L-$Smoothness assumption 2, it can be proved that `FOCUS` converges at the following rates with various extra assumptions on $f_i$:*

- *$\mu-$**Strongly Convex**: Under extra assumption 3, if $\eta \le \min\{\frac{3\mu}{27NL^2}, \frac{1}{3L(\tau-1)}, \frac{q_{\min}^{3/2}}{8L\sqrt{N}}\}$,*

$$\mathbb{E}\|\bar{x}_{R\tau+1} - x^\star\|^2 \le \Psi_R \le (1 - \eta\mu N/2)^R \Psi_0 \qquad (23)$$

- *$\beta-$**PL Condition**: Under extra assumption 4, if $\eta \le \min\{\frac{3q_{\min}}{32N}, \frac{q_{\min}}{12\beta N}, \frac{q_{\min}}{16L^2}, \frac{q_{\min}^{3/2}}{8L\sqrt{N}}\}$,*

$$\mathbb{E}\, F(\bar{x}_{R\tau+1}) - F^\star \le \Phi_R \le (1 - \eta\beta N)^R \Phi_0 \qquad (24)$$

- ***General Nonconvex**: Under no extra assumption, if $\eta \le \min\{\frac{1}{2L(\tau-1)}, \frac{q_{\min}^{3/2}}{8L\sqrt{N}}, \frac{q_{\min}}{16L\sqrt{2N}}, \frac{1}{4LN}\}$,*

$$\frac{1}{R}\sum_{r=0}^{R-1} \mathbb{E}\|\nabla f(x_{R\tau+1})\|^2 \le \frac{8(f(x_1) - f^\star)}{\eta NR}, \qquad (25)$$

*where the Lyapunov functions $\Psi_r := \mathbb{E}\|\bar{x}_{r\tau+1} - x^\star\|^2 + (1 - 8\eta\tau LN)\mathbb{E}\|\mathbb{1}\bar{x}_{(r-1)\tau+1} - \boldsymbol{x}_{r\tau}\|_F^2$, $\Phi_r = \mathbb{E}\, F(\bar{x}_{r\tau+1}) - F^\star + \big(1 - 4\eta L^2\big)\mathbb{E}\|\mathbb{1}\bar{x}_{r\tau+1} - \boldsymbol{x}_{r\tau}\|^2$ and $q_{\min} = \min_i q_i$.* $\qquad\square$

**Remark.** Note the top two error terms are exponentially decayed, which implies the iteration complexity is $\mathcal{O}(\log(1/\epsilon))$. For the general non-convex case, we improve the typical $1/\sqrt{R}$ rate into $1/R$ thanks to the exact convergence property. See the comparison of our proposed algorithm with other common FL algorithms in Table 1. $\mathcal{O}(1/\epsilon^2) > \mathcal{O}(1/\epsilon) \gg \mathcal{O}(\log(1/\epsilon))$ in terms of communication and computation complexity. **Table 1 highlights the superior performance of** FOCUS**, which achieves the fastest convergence rate in all scenarios without particular sampling or heterogeneous gradients assumption.**

**Numerical Validation.** To validate our claims, we conducted a numerical experiment using synthetic data since this is the common approach to verify the exact convergence property. The results, presented in Figure 4, were obtained by applying the algorithms to a simple ridge regression problem with the parameters $d = 100$, $N = 16$, $K = 100$, $\lambda = 0.01$, and $\tau = 5$. The loss function is $F(x) = \frac{1}{N} \sum_{i=1}^{N} \sum_{k=1}^{K} \|a_{i,k}^{\mathsf{T}} x - b_{i,k}\|^2 + \lambda \|x\|^2$ All algorithms employed the same learning rate, $\eta = 2e - 4$. Three distinct sampling scenarios were examined: full client participation, uniform participation, and arbitrary participation. Notably, our FOCUS exhibits linear convergence and outperforms the other algorithms in all scenarios, particularly under arbitrary participation.

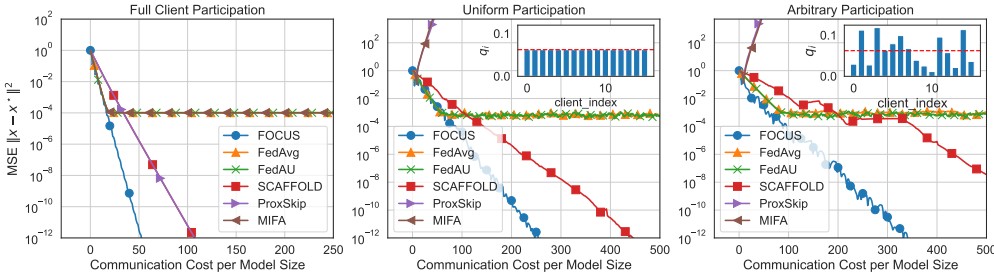

Figure 4: Convergence performance comparison of various FL algorithms. Under full client participation, FedAvg, FedAU, and MIFA exhibit identical performance, as do SCAFFOLD and ProxSkip, due to their theoretical equivalence in this setting. FedAvg and FedAU fail to converge to the optimal solution across all scenarios because their inherent error and bias cannot be eliminated using a fixed learning rate. ProxSkip diverges under uniform and arbitrary participation, as it is not designed for these conditions. We do not understand why MIFA diverges but it works in ML applications. While SCAFFOLD converges in all cases, our proposed algorithm, FOCUS, demonstrates faster convergence, especially under arbitrary participation.

## 5.1 Why FOCUS Can Converge Exactly for Arbitrary Participation Probabilities?

At first glance, the ability of FOCUS to achieve exact convergence under arbitrary client sampling probabilities may appear counterintuitive. Unlike other approaches, FOCUS **neither requires knowledge of the specific participation probabilities nor necessitates adaptively learning these rates**. The sole prerequisite for convergence is that each client maintains a non-zero probability of participation. Plus, the push-pull algorithm was never designed to solve the arbitrary sampling problem.

From an algorithmic perspective, FOCUS closely resembles the delayed/asynchronous gradient descent algorithm even though it is derived from a push-pull algorithm to fit the FL scenario. To see that, leveraging the tracking property of the variable $\boldsymbol{y}_k$ and special construction of matrix $C_k$, we can establish that the server's $y_{r+1} = \sum_{i=1}^{N} \nabla f_i(x_{k+1,i})$. Due to arbitrary client participation, at the iteration $k$, $x_{k+1,i}$ may hold some old version of the server's model if it does not participate. Thus, we arrive at an insightful conclusion: FOCUS **effectively transforms arbitrary participation probabilities into an arbitrary delay in gradient updates.** Hence, any client participation scheme, as long as each client participates with a non-zero probability, will still guarantee exact convergence.

## 5.2 Extension to Stochastic Gradients and ML Applications

In practical machine learning scenarios, computing full gradients is often computationally prohibitive. Therefore, stochastic gradient methods are commonly employed. Our proposed algorithm can be readily extended to incorporate stochastic gradients, resulting in the variant SG-FOCUS. However, due to space constraints, we focus on the deterministic setting in the main body of this paper. A comprehensive description of SG-FOCUS, along with its convergence analysis, is provided in Appendix F. The appendix also benchmarks SG-FOCUS's performance on the CIFAR-10 classification task, highlighting its faster convergence and improved accuracy over other FL algorithms. This performance trend echoes that of its deterministic counterpart.

**Theorem 2** (Informal Convergence Theorem of `SG-FOCUS`). *Under arbitrary participation assumption 1, $L-$Smoothness assumption 2, and unbiased and bounded variance assumption on stochastic gradient (See assumption 6 in appendix F), it can be proved that `SG-FOCUS` converges at the following rates with various extra assumptions on $f_i$:*

- $\mu-$**Strongly Convex**: *Under extra assumption 3, for sufficiently small learning rate $\eta$, we have*

$$\Gamma_R \leq \left(1 - \frac{\eta\mu N}{2}\right)^R \Gamma_0 + \frac{4(q_{\min} + N^2)}{\mu N q_{\min}}\eta\sigma^2, \tag{26}$$

*where the Lyapunov functions $\Gamma_r := \mathbb{E}\|\bar{x}_{(r+1)\tau+1} - x^\star\|^2 + \left(1 - 8\eta L^2 N/\mu\right)\mathbb{E}\|\mathbb{1}\bar{x}_{(r-1)\tau+1} - \boldsymbol{x}_{(r-1)\tau}\|_F^2$ and $\sigma$ is the variance upper bound of the stochastic gradient noise.*

- $\beta-$**PL Condition**: *Under extra assumption 4, for sufficient small learning rate $\eta$, we have*

$$\Omega_R \leq (1 - \eta\beta N)^R\Omega_0 + \left(\frac{L}{2} + 32(\tau - 1)^2 L^2 + \frac{8}{q_{\min}}\right)\frac{\eta}{\beta}\sigma^2, \tag{27}$$

*where $\Omega_r := F(\bar{x}_{r\tau+1}) - F^\star + \left(1 - 4\eta L^2\right)\mathbb{E}\|\mathbb{1}\bar{x}_{(r-1)\tau+1} - \boldsymbol{x}_{(r-1)\tau}\|^2$.*

- *General Nonconvex*: *Under no extra assumption, for sufficient small learning rate $\eta$, we have*

$$\frac{1}{R}\sum_{r=0}^{R-1}\mathbb{E}\|\nabla F(\bar{x}_{r\tau+1})\|^2 \leq \frac{8\left(F(\bar{x}_1) - F^\star\right)}{\eta N R} + \left(2LN + \frac{8N^2}{(N - 8L^2)q_{\min}}\right)\eta^2\sigma^2. \tag{28}$$

*See the formal statement and the proof in Appendix F.* □

**Remarks on Linear Speedup**. A common expectation in FL theory is the demonstration of a linear speedup in the convergence rate, where the rate scales proportionally with the number of clients, $N$. By inspecting the general non-convex convergence rate in Theorem 1, we see the error residual term is $\mathcal{O}(1/(\eta N R))$. Yet, the stability of `FOCUS` necessitates a learning rate $\eta$ that is restricted by $\mathcal{O}(1/N)$. This $N$ dependence cancels out. We want to highlight that this result is expected. The linear speedup typically holds when N is used to average out stochastic noise (like in Stochastic Gradient Descent variants). Since `FOCUS` is an exact algorithm, it does not introduce this stochasticity, and therefore, it is natural that the linear speedup benefit from increasing the number of clients is not reflected in the convergence bound. In contrast to the analysis of `FOCUS`, the convergence rate of `SG-FOCUS` in the general non-convex setting does indeed reflect the benefits of client aggregation. Specifically, by setting the learning rate $\eta$ to $\mathcal{O}(1/N)$, the stochastic variance term diminishes proportionally to $\mathcal{O}(1/N)$, which confirms the presence of the linear speedup.

## 6 Conclusion

This work addresses the critical challenges of arbitrary client participation and client drift in Federated Learning, two factors that prevent traditional algorithms from achieving exact convergence. By introducing a novel framework based on stochastic matrices and time-varying graphs, we model these dynamics and reformulate the FL problem as a constrained optimization task. This principled approach, moving beyond simple heuristics, led to the development of `FOCUS`, an algorithm derived from the push-pull optimization strategy. Our theoretical analysis and numerical experiments demonstrate that `FOCUS` can achieve exact linear convergence under any client participation scheme, without needing to know or learn participation probabilities. The extension to a stochastic gradient setting, `SG-FOCUS`, further validates its practical effectiveness.

**Limitations** The arbitrary client participation modeling used in the proof of this paper did not consider the Markov process, i.e., the client participation probabilities depend on the participation status in the last round. We believe that `FOCUS` still converges exactly under this scenario since the stochastic matrix modeling and push-pull strategy still hold for any realizations. However, the extension of the proof is non-trivial due to the correlation between stochastic matrices. We leave this and a more general arbitrary participation scenario for future research directions.

**Future Works.** The framework of stochastic matrices and time-varying graphs provides a novel tool for modeling arbitrary client participation and local update dynamics in FL. By leveraging this approach, we establish a formal connection between FL and the rich field of decentralized optimization. While this paper focused on FedAvg and the Push-Pull algorithm as initial examples, a promising avenue for future research is to adapt other sophisticated decentralized algorithms.

## Acknowledgement

The authors gratefully acknowledge Edward Duc Hien Nguyen and Xin Jiang for the discussion that inspired the connection between time-varying graphs and client sampling. Research reported in this publication was supported by the National Institute Of General Medical Sciences of the National Institutes of Health under Award Number R16GM159671. The content is solely the responsibility of the authors and does not necessarily represent the official views of the National Institutes of Health.

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

# Appendix

## A    Review of Decentralized Algorithms

The most widely adopted decentralized algorithm is decentralized gradient descent (DGD) [Nedic and Ozdaglar, 2009, Yuan et al., 2016], along with its adapt-then-combine version diffusion algorithm [Cattivelli et al., 2008, Chen and Sayed, 2012]. It is also a distributed algorithm where multiple workers collaboratively train a model without sharing local data. **In contrast to FedAvg, DGD has three key distinctions.** First, it operates without a central server; instead, workers communicate directly with their neighbors according to a predefined network topology [Nedic and Ozdaglar, 2009, Sayed et al., 2014, Lian et al., 2017]. Second, rather than relying on a server for model aggregation and synchronizing models, workers keep their local models, which differ (slightly) from each other typically. In each iteration, they exchange and combine model parameters with their

neighbors through linear combinations dictated by the network structure. This process can be formally expressed as

$$x_{k,i}^+ = x_{k,i} - \eta \nabla f_i(x_{k,i}) \qquad \text{(Local Update)} \qquad (29)$$

$$x_{k+1,i} = \sum_{j \in \mathcal{N}_i} w_{i,j} x_{k,j}^+ \qquad \text{(Graph Combination)} \qquad (30)$$

where $x_{k,i}$ stands for the $i$-th worker's parameters at the $k$-th iteration, the set $\mathcal{N}_i$ represents the neighbor's indices of worker $i$, and the non-negative weights $w_{i,j}$ satisfy $\sum_{j \in \mathcal{N}_i} w_{i,j} = 1$ for all $i$. Third, the algorithm is typically written as a single for-loop style instead of a two-level for-loop representation. It is straightforward to incorporate the multi-local-update concept into decentralized algorithms. However, it is not popular in the decentralized community.

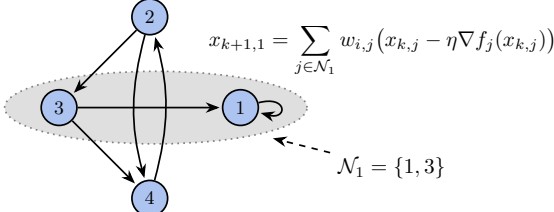

Figure 5: An illustration of Decentralized Gradient Descent.

It is well-known that the DGD with a fixed step size $\eta$ only converges to an $\mathcal{O}(\eta)$-sized neighborhood of the solution of the original algorithm [Yuan et al., 2016]. This resembles the FedAvg algorithm exactly. Subsequently, several extract decentralized optimization algorithms have been proposed, such as Extra [Shi et al., 2015], exact-diffusion/NIDS [Yuan et al., 2018, Li et al., 2019], DIGing/Gradient tracking [Nedic et al., 2017], etc. The key advancement of these algorithms is their capability to achieve extra convergence under a fixed step size. They formulate the original sum-of-cost problem into a constrained optimization problem and then apply the primal-dual style approach to solve the constrained problem [Ryu and Yin, 2022].

While it is common in the analysis of decentralized algorithms to assume a static and strongly connected underlying graph structure, a significant body of research also investigates time-varying graph topologies [Lan et al., 2020, Saadatniaki et al., 2020, Assran et al., 2019, Ying et al., 2021, Nguyen et al., 2025]. These studies often adopt one of two common assumptions regarding the dynamics of such graphs: either the union of graphs over any consecutive $\tau$ iterations or the expected graph is strongly connected [Nedić and Olshevsky, 2014, Koloskova et al., 2020]. Whereas previous research on time-varying topologies often relies on specific graph assumptions, this paper takes a different approach. We model client sampling and local updates using a graph representation, thereby avoiding any presuppositions about the underlying graph structure.

Lastly, we want to point out that this work focuses quite differently from the decentralized FL work [Beltrán et al., 2023, Shi et al., 2023, Fang et al., 2024], which is more closely related to decentralized algorithms instead of FL settings.

## B    Conventions and Notations

Under the decentralized framework, it is common to use matrix notation. We adopt the convention that the bold symbol, such as $\boldsymbol{x}$, is the stacked vector and the normal symbol, such as $x$, is the vector. With slight abuse of notation, we adopt the row vector convention and denote that

$$\boldsymbol{x}_k = \begin{bmatrix} - \ x_{k,0} \ - \\ - \ x_{k,1} \ - \\ \dots \\ - \ x_{k,N} \ - \end{bmatrix} \in \mathbb{R}^{N+1 \times d}, \quad \nabla \boldsymbol{f}(\boldsymbol{x}_k) = \begin{bmatrix} \nabla f_0(x_{k,0}) \\ \nabla f_1(x_{k,1}) \\ \dots \\ \nabla f_N(x_{k,N}) \end{bmatrix} \in \mathbb{R}^{N+1 \times d},$$

$$\nabla \boldsymbol{f}(\mathbb{1}\bar{x}_k) = \begin{bmatrix} \nabla f_0(\bar{x}_k) \\ \nabla f_1(\bar{x}_k) \\ \dots \\ \nabla f_N(\bar{x}_k) \end{bmatrix} \in \mathbb{R}^{N+1 \times d},$$

where $\mathbb{1}$ is an all-one vector. Note that in $\nabla \boldsymbol{f}(\boldsymbol{x}_k)$, each entry uses different $f_i$ and $x_{k,i}$. Similar usage for $\boldsymbol{y}_k$ as well. Except for $\boldsymbol{x}$, $\boldsymbol{y}$, and $\nabla \boldsymbol{f}$, other vectors are standard column vectors. Unlike most matrix conventions, the index of the matrix element starts from 0 instead of 1 in this paper since we set the index 0 to represent the server. Another common identity we used in the proof is

$$N \nabla F(\bar{x}) = \mathbb{1}^{\mathsf{T}} \nabla \boldsymbol{f}(\mathbb{1}\bar{x}), \tag{31}$$

This can be easily verified when substituting the definition of $F$. The rest usage of symbols is summarized in Table 2.

Table 2: Notations in this paper

| Notation | Meaning |
|---|---|
| $i$ | Index of clients |
| $k$ | Index of iterations |
| $r$ | Index of communication round and $r = \lfloor k/\tau \rfloor \tau$ |
| $\tau$ | The number of local update steps |
| $S_r$ | Indices set of clients sampled at $r$−th round |
| $d$ | Model parameter dimension |
| $u, q$ | Uniform / Arbitrary weighted averaging vector |
| $f_i, F$ | Local and global loss function |
| $R, V$ | Some row stochastic matrix |
| $C$ | Some column stochastic matrix |

In this paper, $\| \cdot \|$ denotes (induced) $\ell_2$ norm for both vector and matrix usage while $\| \cdot \|_F$ denotes the Frobenius norm.

## C  Brief Review about Stochastic Matrices and their Properties

Before applying the formalism of stochastic matrices to FL algorithm, we review the key properties of row- and column-stochastic matrices, as a clear understanding of these concepts is essential for the analysis that follows. There are three different types of stochastic matrices.

1. A *row stochastic matrix*, also called *right stochastic matrix*, is a square matrix of nonnegative real numbers denoted as $R \in \mathbb{R}_{\geq 0}^{n \times n}$, with each row summing to 1, i.e., $R\mathbb{1} = \mathbb{1}$, where $\mathbb{1}$ denotes the all-ones vector of size $n$.

2. A *column stochastic matrix*, also called *left stochastic matrix*, is a square matrix of nonnegative real numbers denoted as $C \in \mathbb{R}_{\geq 0}^{n \times n}$, with each column summing to 1, i.e., $\mathbb{1}^{\mathsf{T}} C = \mathbb{1}^{\mathsf{T}}$.

3. A *doubly stochastic matrix* is a square matrix $W \in \mathbb{R}_{\geq 0}^{n \times n}$ that is both row stochastic and column stochastic, i.e., $P\mathbb{1} = \mathbb{1}$ and $\mathbb{1}^{\mathsf{T}} W = \mathbb{1}^{\mathsf{T}}$.

**Consensus property of Row Stochastic matrix**   A row-stochastic matrix exhibits a consensus property. By the definition of row stochastic matrix $R$, we know $R\mathbb{1} = \mathbb{1}$. If the matrix $R$ is also primitive, the Perron-Frobenius theorem guarantees that all its other eigenvalues have a magnitude strictly less than 1. Consequently, the recursion

$$\boldsymbol{x}_{k+1} = R\boldsymbol{x}_k, \quad \text{where } \boldsymbol{x}_0 \in \mathbb{R}^{N \times d}. \tag{32}$$

converges to a consensus vector, $\boldsymbol{x}_\infty := \lim_{k \to \infty} \boldsymbol{x}_k$, where all elements of $\boldsymbol{x}_\infty$ are identical. However, unlike the doubly stochastic matrix case, $\boldsymbol{x}_\infty \neq \frac{1}{n}\mathbb{1}\mathbb{1}^{\mathsf{T}}\boldsymbol{x}_0$ in general. Suppose that the corresponding left eigenvector of $R$ with eigenvalue is $p$, that is. $p^{\mathsf{T}} R = p^{\mathsf{T}}$. Then, it is easy to show that

$$\boldsymbol{x}_\infty = \lim_{k \to \infty} R^k \boldsymbol{x}_0 = p\mathbb{1}^{\mathsf{T}}\boldsymbol{x}_0 \tag{33}$$

In general, the Perron vector $p$ is not a uniform vector $\frac{1}{n}\mathbb{1}$. Hence, the recursion of a row stochastic matrix typically converges to the consensus value of some weighted average of initial values.

To provide some concrete examples, consider the following row stochastic matrix:

$$R = \frac{1}{2}\begin{bmatrix} 1 & 0 & 1 & 0 \\ 1 & 1 & 0 & 0 \\ 1 & 0 & 1 & 0 \\ 1 & 0 & 0 & 1 \end{bmatrix}, \quad R^\infty = \frac{1}{2}\begin{bmatrix} 1 & 0 & 1 & 0 \\ 1 & 0 & 1 & 0 \\ 1 & 0 & 1 & 0 \\ 1 & 0 & 1 & 0 \end{bmatrix} \tag{34}$$

That is $p = [0.5, 0, 0.5, 0]$ in this case.

**Mass preservation property of Column Stochastic matrix**   A column-stochastic matrix exhibits properties markedly different from a row-stochastic matrix. While repeatedly multiplying a vector by a row-stochastic matrix leads to consensus, this is generally not true for a column-stochastic matrix.

Instead, column-stochastic matrices possess a crucial property we call mass (or information) preservation. To understand this property, consider the following recursion:

$$\boldsymbol{x}_{k+1} = C_k \boldsymbol{x}_k, \quad \text{where } \boldsymbol{x}_0 \in \mathbb{R}^{N \times d}. \tag{35}$$

Note the subscript in $C_k$ indicates that a different column-stochastic matrix can be applied at each step $k$ of the recursion. If we imagine that each element $\boldsymbol{x}_0(i)$ represents the mass of the $i-$th object, then the recursion $\boldsymbol{x}_{k+1} = C_k \boldsymbol{x}_k$ can be interpreted as a mass redistribution process where the total mass is conserved at every step. To see that, left multiplying the $\mathbb{1}^\mathsf{T}$ to both sides of above equation we get

$$\mathbb{1}^\mathsf{T} x^{k+1} = \mathbb{1}^\mathsf{T} C_k x^k = \mathbb{1}^\mathsf{T} x^k \tag{36}$$

By induction, we can conclude $\mathbb{1}^\mathsf{T} x^k = \mathbb{1}^\mathsf{T} x^0, \forall k$.

Finally, since a doubly-stochastic matrix is both row- and column-stochastic, it automatically inherits both of the properties discussed above. Combining the consensus-driving property (from being row-stochastic) with the mass-preservation property (from being column-stochastic) allows us to recover (unbiased) average consensus. The recursion converges to an average consensus, where every element of the final vector is equal to the average of the elements in the initial vector.

# D   Proof of the Convergence of FedAvg under Arbitrary Client Participation

This section presents a convergence analysis of the FedAvg algorithm with an arbitrary sampling/participation scheme. We focus on the strongly-convex case with a constant step size since this setting best illustrates the impact of client drift induced by local updates and bias introduced by non-uniform sampling. The following proof draws inspiration from and synthesizes several existing works [Wang and Ji, 2022, Koloskova et al., 2020, Li et al., 2020], adapting their insights to a decentralized framework. Leveraging this framework, we are able to present a more concise proof and provide a clearer conclusion. Unlike `FOCUS`, most FL algorithms require an extra bounded heterogeneity assumptions:

**Assumption 5** (Bounded Heterogeneity). *For any $x$ and the local cost function $f_i$, $\|\nabla f_i(x) - \nabla F(x)\| \leq \sigma_G$.*

We further introduce a quantity, denoted as $\delta_q^2$, to bound the discrepancy between the unbiased gradient average and that resulting from arbitrary distribution.

$$\left\| q^\mathsf{T} \nabla \boldsymbol{f}(\boldsymbol{x}) - u^\mathsf{T} \nabla \boldsymbol{f}(\boldsymbol{x})) \right\|^2 \leq \delta_q^2, \quad \forall \boldsymbol{x}. \tag{37}$$

where $u^\mathsf{T}$ is a uniform distribution vector: $\frac{1}{N}[0, 1, 1, \cdots, 1]$ and $q^\mathsf{T} = [0, q_1, q_2, \cdots, q_N]$, the one introduced in Assumption 1. This quantity $\delta_q^2$ must exist because $\delta_q^2 \leq \sigma_G^2$ due to Jensen's inequality $\left\| q^\mathsf{T} \nabla \boldsymbol{f}(\boldsymbol{x}) - \nabla F(\boldsymbol{x})) \right\|^2 \leq \sum_{i=1}^N q_i \|\nabla f_i(\bar{x}_k) - \nabla F(\bar{x}_k))\|^2 = \sigma_G^2$. If $q = u$, i.e., the uniform sampling case, $\delta_q^2 = 0$.

**Theorem 3** (Convergence of FedAvg Under Arbitrary Activation). *Under the assumption 1, 2, 3, and 5, when the learning rate satisfies $\eta \leq \min\{\frac{1}{3L}, \frac{1}{4(\tau-1)L}\}$, the limiting point of FedAvg satisfies:*

$$\limsup_{K \to \infty} \mathbb{E}\|\bar{x}_K - x^\star\|^2 \leq \underbrace{80\eta^2 \kappa^2 L^2 (\tau-1)^2 (\delta_q^2 + \sigma_G^2)}_{\text{client drift by local update}} + \underbrace{10\kappa\delta_q^2}_{\text{biased sampling}} + \underbrace{16\eta\tau^2/\mu(\delta_q^2 + \sigma_G^2)}_{\text{data heterogeneity}},$$

*where $\bar{x}_k = q^\mathsf{T} \boldsymbol{x}_k$, $x^\star$ is the optimal point of (1) and $\kappa = L/\mu$ is the condition number.* $\qquad \square$

**Remark of Theorem 3.** Each of these three terms possesses a distinct interpretation. Notably, when $\tau = 1$, indicating a single local update step, the first term, representing client drift introduced by local updates, vanishes. Both the first and third terms are scaled by the step size, $\eta$, implying that their magnitudes can be controlled and reduced below an arbitrary threshold, $\epsilon$, by employing a sufficiently small step size. The second term, however, is a constant related to the non-uniform sampling distribution and is, therefore, independent of the learning rate. This term underscores the significant impact of non-uniform sampling on the convergence behavior. Even under the simplified conditions of $\tau = 1$ and uniform sampling, FedAvg fails to converge to the optimal solution unless the objective functions are homogeneous. This observation aligns with previous findings that a diminishing learning rate is necessary for FedAvg to achieve exact convergence.

### D.1 Reformation and Mixing Matrices

First, we rewrite the FedAvg algorithm using the decentralized matrix notation as introduced before:

$$\boldsymbol{y}_{k+1} = R_k \boldsymbol{x}_k \tag{38}$$

$$\boldsymbol{x}_{k+1} = A_k(\boldsymbol{y}_{k+1} - \eta D_k \nabla \boldsymbol{f}(\boldsymbol{y}_{k+1})), \tag{39}$$

where both $R_k$ and $A_k$ are row-stochastic matrices, and they are some realizations of random matrices representing the arbitrary sampling and $D_k$ are the diagonal matrices with value 0 or 1 to control the turning on and off of clients. For $\tau$ local update, it satisfies

$$R_k = \begin{cases} R(S_r), & \text{if } k = r\tau + 1 \\ I, & \text{otherwsie.} \end{cases} \qquad A_k = \begin{cases} V(S_r), & \text{if } k = r\tau \\ I, & \text{otherwsie.} \end{cases} \tag{40}$$

See the definition of $R(S_r)$ and $V(S_r)$ in main context and example in Figure 6. A noteworthy observation from [Li et al., 2020] is that FedAvg can be equivalently reformulated without altering the trajectory of the server's model. This reformulation considers activating all devices, where each device pulls the model from the server and performs a local update, while maintaining the same set of contributing clients for averaging. See Figure 7 as an illustration. Swap the order of $x$ and $y$ update, we arrive at

$$\boldsymbol{y}_{k+1} = \boldsymbol{x}_k - \eta \nabla \boldsymbol{f}(\boldsymbol{x}_k) \tag{41}$$

$$\boldsymbol{x}_{k+1} = W_k \boldsymbol{y}_{k+1} \tag{42}$$

where $W_k$ is a row-stochastic matrix:

$$W_k = \begin{cases} I, & k \neq r\tau, & \forall r = 1, 2, \cdots \\ W(S_r), & k = r\tau. & \forall r = 1, 2, \cdots \end{cases} \tag{43}$$

To better understand the property of $W(S_r)$, we provide a few concrete examples of $W(S_r)$. Suppose there are 4 clients. Under the arbitrary participation case, each round the number of participated clients is not fixed. Maybe in one round, client 1 and 3 are sampled while in another round, clients 2, 3, and 4 are sampled. The corresponding matrices are

$$W_{\{1,3\}} = \begin{bmatrix} 0 & 1/2 & 0 & 1/2 & 0 \\ 0 & 1/2 & 0 & 1/2 & 0 \\ 0 & 1/2 & 0 & 1/2 & 0 \\ 0 & 1/2 & 0 & 1/2 & 0 \\ 0 & 1/2 & 0 & 1/2 & 0 \end{bmatrix}, \quad W_{\{2,3,4\}} = \begin{bmatrix} 0 & 0 & 1/3 & 1/3 & 1/3 \\ 0 & 0 & 1/3 & 1/3 & 1/3 \\ 0 & 0 & 1/3 & 1/3 & 1/3 \\ 0 & 0 & 1/3 & 1/3 & 1/3 \\ 0 & 0 & 1/3 & 1/3 & 1/3 \end{bmatrix} \tag{44}$$

It is crucial to observe that $W_{S_r}$ has identical rows for any possible subset $S_r$. Thus, it suffices to compute the expected value of the entries in any single row.

Now, we can state the property of $W(S_r)$. As the consequence of arbitrary participation assumption 1 and previous single row observation, we can show that

$$\bar{W} = \mathbb{E}_{S_r} W(S_r) = \begin{bmatrix} - & q^\mathsf{T} & - \\ - & q^\mathsf{T} & - \\ & \cdots & \\ - & q^\mathsf{T} & - \end{bmatrix} = \begin{bmatrix} 0 & q_1 & q_2 & \cdots & q_N \\ 0 & q_1 & q_2 & \cdots & q_N \\ \vdots & \vdots & \vdots & \vdots & \vdots \\ 0 & q_1 & q_2 & \cdots & q_N \end{bmatrix} \tag{45}$$

where the values $q_i$ is value defined in the assumption. It follows directly that this row vector is also a left eigenvector of $\mathbb{E} W_{S_r}$, i.e., $q^\mathsf{T} \mathbb{E} W_{S_r} = q^\mathsf{T}$, since $\sum_i q_i = 1$.

Pull Model:

$$R_{\{1,3\}} = \begin{bmatrix} 1 & 0 & 0 & 0 & 0 \\ 1 & 0 & 0 & 0 & 0 \\ 0 & 0 & 1 & 0 & 0 \\ 1 & 0 & 0 & 0 & 0 \\ 0 & 0 & 0 & 0 & 1 \end{bmatrix}, \quad A_{\{1,3\}} = \begin{bmatrix} 1 & 0 & 0 & 0 & 0 \\ 0 & 1 & 0 & 0 & 0 \\ 0 & 0 & 1 & 0 & 0 \\ 0 & 0 & 0 & 1 & 0 \\ 0 & 0 & 0 & 0 & 1 \end{bmatrix}$$

Push Model:

$$R_{\{1,3\}} = \begin{bmatrix} 1 & 0 & 0 & 0 & 0 \\ 0 & 1 & 0 & 0 & 0 \\ 0 & 0 & 1 & 0 & 0 \\ 0 & 0 & 0 & 1 & 0 \\ 0 & 0 & 0 & 0 & 1 \end{bmatrix}, \quad A_{\{1,3\}} = \begin{bmatrix} 0 & \frac{1}{2} & 0 & \frac{1}{2} & 0 \\ 0 & 1 & 0 & 0 & 0 \\ 0 & 0 & 1 & 0 & 0 \\ 0 & 0 & 0 & 1 & 0 \\ 0 & 0 & 0 & 0 & 1 \end{bmatrix}$$

Figure 6: Illustration of federated learning using a graph and mixing matrix. The top row depicts the pull model step, while the bottom row shows the push model and subsequent averaging step.

Pull Model:

$$R_{\mathrm{all}} = \begin{bmatrix} 1 & 0 & 0 & 0 & 0 \\ 1 & 0 & 0 & 0 & 0 \\ 1 & 0 & 0 & 0 & 0 \\ 1 & 0 & 0 & 0 & 0 \\ 1 & 0 & 0 & 0 & 0 \end{bmatrix}, \quad A_{\{1,3\}} = \begin{bmatrix} 1 & 0 & 0 & 0 & 0 \\ 0 & 1 & 0 & 0 & 0 \\ 0 & 0 & 1 & 0 & 0 \\ 0 & 0 & 0 & 1 & 0 \\ 0 & 0 & 0 & 0 & 1 \end{bmatrix}$$

Push Model:

$$R_{\mathrm{all}} = \begin{bmatrix} 1 & 0 & 0 & 0 & 0 \\ 0 & 1 & 0 & 0 & 0 \\ 0 & 0 & 1 & 0 & 0 \\ 0 & 0 & 0 & 1 & 0 \\ 0 & 0 & 0 & 0 & 1 \end{bmatrix}, \quad A_{\{1,3\}} = \begin{bmatrix} 0 & \frac{1}{2} & 0 & \frac{1}{2} & 0 \\ 0 & 1 & 0 & 0 & 0 \\ 0 & 0 & 1 & 0 & 0 \\ 0 & 0 & 0 & 1 & 0 \\ 0 & 0 & 0 & 0 & 1 \end{bmatrix}$$

Figure 7: An equivalent FedAvg algorithm as Figure 6 in terms of the server model. The difference is that all clients (virtually) pull the model and run the local update but server run the partially average.

### D.2 Convergence Proof

As previously discussed, we will analyze FedAvg in its decentralized form using the following simplified representation:

$$\boldsymbol{y}_{k+1} = \boldsymbol{x}_k - \eta \nabla \boldsymbol{f}(\boldsymbol{x}_k) \tag{46}$$
$$\boldsymbol{x}_{k+1} = W_k \boldsymbol{y}_{k+1} \tag{47}$$

To start with, we define the virtual weighted iterates $\bar{x}_k := q^\mathsf{T} \boldsymbol{x}_k$, recalling that vector $q$ is the averaging weights introduced in Assumption 1. The crucial observation is that conditional expectation $\mathbb{E}_{|\boldsymbol{y}_k} \bar{x}_k = q^\mathsf{T} \mathbb{E} W_k \boldsymbol{y}_k = q^\mathsf{T} \boldsymbol{y}_k$ holds for any $k$, including both local update step and model average step. When there is no ambiguity, we will just use $\mathbb{E}$ for conditional expectation instead of $\mathbb{E}_{|\boldsymbol{y}_k}$. Expanding the conditional expectation of $\mathbb{E} \|\bar{x}_{k+1} - x^\star\|^2$, we have

$$\begin{aligned} \mathbb{E} \|\bar{x}_{k+1} - x^\star\|^2 &= \mathbb{E} \|\bar{x}_{k+1} - \mathbb{E}\bar{x}_{k+1} + \mathbb{E}\bar{x}_{k+1} - x^\star\|^2 \\ &= \mathbb{E} \|\bar{x}_{k+1} - \mathbb{E}\bar{x}_{k+1}\|^2 + \|\mathbb{E}\bar{x}_{k+1} - x^\star\|^2 \\ &= \begin{cases} \left\|\bar{x}_k - \eta q^\mathsf{T} \nabla \boldsymbol{f}(\boldsymbol{x}_k) - x^\star\right\|^2, & k \neq r\tau \\ \mathbb{E} \left\|\bar{x}_{k+1} - q^\mathsf{T} \boldsymbol{y}_{k+1}\right\|^2 + \left\|\bar{x}_k - \eta q^\mathsf{T} \nabla \boldsymbol{f}(\boldsymbol{x}_k) - x^\star\right\|^2, & k = r\tau, \end{cases} \end{aligned} \tag{48}$$

where the first equality is because the cross term is zero and the second equality holds because $\mathbb{E} \|\bar{x}_{k+1} - q^\mathsf{T} \boldsymbol{y}_{k+1}\| = 0$ during the local update iterations ($k \neq r\tau$).

The subsequent proof follows a standard framework for analyzing decentralized algorithms. It initially establishes a descent lemma, showing that the virtual weighted iterates $\bar{x}_k$ progressively approach a neighborhood of the optimal solution in each iteration. Subsequently, a consensus lemma is established, showing that the individual client iterates, $x_{k,i}$, gradually converge towards this weighted iterate $\bar{x}_k$. Finally, by combining these two lemmas, we will derive the overall convergence theorem.

### D.2.1 Descent Lemma

**Lemma 1** (Descent Lemma of FedAvg). *Under the assumption 1, 2, and 3, the following inequality holds when the learning rate satisfies $\eta \leq \frac{1}{3L}$:*

$$\left\|\bar{x}_k - \eta q^\mathsf{T}\nabla \boldsymbol{f}(\boldsymbol{x}_k) - x^\star\right\|^2 \leq (1 - \frac{\eta\mu}{2})\|\bar{x}_k - x^\star\|^2 + \frac{5\eta L^2}{\mu}\sum_{i=1}^{N} q_i\|\bar{x}_k - x_{k,i}\|^2 + \frac{5\eta}{\mu}\delta_q^2, \quad (49)$$

*Proof:* To bound the common descent term $\|\bar{x}_k - \eta q^\mathsf{T}\nabla \boldsymbol{f}(\boldsymbol{x}_k) - x^\star\|^2$, we have

$$\begin{aligned}
&\left\|\bar{x}_k - \eta q^\mathsf{T}\nabla \boldsymbol{f}(\boldsymbol{x}_k) - x^\star\right\|^2 \\
=&\|\bar{x}_k - x^\star\|^2 + \eta^2\|q^\mathsf{T}\nabla \boldsymbol{f}(\boldsymbol{x}_k)\|^2 - 2\eta\langle \bar{x}_k - x^\star, q^\mathsf{T}\nabla \boldsymbol{f}(\boldsymbol{x}_k)\rangle \\
\leq&\|\bar{x}_k - x^\star\|^2 + 3\eta^2\left\|q^\mathsf{T}\nabla \boldsymbol{f}(\boldsymbol{x}_k) - q^\mathsf{T}\nabla \boldsymbol{f}(\mathbb{1}\bar{x}_k))\right\|^2 + 3\eta^2\left\|q^\mathsf{T}\nabla \boldsymbol{f}(\mathbb{1}\bar{x}_k)) - u^\mathsf{T}\nabla \boldsymbol{f}(\mathbb{1}\bar{x}_k))\right\|^2 \\
&+ 3\eta^2\|\nabla F(\bar{x}_k)\|^2 - 2\eta\langle \bar{x}_k - x^\star, q^\mathsf{T}\nabla \boldsymbol{f}(\boldsymbol{x}_k)\rangle \\
\leq&\|\bar{x}_k - x^\star\|^2 + 3\eta^2 L^2\sum_{i=1}^{N} q_i\|x_{k,i} - \bar{x}_k\|^2 + 3\eta^2\delta_q^2 + 6\eta^2 L\big(F(\bar{x}_k) - F(x^\star)\big) - 2\eta\langle \bar{x}_k - x^\star, q^\mathsf{T}\nabla \boldsymbol{f}(\boldsymbol{x}_k)\rangle,
\end{aligned}$$

$$(50)$$

where the first inequality results from Jensen's inequality, and the second inequality utilizes (37) and the consequence of $L-$ Lipschitz smooth condition with a convex function, we have $F(x) - F(x^\star) \geq \frac{1}{2L}\|\nabla F(x)\|^2, \forall x$.

Next, an upper bound for the cross term can be given by

$$\begin{aligned}
&-2\eta\langle \bar{x}_k - x^\star, q^\mathsf{T}\nabla \boldsymbol{f}(\boldsymbol{x}_k)\rangle \\
=&-2\eta\langle \bar{x}_k - x^\star, u^\mathsf{T}\nabla \boldsymbol{f}(\bar{\boldsymbol{x}}_k)\rangle + 2\eta\langle \bar{x}_k - x^\star, u^\mathsf{T}\nabla \boldsymbol{f}(\bar{\boldsymbol{x}}_k) - q^\mathsf{T}\nabla \boldsymbol{f}(\boldsymbol{x}_k)\rangle \\
\leq&-2\eta(F(\bar{x}_k) - F(x^\star) + \frac{\mu}{2}\|\bar{x}_k - x^\star\|^2) + \frac{\eta\mu}{2}\|\bar{x}_k - x^\star\|^2 + \frac{2\eta}{\mu}\|u^\mathsf{T}\nabla \boldsymbol{f}(\bar{\boldsymbol{x}}_k) - q^\mathsf{T}\nabla \boldsymbol{f}(\boldsymbol{x}_k)\|^2 \\
\leq&-\frac{\eta\mu}{2}\|\bar{x}_k - x^\star\|^2 - 2\eta\big(F(\bar{x}_k) - F(x^\star)\big) + \frac{4\eta}{\mu}\|q^\mathsf{T}\nabla \boldsymbol{f}(\bar{\boldsymbol{x}}_k) - q^\mathsf{T}\nabla \boldsymbol{f}(\boldsymbol{x}_k)\|^2 \\
&+ \frac{4\eta}{\mu}\|u^\mathsf{T}\nabla \boldsymbol{f}(\bar{\boldsymbol{x}}_k) - q^\mathsf{T}\nabla \boldsymbol{f}(\bar{\boldsymbol{x}}_k)\|^2 \\
\leq&-\frac{\eta\mu}{2}\|\bar{x}_k - x^\star\|^2 - 2\eta\big(F(\bar{x}_k) - F(x^\star)\big) + \frac{4\eta L^2}{\mu}\sum_{i=1}^{N} q_i\|x_{k,i} - \bar{x}_k\|^2 + \frac{4\eta}{\mu}\delta_q^2,
\end{aligned}$$

$$(51)$$

where the first inequality is obtained from Young's inequality $2\langle a, b\rangle \leq \epsilon\|a\|^2 + \frac{1}{\epsilon}\|b\|^2$ with $\epsilon = \mu/2$, the second inequality is due to Jensen's inequality, and the third inequality is obtained by (37).

Combining (50) and (51), we have

$$\begin{aligned}
&\left\|\bar{x}_k - \eta q^\mathsf{T}\nabla \boldsymbol{f}(\boldsymbol{x}_k) - x^\star\right\|^2 \\
\leq&(1 - \frac{\eta\mu}{2})\|\bar{x}_k - x^\star\|^2 + \eta\left(\frac{4L^2}{\mu} + 3\eta L^2\right)\sum_{i=1}^{N} q_i\|\bar{x}_k - x_{k,i}\|^2 + \left(\frac{4\eta}{\mu} + 3\eta^2\right)\delta_q^2 \\
&+ (6\eta^2 L - 2\eta)\big(F(\bar{x}_k) - F(x^\star)\big)
\end{aligned}$$

$$(52)$$

Letting $\eta \leq \min(\frac{1}{3L}, \frac{1}{3\mu}) = \frac{1}{3L}$, we further have

$$\left\|\bar{x}_k - \eta q^\mathsf{T}\nabla \boldsymbol{f}(\boldsymbol{x}_k) - x^\star\right\|^2 \leq (1 - \frac{\eta\mu}{2})\|\bar{x}_k - x^\star\|^2 + \frac{5\eta L^2}{\mu}\sum_{i=1}^{N} q_i\|\bar{x}_k - x_{k,i}\|^2 + \frac{5\eta}{\mu}\delta_q^2, \quad (53)$$

where we discarded the $(6\eta^2 L - 2\eta)\big(F(\bar{x}_k) - F(x^\star)\big)$ term since it is always negative. $\qquad\square$

### D.2.2 Consensus Lemma

**Lemma 2** (Consensus Error of FedAvg). *Under the assumption 1, 2, and 5, the following two (weighted) consensus errors hold for any iteration $k$ when the learning rate satisfies $\eta \leq \frac{1}{4(\tau-1)L}$:*

$$\sum_{i=1}^{N} q_i \|\bar{x}_k - x_{k,i}\|^2 \leq 8\eta^2(\tau-1)^2(\delta_q^2 + \sigma_G^2), \quad \forall k \tag{54}$$

$$\sum_{i=1}^{N} q_i \|\bar{y}_k - y_{k,i}\|^2 \leq 8\eta^2\tau^2(\delta_q^2 + \sigma_G^2), \quad \forall k \tag{55}$$

*Proof:* To evaluate the consensus error, the key observation is at the model average iteration, i.e., $k = r\tau$ that all clients' model parameters $x_{k,i}$ are the same. Hence, we can express the consensus error by referring back to that point:

$$\sum_{i=1}^{N} q_i \|\bar{x}_k - x_{k,i}\|^2 = \sum_{i=1}^{N} q_i \left\| x_{k_0} - \eta \sum_{k'=k_0}^{k-1} q^\mathsf{T}\nabla \boldsymbol{f}(\boldsymbol{x}_{k'}) - x_{k_0} + \eta \sum_{k'=k_0}^{k-1} \nabla f_i(x_{k',i}) \right\|^2$$

$$\leq \eta^2(\tau-1) \sum_{i=1}^{N} \sum_{k'=k_0}^{k-1} q_i \left\| q^\mathsf{T}\nabla \boldsymbol{f}(\boldsymbol{x}_{k'}) - \nabla f_i(x_{k',i}) \right\|^2, \tag{56}$$

where $k_0$ is the iteration that model averaging is performed, which can be calculated via $k_0 = \tau\lfloor\frac{k}{\tau}\rfloor$. The above inequality utilizes Jensen's inequality and observation that $k - k_0 \leq \tau - 1$. Then, we have

$$\left\| q^\mathsf{T}\nabla \boldsymbol{f}(\boldsymbol{x}_{k'}) - \nabla f_i(x_{k',i}) \right\|^2$$

$$\leq 4 \left\| q^\mathsf{T}\nabla \boldsymbol{f}(\boldsymbol{x}_{k'}) - q^\mathsf{T}\nabla \boldsymbol{f}(\mathbb{1}\bar{x}_{k'}) \right\|^2 + 4 \left\| q^\mathsf{T}\nabla \boldsymbol{f}(\mathbb{1}\bar{x}_{k'}) - u^\mathsf{T}\nabla \boldsymbol{f}(\mathbb{1}\bar{x}_{k'}) \right\|^2$$

$$+ 4 \left\| u^\mathsf{T}\nabla \boldsymbol{f}(\mathbb{1}\bar{x}_{k'}) - u^\mathsf{T}\nabla \boldsymbol{f}(\mathbb{1}x_{k',i}) \right\|^2 + 4 \left\| u^\mathsf{T}\nabla \boldsymbol{f}(\mathbb{1}x_{k',i}) - \nabla f_i(\mathbb{1}x_{k',i}) \right\|^2$$

$$\leq 4 \sum_{i'=1}^{N} q_{i'} \|\nabla f_{i'}(x_{k',i'}) - \nabla f_i(\bar{x}_{k'})\|^2 + 4L^2 \|\bar{x}_{k'} - x_{k',i}\|^2 + 4\delta_q^2 + 4\sigma_G^2$$

$$\leq 4L^2 \sum_{i'=1}^{N} q_{i'} \|x_{k',i'} - \bar{x}_{k'}\|^2 + 4L^2 \|\bar{x}_{k'} - x_{k',i}\|^2 + 4\delta_q^2 + 4\sigma_G^2 \tag{57}$$

where we plus and minus $q^\mathsf{T}\nabla \boldsymbol{f}(\mathbb{1}\bar{x}_{k'})$, $u^\mathsf{T}\nabla \boldsymbol{f}(\mathbb{1}\bar{x}_{k'})$ and $u^\mathsf{T}\nabla \boldsymbol{f}(\mathbb{1}x_{k',i})$ then apply Jensen's inequality. Plugging (57) back to (56), we have

$$\sum_{i=1}^{N} q_i \|\bar{x}_k - x_{k,i}\|^2 \leq \eta^2(\tau-1) \sum_{k'=k_0}^{k-1} \left( 8L^2 \sum_{i=1}^{N} q_i \|x_{k',i} - \bar{x}_{k'}\|^2 + 4\delta_q^2 + 4\sigma_G^2 \right) \tag{58}$$

Finally, we can establish a uniform bound for the consensus error using mathematical induction. Initially, note that $\sum_{i=1}^{N} q_i \|\bar{x}_{k_0} - x_{k_0,i}\|^2 = 0$. Now, assume that $\sum_{i=1}^{N} q_i \|\bar{x}_k - x_{k,i}\|^2 \leq \Delta$ for any $k \leq k_0 + \tau - 1$, then

$$\sum_{i=1}^{N} q_i \|\bar{x}_{k+\tau} - x_{k_0+\tau,i}\|^2 \leq 4\eta^2(\tau-1)^2(2L^2\Delta + \delta_q^2 + \sigma_G^2) = \Delta \tag{59}$$

It holds when $\Delta = \frac{4\eta^2(K-1)^2(\delta_q^2+\sigma_G^2)}{1-8\eta^2(K-1)^2L^2}$. If $\eta \leq \frac{1}{4(\tau-1)L}$, then $\Delta \leq 8\eta^2(\tau-1)^2(\delta_q^2 + \sigma_G^2)$. Hence, the uniform upper bound of consensus error is

$$\sum_{i=1}^{N} q_i \|\bar{x}_k - x_{k,i}\|^2 \leq 8\eta^2(\tau-1)^2(\delta_q^2 + \sigma_G^2) \tag{60}$$

This upper bound holds for $\bar{y}_k$ and $y_{k,i}$ similarly with only one difference that $\boldsymbol{y}_k$ has one more inner iteration before applying the $W_k$ compared to $\boldsymbol{x}_k$.

$$\sum_{i=1}^{N} q_i \|\bar{y}_k - y_{k,i}\|^2 \leq 8\eta^2\tau^2(\delta_q^2 + \sigma_G^2) \tag{61}$$

$\square$

### D.2.3 Proof of Convergence Theorem 3

*Proof:* Combining the above two lemmas, we conclude that for $k \neq r\tau$

$$\mathbb{E}\|\bar{x}_{k+1} - x^\star\|^2 \leq (1 - \eta\mu/2)\|\bar{x}_k - x^\star\|^2 + \frac{40\eta^3 L^2(\tau-1)^2}{\mu}(\delta_q^2 + \sigma_G^2) + \frac{5\eta}{\mu}\delta_q^2 \quad (62)$$

To establish the case $k = r\tau$, we need to consider the variance after the local update is done. Through previous established the consensus lemma, it is easy to verify that

$$\begin{aligned}
\mathbb{E}\left\|\bar{x}_{k+1} - q^\mathsf{T}\boldsymbol{y}_{k+1}\right\|^2 &= \mathbb{E}\left\|q^\mathsf{T}W_{s_n}\boldsymbol{y}_{k+1} - q^\mathsf{T}\boldsymbol{y}_{k+1}\right\|^2 \\
&= \mathbb{E}\left\|W_{s_n}[0,:]\boldsymbol{y}_{k+1} - q^\mathsf{T}\boldsymbol{y}_{k+1}\right\|^2 \\
&\leq \sum_{i=1}^N q_i\left\|y_{k+1,i} - q^\mathsf{T}\boldsymbol{y}_{k+1}\right\|^2 \\
&\leq 8\eta^2\tau^2(\delta_q^2 + \sigma_G^2),
\end{aligned}$$

where the first equality holds because any row in the $W_{S_n}$ is the same, the first inequality applies Jensen's inequality and the last inequality utilizes the consensus lemma. Substituting back to (48), we have for $k \neq r\tau$

$$\mathbb{E}\|\bar{x}_{k+1} - x^\star\|^2 \leq (1 - \eta\mu/2)\|\bar{x}_k - x^\star\|^2 + \frac{40\eta^3 L^2(\tau-1)^2}{\mu}(\delta_q^2 + \sigma_G^2) + \frac{5\eta}{\mu}\delta_q^2 + 8\eta^2 K^2(\delta_q^2 + \sigma_G^2)$$

$$(63)$$

We can simplify above two recursion as

$$\begin{aligned}
A_{k+1} &\leq (1-\alpha)A_k + B, & k \neq r\tau && (64) \\
A_{k+1} &\leq (1-\alpha)A_k + B + C, & k = r\tau && (65)
\end{aligned}$$

where $A_k = \mathbb{E}\|\bar{x}_k - x^\star\|^2$, $B = \frac{40\eta^3 L^2(\tau-1)^2}{\mu}(\delta_q^2 + \sigma_G^2) + \frac{5\eta}{\mu}\delta_q^2$, and $C = 8\eta^2 K^2(\delta_q^2 + \sigma_G^2)$. Making it a $K$-step recursion together, we have

$$A_K \leq (1-\alpha)^K A_0 + \sum_{k'=0}^K (1-\alpha)^{k'} B + \sum_{k'=0}^{\lfloor K/\tau \rfloor} (1-\alpha)^{k'} C \quad (66)$$

Letting $K \to \infty$ and substituting back, we conclude

$$\limsup_{K\to\infty} \mathbb{E}\|\bar{x}_K - x^\star\|^2 \leq \underbrace{80\eta^2\kappa^2 L^2(\tau-1)^2(\delta_q^2 + \sigma_G^2)}_{\text{client drift by local update}} + \underbrace{10\kappa\delta_q^2}_{\text{biased sampling}} + \underbrace{16\eta\tau^2/\mu(\delta_q^2 + \sigma_G^2)}_{\text{data heterogeneity}},$$

$$(67)$$

where we introduce the conditional number $\kappa = L/\mu$. $\qquad\square$

As we discussed in the main context, these three terms have their own meanings. We can easily establish the following corollaries. Note $\delta_q^2 = 0$ under the uniform sampling case. We have

**Corollary 1** (FedAvg Under the Uniform Sampling). *Under the same conditions and assumptions as theorem 3, the convergence of FedAvg with uniform sampling satisfies*

$$\limsup_{K\to\infty} \mathbb{E}\|\bar{x}_K - x^\star\|^2 \leq 80\eta^2\kappa^2 L^2(\tau-1)^2\sigma_G^2 + 16\eta\tau^2/\mu\sigma_G^2 \quad (68)$$

**Corollary 2** (FedAvg Under the Uniform Sampling and Single Local Update). *Under the same conditions and assumptions as theorem 3, the convergence of FedAvg with uniform sampling and $\tau = 1$ satisfies*

$$\limsup_{K\to\infty} \mathbb{E}\|\bar{x}_K - x^\star\|^2 \leq 16\eta\tau^2/\mu\sigma_G^2 \quad (69)$$

Lastly, if the function is homogeneous among $f_i$, it implies $\sigma_G^2 = 0$. Further notice $\delta_q^2 \leq \sigma_G^2 = 0$.

**Corollary 3** (FedAvg with Homogeneous Functions). *Under the same conditions of theorem 3, FedAvg can converge exactly when $f_i$ is homogeneous*

$$\lim_{K\to\infty} \mathbb{E}\|\bar{x}_K - x^\star\|^2 = 0 \quad (70)$$

Notably, Corollary 3 holds without requiring the assumptions of $\tau = 1$ or uniform sampling. This is intuitive because arbitrary sampling becomes irrelevant in the case of homogeneous functions.

# E Proof of the Convergence of `FOCUS`

## E.1 Reformulate the Recursion

Similar to the proof of FedAvg under arbitrary client participation, we first rewrite the recursion of `FOCUS` so that it is easier to show the proof. Recall that the original matrix-vector recursion is

$$\boldsymbol{x}_{k+1} = R_k(\boldsymbol{x}_k - \eta D_k \boldsymbol{y}_k) \tag{71}$$

$$\boldsymbol{y}_{k+1} = C_k\big(\boldsymbol{y}_k + \nabla \boldsymbol{f}(\boldsymbol{x}_{k+1}) - \nabla \boldsymbol{f}(\boldsymbol{x}_k)\big) \tag{72}$$

This form is not easy to analyze when noticing the following pattern on the mixing matrix choices:

| Iter. $k$ | 0 | 1 | 2 | $\cdots$ | $\tau$ | $\tau+1$ | $\cdots$ | $r\tau-1$ | $r\tau$ | $r\tau+1$ | $\cdots$ |
|---|---|---|---|---|---|---|---|---|---|---|---|
| $R_k$ | Init. | $R(S_1)$ | $I$ | $\cdots$ | $I$ | $R(S_2)$ | $\cdots$ | $I$ | $R(S_r)$ | $I$ | $\cdots$ |
| $C_k$ | Init. | $I$ | $I$ | $\cdots$ | $C(S_1)$ | $I$ | $\cdots$ | $C(S_{r-1})$ | $I$ | $I$ | $\cdots$ |

$R(S_r)$ and $C(S_r)$ are not applied at the same iteration. Even worse, $R(S_r)$ and $C(S_r)$ are random variables and depend on each other. To avoid these difficulties, we switch the order of $x-$ and $y-$update and get the following equivalent form:

$$\boldsymbol{y}_{k+1} = C_k\big(\boldsymbol{y}_k + \nabla \boldsymbol{f}(\boldsymbol{x}_k) - \nabla \boldsymbol{f}(\boldsymbol{x}_{k-1})\big) \tag{73}$$

$$\boldsymbol{x}_{k+1} = R_k(\boldsymbol{x}_k - \eta D_k \boldsymbol{y}_{k+1}) \tag{74}$$

Notice the subscript's modification. The initial condition becomes $\boldsymbol{y}_0 = \nabla \boldsymbol{f}(\boldsymbol{x}_{-1})$[3] and $\boldsymbol{x}_{-1} = \boldsymbol{x}_0$, which can be any values. Now, the matrices follow this new pattern

| Iter. $k$ | -1 | 0 | 1 | 2 | $\cdots$ | $\tau$ | $\tau+1$ | $\cdots$ | $r\tau-1$ | $r\tau$ | $r\tau+1$ | $\cdots$ |
|---|---|---|---|---|---|---|---|---|---|---|---|---|---|
| $C_k$ | - | Init. | $I$ | $I$ | $\cdots$ | $C(S_1)$ | $I$ | $\cdots$ | $I$ | $C(S_{r-1})$ | $I$ | $\cdots$ |
| $R_k$ | Init. | $R(S_1)$ | $I$ | $I$ | $\cdots$ | $R(S_2)$ | $I$ | $\cdots$ | $I$ | $R(S_r)$ | $I$ | $\cdots$ |

With this shift, both the row stochastic matrix ($R$) and column stochastic matrix ($C$) operations are applied within the same iteration. However, it is crucial to note that these operations do not correspond to the same indices of sampled clients, i.e. $S_r$ versus $S_{r-1}$. To further simplify the analysis, we can leverage a technique similar to the one used in the FedAvg proof: considering the collecting full set of client $y$ rather than just a subset. This is valid because the $y_{k,i}$ of non-participated clients are effectively zero. Consequently, we no longer need to take care about the correlation between R and C, significantly simplifying the analysis.

$$\boldsymbol{y}_{k+1} = C_{k,\text{all}}\big(\boldsymbol{y}_k + \nabla \boldsymbol{f}(\boldsymbol{x}_k) - \nabla \boldsymbol{f}(\boldsymbol{x}_{k-1})\big) \tag{75}$$

$$\boldsymbol{x}_{k+1} = R_k(\boldsymbol{x}_k - \eta D_k \boldsymbol{y}_{k+1}), \tag{76}$$

where the definition of $C_{k,\text{all}}$ are

$$C_{k,\text{all}} = \begin{cases} I & \text{if } k \neq r\tau \\ \begin{bmatrix} 1 & 1 & \cdots & 1 \\ 0 & 0 & \cdots & 0 \\ & & \cdots & \\ 0 & 0 & \cdots & 0 \end{bmatrix} & \text{if } k = r\tau \end{cases} \tag{77}$$

**The rest of proof will use this new form** (75) **-** (76)**.**

## E.2 Useful Observations

Before we proceed with the proof, there are a few critical observations.

Introducing a server index selecting vector $u_R = [1, 0, 0, \cdots, 0]$, it is straightforward to verify that it is the left-eigenvector of $R_k$ for all $k$:

$$u_R^\mathsf{T} R_k = u_R^\mathsf{T}, \quad \forall k. \tag{78}$$

---

[3] We do not really need to calculate the value of $\boldsymbol{y}_0$ since it will be canceled out in the first iteration.

Now we denote the $\bar{x}_k = u_R^\mathsf{T} \boldsymbol{x}_k$ and $\bar{y}_k = u_R^\mathsf{T} \boldsymbol{y}_k$, which can be interpreted as the server's model parameters and gradient tracker. Utilizing the eigenvector properties and definition of $R_k$ and $D_k$, we obtain

$$\bar{x}_{k+1} = \begin{cases} \bar{x}_k & k \neq r\tau \\ \bar{x}_k - \eta \bar{y}_{k+1} & k = r\tau \end{cases}, \qquad \bar{y}_{k+1} = \begin{cases} \bar{y}_k & k \neq r\tau \\ \mathbb{1}^\mathsf{T} \nabla \boldsymbol{f}(\boldsymbol{x}_k) & k = r\tau \end{cases}, \tag{79}$$

where $\bar{y}_{r\tau+1} = \mathbb{1}^\mathsf{T} \nabla \boldsymbol{f}(\boldsymbol{x}_{r\tau})$ is due to the tracking property that

$$\mathbb{1}^\mathsf{T} \boldsymbol{y}_{k+1} = \mathbb{1}^\mathsf{T} C_k \big( \boldsymbol{y}_k + \nabla \boldsymbol{f}(\boldsymbol{x}_k) - \nabla \boldsymbol{f}(\boldsymbol{x}_{k-1}) \big) = \cdots = \mathbb{1}^\mathsf{T} \nabla \boldsymbol{f}(\boldsymbol{x}_k), \tag{80}$$

and the client's $y_{r\tau+1,i} = 0$ for any clients. The main difficulty of the analysis lies in the iteration $r\tau$ to $r\tau + 1$, i.e., the gradient collecting and model pulling step. Given the information before step $r\tau$, it can be easily verified that

$$\mathbb{E} \left\| \boldsymbol{x}_{r\tau+1} - \boldsymbol{x}_{r\tau} \right\|_F^2 = \sum_{i=1}^N q_i \| \bar{x}_{r\tau+1} - x_{r\tau,i} \|^2 := \| \mathbb{1}\bar{x}_{r\tau+1} - \boldsymbol{x}_{r\tau} \|_Q^2 \tag{81}$$

$$\mathbb{E} \left\| \mathbb{1}\bar{x}_{r\tau+1} - \boldsymbol{x}_{r\tau+1} \right\|_F^2 = \sum_{i=1}^N (1 - q_i) \| \bar{x}_{r\tau+1} - x_{r\tau,i} \|^2 := \| \mathbb{1}\bar{x}_{r\tau+1} - \boldsymbol{x}_{r\tau} \|_{I-Q}^2, \tag{82}$$

where the first equation means the difference between the model before pulling and the server's model, and the second equation means the difference between the clients' models after pulling and the server's model. The last observation is the difference between model update

$$x_{(r+1)\tau,i} - x_{r\tau+1,i} = \begin{cases} 0 & \text{if } i \notin S_r \\ \displaystyle\sum_{k=r\tau+2}^{(r+1)\tau} y_{k,i} & \text{if } i \in S_r \end{cases}, \tag{83}$$

$$y_{r\tau+k'+1,i} = \begin{cases} 0 & \text{if } i \notin S_r \\ \nabla f_i(x_{r\tau+k',i}) - \nabla f_i(x_{r\tau,i}) & \text{if } i \in S_r \end{cases}, \tag{84}$$

Using the client-only notation: $\hat{\boldsymbol{x}}_k := [x_{k,1}; x_{k,2}; \cdots ; x_{k,N}] \in \mathbb{R}^{N \times d}$, we have the compact form

$$\hat{\boldsymbol{x}}_{(r+1)\tau} - \hat{\boldsymbol{x}}_{r\tau+1} = D_r \sum_{k=r\tau+2}^{(r+1)\tau} \hat{\boldsymbol{y}}_k \tag{85}$$

$$\hat{\boldsymbol{y}}_{r\tau+k'+1} = D_r \Big( \nabla \boldsymbol{f}(\hat{\boldsymbol{x}}_{r\tau+k'}) - \nabla \boldsymbol{f}(\hat{\boldsymbol{x}}_{r\tau}) \Big) \tag{86}$$

### E.3 Descent Lemma for `FOCUS`

**Lemma 3** (Descent Lemma for `FOCUS`). *Under assumptions 1 (arbitrary client participation), 2 (L-smooth) and 3 ($\mu$-strongly convex), if the learning rate $\eta \leq \frac{1}{3L(\tau-1)}$, the conditional expectation of server's error can be bounded as*

$$\mathbb{E}_{r\tau+1} \left\| \bar{x}_{(r+1)\tau+1} - x^\star \right\|^2 \leq \left( 1 - \frac{1}{2}\eta\mu N \right) \| \bar{x}_{r\tau+1} - x^\star \|^2 - 2\eta N(1 - 2\eta NL)\big( F(\bar{x}_{r\tau+1}) - F(x^\star) \big)$$

$$+ \frac{8\eta L^2 N}{\mu} \| \mathbb{1}\bar{x}_{r\tau+1} - \boldsymbol{x}_{r\tau} \|_F^2, \tag{87}$$

*where $x^\star$ is the optimal point.*

Note that this lemma can only be used for the strongly-convex case, which is the most complicated case. Non-convex proof is simpler than the strongly convex one, and a similar idea can be applied there, so we make this lemma outstanding here.

*Proof of Lemma 3:*

The server's error recursion from the $(r+1)$-th round to the $r$-th round is

$$\mathbb{E} \left\| \bar{x}_{(r+1)\tau+1} - x^\star \right\|^2$$

$$=\mathbb{E}\left\|\bar{x}_{r\tau+1}-x^{\star}-\eta\mathbb{1}^{\mathsf{T}}\nabla\boldsymbol{f}(\boldsymbol{x}_{(r+1)\tau})\right\|^{2}$$

$$=\|\bar{x}_{r\tau+1}-x^{\star}\|^{2}-2\eta\mathbb{E}\left\langle\bar{x}_{r\tau+1}-x^{\star},\mathbb{1}^{\mathsf{T}}\nabla\boldsymbol{f}(\boldsymbol{x}_{(r+1)\tau})\right\rangle+\eta^{2}\mathbb{E}\left\|\mathbb{1}^{\mathsf{T}}\nabla\boldsymbol{f}(\boldsymbol{x}_{(r+1)\tau})\right\|^{2}$$

$$\leq\|\bar{x}_{r\tau+1}-x^{\star}\|^{2}-2\eta\mathbb{E}\left\langle\bar{x}_{r\tau+1}-x^{\star},\mathbb{1}^{\mathsf{T}}\nabla\boldsymbol{f}(\boldsymbol{x}_{(r+1)\tau})\right\rangle+2\eta^{2}\mathbb{E}\left\|\mathbb{1}^{\mathsf{T}}\nabla\boldsymbol{f}(\boldsymbol{x}_{(r+1)\tau})-\mathbb{1}^{\mathsf{T}}\nabla\boldsymbol{f}(\mathbb{1}\bar{x}_{r\tau+1})\right\|^{2}$$
$$+2\eta^{2}N^{2}\|\nabla F(\bar{x}_{r\tau+1})\|^{2}$$

$$\leq\|\bar{x}_{r\tau+1}-x^{\star}\|^{2}-2\eta\mathbb{E}\left\langle\bar{x}_{r\tau+1}-x^{\star},\mathbb{1}^{\mathsf{T}}\nabla\boldsymbol{f}(\boldsymbol{x}_{(r+1)\tau})\right\rangle+2\eta^{2}\mathbb{E}\left\|\mathbb{1}^{\mathsf{T}}\nabla\boldsymbol{f}(\boldsymbol{x}_{(r+1)\tau})-\mathbb{1}^{\mathsf{T}}\nabla\boldsymbol{f}(\mathbb{1}\bar{x}_{r\tau+1})\right\|^{2}$$
$$+4\eta^{2}N^{2}L\big(F(\bar{x}_{r\tau+1})-F(x^{\star})\big),\tag{88}$$

where the first inequality is obtained by Jensen's inequality and the second inequality utilizes the Lipschitz condition with convexity $\frac{1}{2L}\|\nabla F(x)\|^{2}\leq F(x)-F(x^{\star})$. The cross term can be bounded as

$$-2\eta\mathbb{E}\left\langle\bar{x}_{r\tau+1}-x^{\star},\mathbb{1}^{\mathsf{T}}\nabla\boldsymbol{f}(\boldsymbol{x}_{(r+1)\tau})\right\rangle$$

$$=-2\eta\left\langle\bar{x}_{r\tau+1}-x^{\star},N\nabla F(\bar{x}_{r\tau+1})\right\rangle-2\eta\mathbb{E}\left\langle\bar{x}_{r\tau+1}-x^{\star},\mathbb{1}^{\mathsf{T}}\nabla\boldsymbol{f}(\boldsymbol{x}_{(r+1)\tau})-\mathbb{1}^{\mathsf{T}}\nabla\boldsymbol{f}(\mathbb{1}\bar{x}_{r\tau+1})\right\rangle$$

$$\leq-2\eta N(F(\bar{x}_{r\tau+1})-F(x^{\star})+\frac{\mu}{2}\|\bar{x}_{r\tau+1}-x^{\star}\|^{2})+\eta\epsilon\|\bar{x}_{r\tau+1}-x^{\star}\|^{2}$$
$$+\frac{\eta}{\epsilon}\mathbb{E}\left\|\mathbb{1}^{\mathsf{T}}\nabla\boldsymbol{f}(\boldsymbol{x}_{(r+1)\tau})-\mathbb{1}^{\mathsf{T}}\nabla\boldsymbol{f}(\mathbb{1}\bar{x}_{r\tau+1})\right\|^{2}$$

$$\leq-2\eta N(F(\bar{x}_{r\tau+1})-F(x^{\star}))-\frac{\eta\mu N}{2}\|\bar{x}_{r\tau+1}-x^{\star}\|^{2}+\frac{2\eta}{\mu}\mathbb{E}\left\|\mathbb{1}^{\mathsf{T}}\nabla\boldsymbol{f}(\boldsymbol{x}_{(r+1)\tau})-\mathbb{1}^{\mathsf{T}}\nabla\boldsymbol{f}(\mathbb{1}\bar{x}_{r\tau+1})\right\|^{2},\tag{89}$$

where the first inequality utilizes Young's inequality with $\epsilon$, and we set $\epsilon=\mu/2$ in the second inequality.

Plugging (89) into (88), we have

$$\mathbb{E}\left\|\bar{x}_{(r+1)\tau+1}-x^{\star}\right\|^{2}\leq\left(1-\frac{\eta\mu N}{2}\right)\|\bar{x}_{r\tau+1}-x^{\star}\|^{2}-2\eta N(1-2\eta NL)\big(F(\bar{x}_{r\tau+1})-F(x^{\star})\big)$$
$$+2\eta\left(\frac{1}{\mu}+\eta\right)\mathbb{E}\left\|\mathbb{1}^{\mathsf{T}}\nabla\boldsymbol{f}(\boldsymbol{x}_{(r+1)\tau})-\mathbb{1}^{\mathsf{T}}\nabla\boldsymbol{f}(\mathbb{1}\bar{x}_{r\tau+1})\right\|^{2}\tag{90}$$

Next, we focus on this gradient difference term:

$$\mathbb{E}\left\|\mathbb{1}^{\mathsf{T}}\nabla\boldsymbol{f}(\mathbb{1}\bar{x}_{r\tau+1})-\mathbb{1}^{\mathsf{T}}\nabla\boldsymbol{f}(\boldsymbol{x}_{(r+1)\tau})\right\|^{2}$$

$$\leq L^{2}N\mathbb{E}\|\mathbb{1}\bar{x}_{r\tau+1}-\boldsymbol{x}_{(r+1)\tau}\|_{F}^{2}$$

$$=L^{2}N\left\|\mathbb{1}\bar{x}_{r\tau+1}-\boldsymbol{x}_{r\tau+1}+\eta\sum_{k=r\tau+2}^{(r+1)\tau}D_{k}\boldsymbol{y}_{k}\right\|_{F}^{2}$$

$$\leq2L^{2}N\|\mathbb{1}\bar{x}_{r\tau+1}-\boldsymbol{x}_{r\tau+1}\|^{2}+2L^{2}N\left\|\eta\sum_{k=r\tau+2}^{(r+1)\tau}D_{k}\boldsymbol{y}_{k}\right\|_{F}^{2}$$

$$\leq2L^{2}N\|\mathbb{1}\bar{x}_{r\tau+1}-\boldsymbol{x}_{r\tau+1}\|^{2}+2(\tau-1)\eta^{2}L^{2}N\sum_{k=r\tau+2}^{(r+1)\tau}\|D_{k}\boldsymbol{y}_{k}\|_{F}^{2},\tag{91}$$

where the above two inequalities use Jensen's inequality $\|\sum_{i=1}^{N}a_{i}\|^{2}\leq N\sum_{i=1}^{N}\|a_{i}\|^{2}$.

Lastly, we need to bound $y_{k,i}$ by using $L$-Lipschitz assumption and Jensen's inequality. We just need to focus on the index $i$ that is the index among the sampled clients otherwise $y_{k,i}=0,\ \forall i\notin S_{r}$.

$$\|y_{k,i}\|^{2}=\|\nabla f_{i}(x_{k-1,i})-\nabla f_{i}(x_{r\tau,i})\|^{2}$$

$$\leq2\|\nabla f_{i}(x_{k-1,i})-\nabla f_{i}(x_{r\tau+1,i})\|^{2}+2\|\nabla f_{i}(x_{r\tau+1,i})-\nabla f_{i}(x_{r\tau,i})\|^{2}$$

$$\leq2L^{2}\|x_{k-1,i}-x_{r\tau+1,i}\|^{2}+2L^{2}\|x_{r\tau+1,i}-x_{r\tau,i}\|^{2}$$

$$\leq2\eta^{2}L^{2}(k-r\tau-2)\sum_{k'=r\tau+2}^{k}\|y_{k',i}\|^{2}+2L^{2}\|x_{r\tau+1,i}-x_{r\tau,i}\|^{2}$$

$$\leq 2\eta^2 L^2(\tau-1) \sum_{k'=r\tau+2}^{(r+1)\tau} \|y_{k',i}\|^2 + 2L^2\|x_{r\tau+1,i} - x_{r\tau,i}\|^2, \tag{92}$$

where, in the last inequality, we just expand the non-negative term to the maximum difference cases. Hence, taking another summation of $k$ from $r\tau + 2$ to $(r+1)\tau$, we obtain

$$\left(1 - 2\eta^2 L^2(\tau-1)\right) \sum_{k=r\tau+2}^{(r+1)\tau} \|y_{k,i}\|^2 \leq 2(\tau-1)L^2\|x_{r\tau+1,i} - x_{r\tau,i}\|^2 \tag{93}$$

When $\eta \leq \frac{1}{2L\sqrt{\tau-1}}$, we conclude

$$\sum_{k=r\tau+2}^{(r+1)\tau} \|y_{k,i}\|^2 \leq 4(\tau-1)L^2\|x_{r\tau+1,i} - x_{r\tau,i}\|^2 = 4(\tau-1)L^2\|\bar{x}_{r\tau+1} - x_{r\tau,i}\|^2 \tag{94}$$

Plugging (94) back to (91), we obtain

$$\mathbb{E} \left\| \mathbb{1}^\mathsf{T}\nabla f(\mathbb{1}\bar{x}_{r\tau+1}) - \mathbb{1}^\mathsf{T}\nabla f(\boldsymbol{x}_{(r+1)\tau}) \right\|^2$$
$$\leq 2L^2 N \|\mathbb{1}\bar{x}_{r\tau+1} - \boldsymbol{x}_{r\tau}\|^2_{(I-Q)} + 8\eta^2(\tau-1)^2 L^4 N \|\mathbb{1}\bar{x}_{r\tau+1} - \boldsymbol{x}_{r\tau}\|^2_Q, \tag{95}$$

where the first equation means the difference between the model before pulling and the server's model, and the second equation means the difference between the clients' models after pulling and the server's model. The weighted $Q$ and $I - Q$ (ref. (81)-(82)) are strictly smaller than 1, which can make the convergence proof tighter. But for simplicity, we just loosen it to 1 and got

$$\mathbb{E} \left\| \mathbb{1}^\mathsf{T}\nabla f(\mathbb{1}\bar{x}_{r\tau+1}) - \mathbb{1}^\mathsf{T}\nabla f(\boldsymbol{x}_{(r+1)\tau}) \right\|^2 \leq 2L^2 N \|\mathbb{1}\bar{x}_{r\tau+1} - \boldsymbol{x}_{r\tau}\|^2 + 8\eta^2(\tau-1)^2 L^4 N \|\mathbb{1}\bar{x}_{r\tau+1} - \boldsymbol{x}_{r\tau}\|^2$$
$$\leq 3L^2 N \|\mathbb{1}\bar{x}_{r\tau+1} - \boldsymbol{x}_{r\tau}\|^2 \tag{96}$$

where the last inequality holds when $\eta \leq \frac{1}{3(\tau-1)L}$. $\qquad\square$

### E.4 Consensus Lemma for FOCUS

**Lemma 4** (Consensus Lemma for FOCUS). *Under assumptions 1 and 2, if the learning rate* $\eta \leq \min\left\{ \frac{q_{\min}^{3/2}}{8L\sqrt{N}}, \frac{q_{\min}^{3/2}}{8L(\tau-1)} \right\}$, *the difference between the server's global model and the client's local model can be bounded as*

$$\mathbb{E} \|\mathbb{1}\bar{x}_{r\tau+1} - \boldsymbol{x}_{r\tau}\|^2_F \leq \left(1 - \frac{q_{\min}}{3}\right) \left\|\mathbb{1}\bar{x}_{(r-1)\tau+1} - \boldsymbol{x}_{(r-1)\tau}\right\|^2_F + \frac{16\eta^2 N^2}{3q_{\min}} \|\nabla F(\bar{x}_{r\tau+1})\|^2 \tag{97}$$

*Note that this consensus lemma is not related to any convex or strongly convex properties, so it can also be applied in nonconvex cases.*

*Proof of Lemma 4:*

Here, we focus on how different the server's model at the $r$-round and the client's model just before pulling the model from the server. Taking the conditional expectation, we have

$$\mathbb{E} \|\mathbb{1}\bar{x}_{r\tau+1} - \boldsymbol{x}_{r\tau}\|^2_F$$

$$= \mathbb{E} \left\| \mathbb{1}\bar{x}_{(r-1)\tau+1} - \eta\mathbb{1}\bar{y}_{r\tau+1} - \boldsymbol{x}_{(r-1)\tau+1} + \eta \sum_{k=(r-1)\tau+2}^{r\tau} D_k \boldsymbol{y}_k \right\|^2_F$$

$$\leq \frac{1}{\rho} \mathbb{E} \left\| \mathbb{1}\bar{x}_{(r-1)\tau+1} - \boldsymbol{x}_{(r-1)\tau+1} \right\|^2_F + \frac{2\eta^2}{1-\rho} \mathbb{E} \|\mathbb{1}\bar{y}_{r\tau+1}\|^2 + \frac{2\eta^2}{1-\rho} \mathbb{E} \left\| \sum_{k=(r-1)\tau+2}^{r\tau} D_k \boldsymbol{y}_k \right\|^2, \tag{98}$$

where the inequality above uses Jensen's inequality $\|a + b\|^2 \leq \frac{1}{\rho}\|a\|^2 + \frac{1}{1-\rho}\|b\|^2$ and $\rho$ here can be any value between 0 and 1 (exclusively). For each term, we know

$$\mathbb{E} \left\| \mathbb{1}\bar{x}_{(r-1)\tau+1} - \boldsymbol{x}_{(r-1)\tau+1} \right\|^2_F = \sum_{i=1}^N (1-q_i)\|\bar{x}_{(r-1)\tau+1} - x_{(r-1)\tau,i}\|^2$$

$$\leq (1 - q_{\min}) \left\| \mathbb{1}\bar{x}_{(r-1)\tau+1} - \boldsymbol{x}_{(r-1)\tau} \right\|_F^2 \tag{99}$$

and the second term

$$
\begin{aligned}
\left\| \mathbb{1}\bar{y}_{r\tau+1} \right\|^2 &\leq 2 \left\| \mathbb{1}^\mathsf{T}\nabla\boldsymbol{f}(\boldsymbol{x}_{r\tau}) - \mathbb{1}^\mathsf{T}\nabla\boldsymbol{f}(\mathbb{1}\bar{x}_{r\tau+1}) \right\|^2 + 2 \left\| N\nabla F(\bar{x}_{r\tau+1}) \right\|^2 \\
&\leq 2NL^2 \|\mathbb{1}\bar{x}_{r\tau+1} - \boldsymbol{x}_{r\tau}\|_F^2 + 2 \left\| N\nabla F(\bar{x}_{r\tau+1}) \right\|^2
\end{aligned} \tag{100}
$$

and the third term

$$
\begin{aligned}
\mathbb{E} \left\| \sum_{k=(r-1)\tau+2}^{r\tau} D_k \boldsymbol{y}_k \right\|^2 &\leq (\tau-1)\mathbb{E}\sum_{k=r\tau+2}^{(r+1)\tau} \|D_k \boldsymbol{y}_k\|^2 \\
&\leq 4(\tau-1)^2 L^2 \sum_{i=1}^{N} q_i \|\bar{x}_{r\tau+1} - x_{r\tau,i}\|^2 \\
&\leq 4(\tau-1)^2 L^2 \|\mathbb{1}\bar{x}_{r\tau+1} - \boldsymbol{x}_{r\tau}\|^2
\end{aligned} \tag{101}
$$

Putting (99) (100), (101) back to (98) and selecting $\rho = \frac{1-q_{\min}}{1-q_{\min}/2} = 1 - \frac{q_{\min}}{2-q_{\min}}$, we obtain

$$
\begin{aligned}
&\mathbb{E} \|\mathbb{1}\bar{x}_{r\tau+1} - \boldsymbol{x}_{r\tau}\|_F^2 \\
&\leq \left(1 - \frac{q_{\min}}{2}\right) \|\mathbb{1}\bar{x}_{(r-1)\tau+1} - \boldsymbol{x}_{(r-1)\tau}\|_F^2 + \frac{4\eta^2 NL^2(2-q_{\min})}{q_{\min}} \|\mathbb{1}\bar{x}_{r\tau+1} - \boldsymbol{x}_{r\tau}\|_F^2 \\
&\quad + \frac{4\eta^2 N^2(2-q_{\min})}{q_{\min}} \|\nabla F(\bar{x}_{r\tau+1})\|^2 + \frac{8\eta^2(\tau-1)^2 L^2(2-q_{\min})}{q_{\min}} \|\mathbb{1}\bar{x}_{r\tau+1} - \boldsymbol{x}_{r\tau}\|^2 \\
&\leq \left(1 - \frac{q_{\min}}{2}\right) \|\mathbb{1}\bar{x}_{(r-1)\tau+1} - \boldsymbol{x}_{(r-1)\tau}\|_F^2 + \frac{8\eta^2 NL^2}{q_{\min}} \|\mathbb{1}\bar{x}_{r\tau+1} - \boldsymbol{x}_{r\tau}\|_F^2 \\
&\quad + \frac{8\eta^2 N^2}{q_{\min}} \|\nabla F(\bar{x}_{r\tau+1})\|^2 + \frac{16\eta^2(\tau-1)^2 L^2}{q_{\min}} \|\mathbb{1}\bar{x}_{r\tau+1} - \boldsymbol{x}_{r\tau}\|^2
\end{aligned} \tag{102}
$$

If $\eta \leq \min\left\{ \frac{q_{\min}^{3/2}}{8L\sqrt{N}}, \frac{q_{\min}^{3/2}}{8L(\tau-1)} \right\}$, we have

$$\left(1 - \frac{q_{\min}^2}{4}\right) \mathbb{E} \|\mathbb{1}\bar{x}_{r\tau+1} - \boldsymbol{x}_{r\tau}\|_F^2 \leq \left(1 - \frac{q_{\min}}{2}\right) \left\| \mathbb{1}\bar{x}_{(r-1)\tau+1} - \boldsymbol{x}_{(r-1)\tau} \right\|_F^2 + \frac{8\eta^2 N^2}{q_{\min}} \|\nabla F(\bar{x}_{r\tau+1})\|^2 \tag{103}$$

Simplifying it further, we obtain

$$
\begin{aligned}
&\mathbb{E} \|\mathbb{1}\bar{x}_{r\tau+1} - \boldsymbol{x}_{r\tau}\|_F^2 \\
&\leq \left(1 - \frac{q_{\min}}{3}\right) \left\| \mathbb{1}\bar{x}_{(r-1)\tau+1} - \boldsymbol{x}_{(r-1)\tau} \right\|_F^2 + \frac{16\eta^2 N^2}{3q_{\min}} \|\nabla F(\bar{x}_{r\tau+1})\|^2
\end{aligned} \tag{104}
$$

$\square$

### E.5 Proof of the Convergence of `FOCUS` ($\mu$-Strong Convexity)

*Proof of Theorem 1 ($\mu$-Strongly Convex Case):*

By Lemmas 3 and 4, we have two recursions

$$
\begin{aligned}
\mathbb{E} \left\| \bar{x}_{(r+1)\tau+1} - x^\star \right\|^2 &\leq \left(1 - \frac{\eta\mu N}{2}\right) \mathbb{E} \|\bar{x}_{r\tau+1} - x^\star\|^2 - 2\eta N(1 - 2\eta NL)\big(F(\bar{x}_{r\tau+1}) - F(x^\star)\big) \\
&\quad + \frac{8\eta L^2 N}{\mu} \mathbb{E} \|\mathbb{1}\bar{x}_{r\tau+1} - \boldsymbol{x}_{r\tau}\|^2
\end{aligned} \tag{105}
$$

$$\mathbb{E} \|\mathbb{1}\bar{x}_{r\tau+1} - \boldsymbol{x}_{r\tau}\|_F^2 \leq \left(1 - \frac{q_{\min}}{3}\right) \left\| \mathbb{1}\bar{x}_{(r-1)\tau+1} - \boldsymbol{x}_{(r-1)\tau} \right\|_F^2 + \frac{16\eta^2 N^2}{3q_{\min}} \|\nabla F(\bar{x}_{r\tau+1})\|^2 \tag{106}$$

$$\leq \left(1 - \frac{q_{\min}}{3}\right) \left\|\mathbb{1}\bar{x}_{(r-1)\tau+1} - \boldsymbol{x}_{(r-1)\tau}\right\|_F^2 + \frac{32\eta^2 N^2 L}{3q_{\min}}\left(F(\bar{x}_{r\tau+1}) - F(x^\star)\right)$$

$$(107)$$

To lighten the notation, we let $A_{r+1} = \mathbb{E}\|\bar{x}_{(r+1)\tau+1} - x^\star\|^2$ and $B_r = \mathbb{E}\|\mathbb{1}\bar{x}_{r\tau+1} - \boldsymbol{x}_{r\tau}\|_F^2$. Therefore, we have

$$A_{r+1} \leq \left(1 - \frac{\eta\mu N}{2}\right) A_r - 2\eta N(1 - 2\eta NL)\left(F(\bar{x}_{r\tau+1}) - F(x^\star)\right) + \frac{8\eta L^2 N}{\mu} B_r \qquad (108)$$

$$B_r \leq \left(1 - \frac{q_{\min}}{3}\right) B_{r-1} + \frac{32\eta^2 N^2 L}{3q_{\min}}\left(F(\bar{x}_{r\tau+1}) - F(x^\star)\right) \qquad (109)$$

After summing up them, the term about $\left(F(\bar{x}_{r\tau+1}) - F(x^\star)\right)$ is negative, so we directly remove it in the upper bound. Then, we get

$$A_{r+1} + \left(1 - \frac{8\eta L^2 N}{\mu}\right) B_r \leq \left(1 - \frac{\eta\mu N}{2}\right) A_r + \left(1 - \frac{q_{\min}}{3}\right) B_{r-1}$$

$$\leq \left(1 - \frac{\eta\mu N}{2}\right)\left(A_r + \left(1 - \frac{8\eta L^2 N}{\mu}\right) B_{r-1}\right), \qquad (110)$$

where (110) holds when $\eta \leq \frac{2q_{\min}}{3N\left(\frac{8L^2}{\mu} + \frac{\mu}{2}\right)}$. $\qquad\square$

Hence, we conclude that `FOCUS` achieves the linear convergence rate of $(1 - \eta\mu N/2)$ under the strongly-convex condition.

## E.6   Proof of the Convergence of `FOCUS` (Non-Convexity with PL Assumption)

*Proof of Theorem 1 (Nonconvex Case with PL Assumption):*

Using the $L-$Lipschitz condition, we have

$$F(\bar{x}_{(r+1)\tau+1})$$

$$\leq F(\bar{x}_{(r+1)\tau}) - \eta\langle\nabla F(\bar{x}_{(r+1)\tau}), \mathbb{1}^\mathsf{T}\nabla\boldsymbol{f}(\boldsymbol{x}_{(r+1)\tau})\rangle + \frac{\eta^2 L}{2}\left\|\mathbb{1}^\mathsf{T}\nabla\boldsymbol{f}(\boldsymbol{x}_{(r+1)\tau})\right\|^2$$

$$= F(\bar{x}_{r\tau+1}) - \eta\langle\nabla F(\bar{x}_{r\tau+1}), \mathbb{1}^\mathsf{T}\nabla\boldsymbol{f}(\boldsymbol{x}_{(r+1)\tau})\rangle + \frac{\eta^2 L}{2}\left\|\mathbb{1}^\mathsf{T}\nabla\boldsymbol{f}(\boldsymbol{x}_{(r+1)\tau})\right\|^2$$

$$\leq F(\bar{x}_{r\tau+1}) - \eta\langle\nabla F(\bar{x}_{r\tau+1}), \mathbb{1}^\mathsf{T}\nabla\boldsymbol{f}(\boldsymbol{x}_{(r+1)\tau})\rangle + \eta^2 L\left\|N\nabla F(\bar{x}_{r\tau+1}) - \mathbb{1}^\mathsf{T}\nabla\boldsymbol{f}(\boldsymbol{x}_{(r+1)\tau})\right\|^2$$

$$\quad + \eta^2 N^2 L\|\nabla F(\bar{x}_{r\tau+1})\|^2, \qquad (111)$$

where the last inequality uses Jensen's inequality.

Using the parallelogram identity to bound the cross term, we have

$$-\eta\langle\nabla F(\bar{x}_{r\tau+1}), \mathbb{1}^\mathsf{T}\nabla\boldsymbol{f}(\boldsymbol{x}_{(r+1)\tau})\rangle$$

$$= -\frac{\eta}{N}\langle N\nabla F(\bar{x}_{r\tau+1}), \mathbb{1}^\mathsf{T}\nabla\boldsymbol{f}(\boldsymbol{x}_{(r+1)\tau})\rangle$$

$$= \frac{\eta}{2N}\left\|\mathbb{1}^\mathsf{T}\nabla\boldsymbol{f}(\mathbb{1}\bar{x}_{r\tau+1}) - \mathbb{1}^\mathsf{T}\nabla\boldsymbol{f}(\boldsymbol{x}_{(r+1)\tau})\right\|^2 - \frac{\eta N}{2}\|\nabla F(\bar{x}_{r\tau+1})\|^2 - \frac{\eta}{2N}\left\|\mathbb{1}^\mathsf{T}\nabla\boldsymbol{f}(\boldsymbol{x}_{(r+1)\tau})\right\|^2$$

$$\leq \frac{\eta}{2N}\left\|\mathbb{1}^\mathsf{T}\nabla\boldsymbol{f}(\mathbb{1}\bar{x}_{r\tau+1}) - \mathbb{1}^\mathsf{T}\nabla\boldsymbol{f}(\boldsymbol{x}_{(r+1)\tau})\right\|^2 - \frac{\eta N}{2}\|\nabla F(\bar{x}_{r\tau+1})\|^2 \qquad (112)$$

where we discard the non-positive term in the last step. Substituting back, we have

$$F(\bar{x}_{(r+1)\tau+1})$$

$$\leq F(\bar{x}_{r\tau+1}) + \left(\frac{\eta}{2N} + \eta^2 L\right)\left\|\mathbb{1}\nabla\boldsymbol{f}(\mathbb{1}\bar{x}_{r\tau+1}) - \mathbb{1}^\mathsf{T}\nabla\boldsymbol{f}(\boldsymbol{x}_{(r+1)\tau})\right\|^2 - \left(\frac{\eta N}{2} - \eta^2 N^2 L\right)\|\nabla F(\bar{x}_{r\tau+1})\|^2$$

$$\leq F(\bar{x}_{r\tau+1}) + \frac{\eta}{N}\left\|\mathbb{1}\nabla\boldsymbol{f}(\mathbb{1}\bar{x}_{r\tau+1}) - \mathbb{1}^\mathsf{T}\nabla\boldsymbol{f}(\boldsymbol{x}_{(r+1)\tau})\right\|^2 - \eta N\|\nabla F(\bar{x}_{r\tau+1})\|^2$$

$$\leq F(\bar{x}_{r\tau+1}) + \frac{\eta}{N}\left\|\mathbb{1}\nabla \boldsymbol{f}(\mathbb{1}\bar{x}_{r\tau+1}) - \mathbb{1}^\top\nabla \boldsymbol{f}(\boldsymbol{x}_{(r+1)\tau})\right\|^2 - \eta\beta N\left(F(\bar{x}_{r\tau+1}) - F^\star\right) - \frac{\eta N}{2}\|\nabla F(\bar{x}_{r\tau+1})\|^2,$$
$$(113)$$

where the second inequality follows $\eta \leq \min\{\frac{1}{4LN}, \frac{1}{2(\tau-1)LN}\}$ and the last inequality we split $-\eta N\|\nabla F(\bar{x}_{r\tau+1})\|^2$ into two parts and applying PL-condition for one part. Next, we minus $F^\star$ on the both sides to obtain:

$$F(\bar{x}_{(r+1)\tau+1}) - F^\star$$
$$\leq (1 - \eta\beta N)\left(F(\bar{x}_{r\tau+1}) - F^\star\right) + \frac{\eta}{N}\left\|\mathbb{1}\nabla \boldsymbol{f}(\mathbb{1}\bar{x}_{r\tau+1}) - \mathbb{1}^\top\nabla \boldsymbol{f}(\boldsymbol{x}_{(r+1)\tau})\right\|^2 - \frac{\eta N}{2}\|\nabla F(\bar{x}_{r\tau+1})\|^2$$
$$(114)$$

Recalling the previous result (95):

$$\mathbb{E}\left\|\mathbb{1}^\top\nabla \boldsymbol{f}(\mathbb{1}\bar{x}_{r\tau+1}) - \mathbb{1}^\top\nabla \boldsymbol{f}(\boldsymbol{x}_{(r+1)\tau})\right\|^2 \leq \left(2L^2 N + 8\eta^2(\tau-1)^2 L^4 N\right)\|\mathbb{1}\bar{x}_{r\tau+1} - \boldsymbol{x}_{r\tau}\|^2$$
$$(115)$$

Putting (115) back to (114), we have

$$F(\bar{x}_{(r+1)\tau+1}) - F^\star \leq (1 - \eta\beta N)\left(F(\bar{x}_{r\tau+1}) - F^\star\right) + 4\eta L^2\|\mathbb{1}\bar{x}_{r\tau+1} - \boldsymbol{x}_{r\tau}\|^2 - \frac{\eta N}{2}\|\nabla F(\bar{x}_{r\tau+1})\|^2$$
$$(116)$$

Recalling the Lemma 4, we know

$$\mathbb{E}\|\mathbb{1}\bar{x}_{r\tau+1} - \boldsymbol{x}_{r\tau}\|_F^2 \leq \left(1 - \frac{q_{\min}}{3}\right)\left\|\mathbb{1}\bar{x}_{(r-1)\tau+1} - \boldsymbol{x}_{(r-1)\tau}\right\|_F^2 + \frac{16\eta^2 N^2}{3q_{\min}}\|\nabla F(\bar{x}_{r\tau+1})\|^2$$
$$(117)$$

With a condition $\eta \leq \min\{\frac{3q_{\min}}{32N}, \frac{q_{\min}}{3\beta N(4-q_{\min})}, \frac{q_{\min}}{16L^2}\}$, we denote $A_r = F(\bar{x}_{r\tau+1}) - F^\star$ and $B_r = \mathbb{E}\|\mathbb{1}\bar{x}_{r\tau+1} - \boldsymbol{x}_{r\tau}\|^2$, and then we have

$$A_{r+1} + \left(1 - 4\eta L^2\right) B_r \leq (1 - \eta\beta N)A_r + \left(1 - \frac{q_{\min}}{3}\right) B_{r-1}$$
$$= (1 - \eta\beta N)\left(A_r + \frac{1 - \frac{q_{\min}}{3}}{1 - \eta\beta N} B_{r-1}\right)$$
$$\leq (1 - \eta\beta N)\left(A_r + \left(1 - \frac{q_{\min}}{4}\right) B_{r-1}\right)$$
$$\leq (1 - \eta\beta N)\left(A_r + \left(1 - 4\eta L^2\right) B_{r-1}\right)$$
$$(118)$$

Therefore, we conclude that `FOCUS` achieves the linear convergence rate of $(1 - \eta\beta N)$ in the nonconvex case with PL condition. □

### E.7  Proof of the Convergence of `FOCUS` (General Non-Convexity)

*Proof of Theorem 1 (General Nonconvex Case):*

We begin with $L-$Lipschitz condition:

$$F(\bar{x}_{(r+1)\tau+1}) \leq F(\bar{x}_{r\tau+1}) - \eta\left\langle\nabla F(\bar{x}_{r\tau+1}), \mathbb{1}^\top\nabla \boldsymbol{f}(\boldsymbol{x}_{(r+1)\tau})\right\rangle + \frac{\eta^2 L}{2}\left\|\mathbb{1}^\top\nabla \boldsymbol{f}(\boldsymbol{x}_{(r+1)\tau})\right\|^2$$
$$\leq F(\bar{x}_{r\tau+1}) - \eta\left\langle\nabla F(\bar{x}_{r\tau+1}), \mathbb{1}^\top\nabla \boldsymbol{f}(\boldsymbol{x}_{(r+1)\tau})\right\rangle$$
$$+ \eta^2 L\left\|N\nabla F(\bar{x}_{r\tau+1}) - \mathbb{1}^\top\nabla \boldsymbol{f}(\boldsymbol{x}_{(r+1)\tau})\right\|^2 + \eta^2 N^2 L\|\nabla F(\bar{x}_{r\tau+1})\|^2, \quad (119)$$

where the last inequality utilizes the Jensen's inequality.

Then, we deal with the cross term:

$$-\eta\left\langle\nabla F(\bar{x}_{r\tau+1}), \mathbb{1}^\top\nabla \boldsymbol{f}(\boldsymbol{x}_{(r+1)\tau})\right\rangle$$
$$= -\eta N\|\nabla F(\bar{x}_{r\tau+1})\|^2 + \eta\left\langle\nabla F(\bar{x}_{r\tau+1}), N\nabla F(\bar{x}_{r\tau+1}) - \mathbb{1}^\top\nabla \boldsymbol{f}(\boldsymbol{x}_{(r+1)\tau})\right\rangle$$

$$\leq -\frac{\eta N}{2}\|\nabla F(\bar{x}_{r\tau+1})\|^2 + \frac{\eta}{2N}\left\|\mathbb{1}\nabla\boldsymbol{f}(\mathbb{1}\bar{x}_{r\tau+1}) - \mathbb{1}^\top\nabla\boldsymbol{f}(\boldsymbol{x}_{(r+1)\tau})\right\|^2 \tag{120}$$

Substituting (120) back to (119), we have

$$\frac{\eta N}{2}(1 - 2\eta NL)\mathbb{E}\|\nabla F(\bar{x}_{r\tau+1})\|^2$$

$$\leq F(\bar{x}_{r\tau+1}) - \mathbb{E}\,F(\bar{x}_{(r+1)\tau+1}) + \left(\frac{\eta}{2N} + \eta^2 L\right)\mathbb{E}\left\|\mathbb{1}^\top\nabla\boldsymbol{f}(\bar{x}_{r\tau+1}) - \mathbb{1}^\top\nabla\boldsymbol{f}(\boldsymbol{x}_{(r+1)\tau})\right\|^2 \tag{121}$$

To bound the term $\left\|\mathbb{1}^\top\nabla\boldsymbol{f}(\bar{x}_{r\tau+1}) - \mathbb{1}^\top\nabla\boldsymbol{f}(\boldsymbol{x}_{(r+1)\tau})\right\|^2$, we can directly use the result (95), so we will obtain:

$$\frac{\eta N}{2}(1 - 2\eta NL)\mathbb{E}\|\nabla F(\bar{x}_{r\tau+1})\|^2$$

$$\leq F(\bar{x}_{r\tau+1}) - \mathbb{E}\,F(\bar{x}_{(r+1)\tau+1}) + \left(\frac{\eta}{2N} + \eta^2 L\right)\left(2L^2 N + 8\eta^2(\tau-1)^2 L^4 N\right)\mathbb{E}\|\mathbb{1}\bar{x}_{r\tau+1} - \boldsymbol{x}_{r\tau}\|^2 \tag{122}$$

Then, when $\eta \leq \min\{\frac{1}{4LN}, \frac{1}{2L(\tau-1)}\}$, we can get $1/2 \leq (1 - 2\eta NL)$, $\eta^2 L \leq \eta/2N$ and $8\eta^2(\tau-1)^2 L^4 N \leq 2L^2 N$. Thus, we can simplify the coefficients further as follows:

$$\mathbb{E}\|\nabla F(\bar{x}_{r\tau+1})\|^2 \leq \frac{4}{\eta N}\left(F(\bar{x}_{r\tau+1}) - F(\bar{x}_{(r+1)\tau+1})\right) + \frac{16L^2}{N}\mathbb{E}\|\mathbb{1}\bar{x}_{r\tau+1} - \boldsymbol{x}_{r\tau}\|^2 \tag{123}$$

Recalling the Lemma 4, we know

$$\mathbb{E}\|\mathbb{1}\bar{x}_{r\tau+1} - \boldsymbol{x}_{r\tau}\|_F^2 \leq \left(1 - \frac{q_{\min}}{3}\right)\left\|\mathbb{1}\bar{x}_{(r-1)\tau+1} - \boldsymbol{x}_{(r-1)\tau}\right\|_F^2 + \frac{16\eta^2 N^2}{3q_{\min}}\|\nabla F(\bar{x}_{r\tau+1})\|^2 \tag{124}$$

We denote that $A_r = \mathbb{E}\|\nabla F(\bar{x}_{r\tau+1})\|^2$, $B_r = F(\bar{x}_{r\tau+1})$ and $C_r = \mathbb{E}\|\mathbb{1}\bar{x}_{r\tau+1} - \boldsymbol{x}_{r\tau}\|^2$.

Through (124) and (123), we have the following two recursions:

$$A_r \leq \frac{4}{\eta N}\left(B_r - B_{r+1}\right) + \frac{16L^2}{N}C_r \tag{125}$$

$$C_r \leq \left(1 - \frac{q_{\min}}{3}\right)C_{r-1} + \frac{16\eta^2 N^2}{3q_{\min}}A_r \tag{126}$$

Taking the summation from $r = 0$ to $R - 1$, we have

$$\frac{1}{R}\sum_{r=0}^{R-1} C_r = \frac{1}{R}\sum_{r=1}^{R-1} C_r \leq \left(1 - \frac{q_{\min}}{3}\right)\sum_{r=1}^{R-1} C_{r-1} + \frac{16\eta^2 N^2}{3Rq_{\min}}\frac{1}{R}\sum_{r=1}^{R-1} A_r$$

$$\leq \left(1 - \frac{q_{\min}}{3}\right)\frac{1}{R}\sum_{r=0}^{R-1} C_r + \frac{16\eta^2 N^2}{3Rq_{\min}}\sum_{r=0}^{R-1} A_r \tag{127}$$

Note $C_0 = 0$ and we shift the subscripts in the second inequation since adding a non-negative term always holds. Therefore,

$$\frac{1}{R}\sum_{r=0}^{R-1} C_r \leq \frac{16\eta^2 N^2}{Rq_{\min}^2}\sum_{r=0}^{R-1} A_r \tag{128}$$

Lastly, noting

$$\frac{1}{R}\sum_{r=0}^{R-1} A_r \leq \frac{4}{\eta N}(B_0 - B_R) + \frac{16L^2}{NR}\sum_{r=0}^{R-1} C_r$$

$$\implies \left(1 - \frac{256\eta^2 L^2 N}{q_{\min}^2}\right)\frac{1}{R}\sum_{r=0}^{R-1} A_r \leq \frac{4}{\eta N}(B_0 - B_R)$$

$$\implies \frac{1}{R}\sum_{r=0}^{R-1}\mathbb{E}\|\nabla F(\bar{x}_{r\tau+1})\|^2 \leq \frac{8\left(F(\bar{x}_1) - F(\bar{x}^\star)\right)}{\eta NR}, \tag{129}$$

where the last inequality follows $\eta \leq \frac{q_{\min}}{16L\sqrt{2N}}$. $\qquad\square$

# F Extensions to Stochastic Gradient Case (`SG-FOCUS`)

When applying the optimization algorithm to the machine learning problem, we need to use stochastic gradients instead of gradient oracle. Only one line change of `FOCUS` is required to support the stochastic gradient case, which is highlighted in Algorithm 2. Notably, **the past stochastic gradient is $\nabla f_i(x_{k-1,i}; \xi_{k-1})$ instead of $\nabla f_i(x_{k-1,i}; \xi_k)$. This choice preserves the crucial tracking property**. Furthermore, this approach offers a computational advantage by allowing us to store and reuse the prior stochastic gradient, thus avoiding redundant computations.

---

**Algorithm 2** `SG-FOCUS` (Stochastic Gradient Version of `FOCUS`)

---

1: **Initialize**: Choose learning rate $\eta$ and local update $\tau$. At server, set a random $x_0$, $y_0 = 0$ and set $\nabla f_i(x_{-1,i}^{(0)}; \xi_{-1}) = y_{0,i}^{(r)}$ at all clients.
2: **for** $r = 0, 1, ..., R-1$ **do**
3:     Arbitrarily sample client index set $S_r$
4:     **for** $i$ in $S_r$ **parallel do**
5:         $x_{0,i}^{(r)} \Leftarrow x_r, \ y_{0,i}^{(r)} \Leftarrow 0$            ▷ Pull model $x_r$ from server while $y_r$ is NOT pulled
6:         **for** $t = 0, \cdots, \tau - 1$ **do**
7:             $y_{t+1,i}^{(r)} = y_{t,i}^{(r)} + \nabla f_i(x_{t,i}^{(r)}; \xi_t) - \nabla f_i(x_{t-1,i}^{(r)}; \xi_{t-1})$      ▷ Current and last stochastic grad.
8:             $x_{t+1,i} = x_{t,i} - \eta y_{t+1,i}$
9:         **end for**
10:     **end for**
11:     $y_{r+1} = y_r + \sum_{i \in S_r} y_{\tau,i}^{(r)}$                ▷ Pushing $y_{i,\tau}^{(r)}$ to server (Not averaging)
12:     $x_{r+1} = x_r - \eta y_{r+1}$                    ▷ Client model $x_{\tau,i}^{(r)}$ is NEVER pushed.
13: **end for**

---

Before showing the lemmas and proof of `SG-FOCUS`, we introduce an assumption of unbiased stochastic gradients with bounded variance as follows.

**Assumption 6** (Unbiased Stochastic Gradients with Bounded Variance). *The stochastic gradient computed by clients or the server is unbiased with bounded variance:*

$$\mathbb{E}\left[\nabla f_i(x; \xi)\right] = \nabla f_i(x) \quad and \quad \mathbb{E}\left\|\nabla f_i(x; \xi) - \nabla f_i(x)\right\|^2 \le \sigma^2, \tag{130}$$

*where $\xi$ is the data sample.*

## F.1 Descent Lemma for `SG-FOCUS`

**Lemma 5.** *Under assumption 1, 2, 3 and 6, if $\eta \le \min\{\frac{\mu}{128(\tau-1)^2 L^2 N}, \frac{1}{2\sqrt{2}(\tau-1)L}\}$, the expectation of server's error can be bounded as*

$$\mathbb{E}\left\|\bar{x}_{(r+1)\tau+1} - x^\star\right\|^2 \le \left(1 - \frac{\eta\mu N}{2}\right)\mathbb{E}\left\|\bar{x}_{r\tau+1} - x^\star\right\|^2 - 2\eta N(1 - 2\eta NL)\left(F(\bar{x}_{r\tau+1}) - F(x^\star)\right)$$

$$+ \frac{16\eta L^2 N}{\mu}\left\|\mathbb{1}\bar{x}_{r\tau+1} - \boldsymbol{x}_{r\tau}\right\|^2 + 2\eta^2\sigma^2,$$

*where $x^\star$ is the optimal point. Note that this lemma can only be used for strongly convex cases.*

*Proof of Lemma 5:*

The server's error recursion from the $(r+1)$-th round to the $r$-th round is

$$\mathbb{E}\left\|\bar{x}_{(r+1)\tau+1} - x^\star\right\|^2$$
$$= \mathbb{E}\left\|\bar{x}_{r\tau+1} - x^\star - \eta\mathbb{1}^\mathsf{T}\nabla\boldsymbol{f}(\boldsymbol{x}_{(r+1)\tau}; \xi_{(r+1)\tau})\right\|^2$$
$$= \|\bar{x}_{r\tau+1} - x^\star\|^2 - 2\eta\mathbb{E}\left\langle\bar{x}_{r\tau+1} - x^\star, \mathbb{1}^\mathsf{T}\nabla\boldsymbol{f}(\boldsymbol{x}_{(r+1)\tau}; \xi_{(r+1)\tau})\right\rangle + \eta^2\mathbb{E}\left\|\mathbb{1}^\mathsf{T}\nabla\boldsymbol{f}(\boldsymbol{x}_{(r+1)\tau}; \xi_{(r+1)\tau})\right\|^2$$
$$\le \|\bar{x}_{r\tau+1} - x^\star\|^2 - 2\eta\left\langle\bar{x}_{r\tau+1} - x^\star, \mathbb{1}^\mathsf{T}\nabla\boldsymbol{f}(\boldsymbol{x}_{(r+1)\tau})\right\rangle + \eta^2\sigma^2$$
$$+ 2\eta^2\left\|\mathbb{1}^\mathsf{T}\nabla\boldsymbol{f}(\boldsymbol{x}_{(r+1)\tau}) - \mathbb{1}^\mathsf{T}\nabla\boldsymbol{f}(\mathbb{1}\bar{x}_{r\tau+1})\right\|^2 + 2\eta^2 N^2\|\nabla F(\bar{x}_{r\tau+1})\|^2$$

$$\leq \|\bar{x}_{r\tau+1} - x^\star\|^2 - 2\eta\langle\bar{x}_{r\tau+1} - x^\star, \mathbb{1}^\mathsf{T}\nabla\boldsymbol{f}(\boldsymbol{x}_{(r+1)\tau})\rangle + \eta^2\sigma^2$$

$$+ 2\eta^2\left\|\mathbb{1}^\mathsf{T}\nabla\boldsymbol{f}(\boldsymbol{x}_{(r+1)\tau}) - \mathbb{1}^\mathsf{T}\nabla\boldsymbol{f}(\mathbb{1}\bar{x}_{r\tau+1})\right\|^2 + 4\eta^2 N^2 L\big(F(\bar{x}_{r\tau+1}) - F(x^\star)\big), \qquad (131)$$

where the first inequality is obtained by Jensen's inequality and the second inequality utilizes the Lipschitz condition $\frac{1}{2L}\|\nabla F(x)\|^2 \leq F(x) - F(x^\star)$. The cross term can be bounded as

$$-2\eta\langle\bar{x}_{r\tau+1} - x^\star, \mathbb{1}^\mathsf{T}\nabla\boldsymbol{f}(\boldsymbol{x}_{(r+1)\tau})\rangle$$

$$= -2\eta\langle\bar{x}_{r\tau+1} - x^\star, N\nabla F(\bar{x}_{r\tau+1})\rangle - 2\eta\langle\bar{x}_{r\tau+1} - x^\star, \mathbb{1}^\mathsf{T}\nabla\boldsymbol{f}(\boldsymbol{x}_{(r+1)\tau}) - \mathbb{1}^\mathsf{T}\nabla\boldsymbol{f}(\mathbb{1}\bar{x}_{r\tau+1})\rangle$$

$$\leq -2\eta N(F(\bar{x}_{r\tau+1}) - F(x^\star) + \frac{\mu}{2}\|\bar{x}_{r\tau+1} - x^\star\|^2) + \eta\epsilon\|\bar{x}_{r\tau+1} - x^\star\|^2$$

$$+ \frac{\eta}{\epsilon}\left\|\mathbb{1}^\mathsf{T}\nabla\boldsymbol{f}(\boldsymbol{x}_{(r+1)\tau}) - \mathbb{1}^\mathsf{T}\nabla\boldsymbol{f}(\mathbb{1}\bar{x}_{r\tau+1})\right\|^2$$

$$\leq -2\eta N(F(\bar{x}_{r\tau+1}) - F(x^\star)) - \frac{\eta\mu N}{2}\|\bar{x}_{r\tau+1} - x^\star\|^2 + \frac{2\eta}{\mu}\left\|\mathbb{1}^\mathsf{T}\nabla\boldsymbol{f}(\boldsymbol{x}_{(r+1)\tau}) - \mathbb{1}^\mathsf{T}\nabla\boldsymbol{f}(\mathbb{1}\bar{x}_{r\tau+1})\right\|^2,$$
$$(132)$$

where the first inequality utilizes Young's inequality with $\epsilon$, and we set $\epsilon = \mu/2$ in the second inequality.

Plugging (132) into (131), we have

$$\mathbb{E}\left\|\bar{x}_{(r+1)\tau+1} - x^\star\right\|^2 \leq \left(1 - \frac{\eta\mu N}{2}\right)\|\bar{x}_{r\tau+1} - x^\star\|^2 - 2\eta N(1 - 2\eta NL)\big(F(\bar{x}_{r\tau+1}) - F(x^\star)\big)$$

$$+ 2\eta\left(\frac{1}{\mu} + \eta\right)\left\|\mathbb{1}^\mathsf{T}\nabla\boldsymbol{f}(\boldsymbol{x}_{(r+1)\tau}) - \mathbb{1}^\mathsf{T}\nabla\boldsymbol{f}(\mathbb{1}\bar{x}_{r\tau+1})\right\|^2 + \eta^2\sigma^2$$
$$(133)$$

Now, we focus on this gradient difference term:

$$\mathbb{E}\left\|\mathbb{1}^\mathsf{T}\nabla\boldsymbol{f}(\mathbb{1}\bar{x}_{r\tau+1}) - \mathbb{1}^\mathsf{T}\nabla\boldsymbol{f}(\boldsymbol{x}_{(r+1)\tau})\right\|^2$$

$$\leq L^2 N\mathbb{E}\left\|\mathbb{1}\bar{x}_{r\tau+1} - \boldsymbol{x}_{(r+1)\tau}\right\|_F^2$$

$$= L^2 N\mathbb{E}\left\|\mathbb{1}\bar{x}_{r\tau+1} - \boldsymbol{x}_{r\tau+1} + \eta\sum_{k=r\tau+2}^{(r+1)\tau} D_k\boldsymbol{y}_k\right\|_F^2$$

$$\leq 2L^2 N\|\mathbb{1}\bar{x}_{r\tau+1} - \boldsymbol{x}_{r\tau+1}\|^2 + 2L^2 N\mathbb{E}\left\|\eta\sum_{k=r\tau+2}^{(r+1)\tau} D_k\boldsymbol{y}_k\right\|_F^2$$

$$\leq 2L^2 N\|\mathbb{1}\bar{x}_{r\tau+1} - \boldsymbol{x}_{r\tau+1}\|^2 + 2(\tau-1)\eta^2 L^2 N\mathbb{E}\sum_{k=r\tau+2}^{(r+1)\tau}\|D_k\boldsymbol{y}_k\|_F^2, \qquad (134)$$

where the last two inequalities use the Jensen's inequality $\|\sum_{i=1}^N a_i\|^2 \leq N\sum_{i=1}^N \|a_i\|^2$.

Next, we need to bound $y_{k,i}$ by using $L-$Lipshitz assumption and Jensen's inequality. Assume $i$ is the index among the sampled clients:

$$\mathbb{E}\|y_{k,i}\|^2 = \mathbb{E}\|\nabla f_i(x_{k-1,i}; \xi_{k-1,i}) - \nabla f_i(x_{r\tau,i}; \xi_{r\tau,i})\|^2$$

$$\leq 4\mathbb{E}\|\nabla f_i(x_{k-1,i}; \xi_{k-1,i}) - \nabla f_i(x_{k-1,i})\|^2 + 4\mathbb{E}\|\nabla f_i(x_{k-1,i}) - \nabla f_i(x_{r\tau+1,i})\|^2$$

$$+ 4\mathbb{E}\|\nabla f_i(x_{r\tau+1,i}) - \nabla f_i(x_{r\tau,i})\|^2 + 4\mathbb{E}\|\nabla f_i(x_{r\tau,i}) - \nabla f_i(x_{r\tau,i}; \xi_{r\tau,i})\|^2$$

$$\leq 4L^2\|x_{k-1,i} - x_{r\tau+1,i}\|^2 + 4L^2\|x_{r\tau+1,i} - x_{r\tau,i}\|^2 + 8\sigma^2$$

$$\leq 4\eta^2 L^2(k-r\tau-2)\sum_{k'=r\tau+2}^{k}\|y_{k',i}\|^2 + 4L^2\|x_{r\tau+1,i} - x_{r\tau,i}\|^2 + 8\sigma^2$$

$$\leq 4\eta^2 L^2(\tau-1)\sum_{k'=r\tau+2}^{(r+1)\tau}\|y_{k',i}\|^2 + 4L^2\|x_{r\tau+1,i} - x_{r\tau,i}\|^2 + 8\sigma^2,$$

where, in the last inequality, we just expand the non-negative term to the maximum difference cases. Hence, taking another summation of $k$ from $r\tau + 2$ to $(r+1)\tau$, we obtain

$$\left(1 - 4\eta^2 L^2(\tau - 1)\right) \sum_{k=r\tau+2}^{(r+1)\tau} \mathbb{E}\,\|y_{k,i}\|^2 \leq 4(\tau - 1)L^2\|x_{r\tau+1,i} - x_{r\tau,i}\|^2 + 8(\tau - 1)\sigma^2$$

When $1/2 \leq \left(1 - 4\eta^2 L^2(\tau - 1)\right)$ i.e., $\eta \leq \frac{1}{2L\sqrt{2(\tau-1)}}$, we conclude

$$\mathbb{E} \sum_{k=r\tau+2}^{(r+1)\tau} \|y_{k,i}\|^2 \leq 8(\tau - 1)L^2\|x_{r\tau+1,i} - x_{r\tau,i}\|^2 + 16(\tau - 1)\sigma^2$$

$$= 8(\tau - 1)L^2\|\bar{x}_{r\tau+1} - x_{r\tau,i}\|^2 + 16(\tau - 1)\sigma^2 \tag{135}$$

Plugging (135) back to (134), we obtain

$$\mathbb{E}\,\left\|\mathbb{1}^\top \nabla f(\mathbb{1}\bar{x}_{r\tau+1}) - \mathbb{1}^\top \nabla f(\boldsymbol{x}_{(r+1)\tau})\right\|^2$$
$$\leq \left(2L^2 N + 16\eta^2(\tau - 1)^2 L^4 N\right)\|\mathbb{1}\bar{x}_{r\tau+1} - \boldsymbol{x}_{r\tau}\|^2 + 32\eta^2(\tau - 1)^2 L^2 N\sigma^2, \tag{136}$$

Plugging (136) into (133), we obtain

$$\mathbb{E}\,\left\|\bar{x}_{(r+1)\tau+1} - x^\star\right\|^2 \leq \left(1 - \frac{\eta\mu N}{2}\right)\mathbb{E}\,\|\bar{x}_{r\tau+1} - x^\star\|^2 - 2\eta N(1 - 2\eta N L)\left(F(\bar{x}_{r\tau+1}) - F(x^\star)\right)$$

$$+ 2\eta\left(\frac{1}{\mu} + \eta\right)\left(2L^2 N + 16\eta^2(\tau - 1)^2 L^4 N\right)\|\mathbb{1}\bar{x}_{r\tau+1} - \boldsymbol{x}_{r\tau}\|^2$$

$$+ \eta^2\sigma^2 + 2\eta\left(\frac{1}{\mu} + \eta\right)32\eta^2(\tau - 1)^2 L^2 N\sigma^2$$

$$\leq \left(1 - \frac{\eta\mu N}{2}\right)\mathbb{E}\,\|\bar{x}_{r\tau+1} - x^\star\|^2 - 2\eta N(1 - 2\eta N L)\left(F(\bar{x}_{r\tau+1}) - F(x^\star)\right)$$

$$+ \frac{16\eta L^2 N}{\mu}\|\mathbb{1}\bar{x}_{r\tau+1} - \boldsymbol{x}_{r\tau}\|^2 + 2\eta^2\sigma^2,$$

where the last inequality follows:

$$\eta \leq \min\{\frac{1}{\mu}, \frac{\mu}{128(\tau - 1)^2 L^2 N}, \frac{1}{2\sqrt{2}(\tau - 1)L}\} = \min\{\frac{\mu}{128(\tau - 1)^2 L^2 N}, \frac{1}{2\sqrt{2}(\tau - 1)L}\}.$$

$\square$

### F.2   Consensus Lemma for `SG-FOCUS`

**Lemma 6** (Consensus Lemma for `SG-FOCUS`). *Under assumptions 1, 2, 6, if the learning rate* $\eta \leq \min\left\{\frac{q_{\min}^{3/2}}{4L\sqrt{6N}}, \frac{q_{\min}^{3/2}}{8\sqrt{2}L(\tau-1)}\right\}$, *the difference between the server's global model and the client's local model can be bounded as*

$$\mathbb{E}\,\|\mathbb{1}\bar{x}_{r\tau+1} - \boldsymbol{x}_{r\tau}\|_F^2 \leq \left(1 - \frac{q_{\min}}{3}\right)\left\|\mathbb{1}\bar{x}_{(r-1)\tau+1} - \boldsymbol{x}_{(r-1)\tau}\right\|_F^2 + \frac{8\eta^2 N^2}{q_{\min}}\|\nabla F(\bar{x}_{r\tau+1})\|^2 + \frac{8\eta^2 N}{q_{\min}}\sigma^2$$

*Note that this consensus lemma is not related to any convex or strongly convex properties, so it can also be applied in nonconvex cases.*

*Proof of Lemma 6:*

Here, we focus on how different the server's model at the $r$-round and the client's model just before pulling the model from the server. Taking the conditional expectation, we have

$$\mathbb{E}\,\|\mathbb{1}\bar{x}_{r\tau+1} - \boldsymbol{x}_{r\tau}\|_F^2$$

$$= \mathbb{E}\,\left\|\mathbb{1}\bar{x}_{(r-1)\tau+1} - \eta\mathbb{1}\bar{y}_{r\tau+1} - \boldsymbol{x}_{(r-1)\tau+1} + \eta\sum_{k=(r-1)\tau+2}^{r\tau} D_k \boldsymbol{y}_k\right\|_F^2$$

$$\leq \frac{1}{\rho} \mathbb{E} \left\| \mathbb{1}\bar{x}_{(r-1)\tau+1} - \boldsymbol{x}_{(r-1)\tau+1} \right\|_F^2 + \frac{2\eta^2}{1-\rho} \mathbb{E} \left\| \mathbb{1}\bar{y}_{r\tau+1} \right\|^2 + \frac{2\eta^2}{1-\rho} \mathbb{E} \left\| \sum_{k=(r-1)\tau+2}^{r\tau} D_k \boldsymbol{y}_k \right\|^2, \tag{137}$$

where the inequality above uses Jensen's inequality $\|a+b\|^2 \leq \frac{1}{\rho}\|a\|^2 + \frac{1}{1-\rho}\|b\|^2$ and $\rho$ here can be any value between 0 and 1 (exclusive). For each term, we know

$$\mathbb{E} \left\| \mathbb{1}\bar{x}_{(r-1)\tau+1} - \boldsymbol{x}_{(r-1)\tau+1} \right\|_F^2 = \sum_{i=1}^{N} (1-q_i) \| \bar{x}_{(r-1)\tau+1} - x_{(r-1)\tau,i} \|^2$$

$$\leq (1-q_{\min}) \left\| \mathbb{1}\bar{x}_{(r-1)\tau+1} - \boldsymbol{x}_{(r-1)\tau} \right\|_F^2 \tag{138}$$

and

$$\|\bar{y}_{r\tau+1}\|^2 \leq 3 \left\| \mathbb{1}^\mathsf{T} \nabla \boldsymbol{f}(\boldsymbol{x}_{r\tau}; \xi_{r\tau}) - \mathbb{1}^\mathsf{T} \nabla \boldsymbol{f}(\mathbb{1}\bar{x}_{r\tau}) \right\|^2$$

$$+ 3 \left\| \mathbb{1}^\mathsf{T} \nabla \boldsymbol{f}(\mathbb{1}\bar{x}_{r\tau}) - \mathbb{1}^\mathsf{T} \nabla \boldsymbol{f}(\mathbb{1}\bar{x}_{r\tau+1}) \right\|^2 + 3 \left\| N \nabla F(\bar{x}_{r\tau+1}) \right\|^2$$

$$\leq 3N\sigma^2 + 3NL^2 \|\mathbb{1}\bar{x}_{r\tau+1} - \boldsymbol{x}_{r\tau}\|_F^2 + 3 \left\| N \nabla F(\bar{x}_{r\tau+1}) \right\|^2 \tag{139}$$

and

$$\mathbb{E} \left\| \sum_{k=(r-1)\tau+2}^{r\tau} D_k \boldsymbol{y}_k \right\|^2 \leq (\tau-1) \mathbb{E} \sum_{k=r\tau+2}^{(r+1)\tau} \|D_k \boldsymbol{y}_k\|^2$$

$$\leq 4(\tau-1)^2 L^2 \sum_{i=1}^{N} q_i \|\bar{x}_{r\tau+1} - x_{r\tau,i}\|^2$$

$$\leq 4(\tau-1)^2 L^2 \|\mathbb{1}\bar{x}_{r\tau+1} - \boldsymbol{x}_{r\tau}\|^2 \tag{140}$$

Putting (138), (139), (140) back to (137) and selecting $\rho = \frac{1-q_{\min}}{1-q_{\min}/2} = 1 - \frac{q_{\min}}{2-q_{\min}}$, we obtain

$$\mathbb{E} \left\| \mathbb{1}\bar{x}_{r\tau+1} - \boldsymbol{x}_{r\tau} \right\|_F^2$$

$$\leq \left(1 - \frac{q_{\min}}{2}\right) \|\mathbb{1}\bar{x}_{(r-1)\tau+1} - \boldsymbol{x}_{(r-1)\tau}\|_F^2 + \frac{6\eta^2 NL^2(2-q_{\min})}{q_{\min}} \|\mathbb{1}\bar{x}_{r\tau+1} - \boldsymbol{x}_{r\tau}\|_F^2$$

$$+ \frac{6\eta^2 N^2(2-q_{\min})}{q_{\min}} \|\nabla F(\bar{x}_{r\tau+1})\|^2 + \frac{6\eta^2 N(2-q_{\min})}{q_{\min}} \sigma^2$$

$$+ \frac{8\eta^2 (\tau-1)^2 L^2 (2-q_{\min})}{q_{\min}} \|\mathbb{1}\bar{x}_{r\tau+1} - \boldsymbol{x}_{r\tau}\|^2$$

$$\leq \left(1 - \frac{q_{\min}}{2}\right) \|\mathbb{1}\bar{x}_{(r-1)\tau+1} - \boldsymbol{x}_{(r-1)\tau}\|_F^2 + \frac{12\eta^2 NL^2}{q_{\min}} \|\mathbb{1}\bar{x}_{r\tau+1} - \boldsymbol{x}_{r\tau}\|_F^2$$

$$+ \frac{12\eta^2 N^2}{q_{\min}} \|\nabla F(\bar{x}_{r\tau+1})\|^2 + \frac{12\eta^2 NL^2}{q_{\min}} \sigma^2 + \frac{16\eta^2 (\tau-1)^2 L^2}{q_{\min}} \|\mathbb{1}\bar{x}_{r\tau+1} - \boldsymbol{x}_{r\tau}\|^2$$

If $\eta \leq \min \left\{ \frac{q_{\min}^{3/2}}{4L\sqrt{6N}}, \frac{q_{\min}^{3/2}}{8\sqrt{2}L(\tau-1)} \right\}$, we have

$$\left(1 - \frac{q_{\min}^2}{4}\right) \mathbb{E} \left\| \mathbb{1}\bar{x}_{r\tau+1} - \boldsymbol{x}_{r\tau} \right\|_F^2$$

$$\leq \left(1 - \frac{q_{\min}}{2}\right) \left\| \mathbb{1}\bar{x}_{(r-1)\tau+1} - \boldsymbol{x}_{(r-1)\tau} \right\|_F^2 + \frac{12\eta^2 N^2}{q_{\min}} \|\nabla F(\bar{x}_{r\tau+1})\|^2 + \frac{12\eta^2 N}{q_{\min}} \sigma^2$$

Simplifying it further, we obtain

$$\mathbb{E} \left\| \mathbb{1}\bar{x}_{r\tau+1} - \boldsymbol{x}_{r\tau} \right\|_F^2 \leq \left(1 - \frac{q_{\min}}{3}\right) \left\| \mathbb{1}\bar{x}_{(r-1)\tau+1} - \boldsymbol{x}_{(r-1)\tau} \right\|_F^2 + \frac{8\eta^2 N^2}{q_{\min}} \|\nabla F(\bar{x}_{r\tau+1})\|^2 + \frac{8\eta^2 N}{q_{\min}} \sigma^2$$

$$\square$$

### F.3 Proof of the Convergence of `SG-FOCUS` ($\mu$-Strong Convexity)

**Theorem 4** (Convergence of `SG-FOCUS` for Strongly Convex Functions). *Under assumption 1, 2, 3 and 6, if the learning rate $\eta \leq \min\left\{\frac{2q_{\min}}{3N\left(\frac{16L^2}{\mu}+\frac{\mu}{2}\right)}, \frac{q_{\min}^{3/2}}{4L\sqrt{6N}}, \frac{q_{\min}^{3/2}}{8\sqrt{2}L(\tau-1)}, \frac{\mu}{128(\tau-1)^2L^2N}\right\}$,*

$$\Gamma_R \leq \left(1 - \frac{\eta\mu N}{2}\right)^R \Gamma_0 + \frac{4(q_{\min}+N^2)}{\mu N q_{\min}}\eta\sigma^2,$$

*where $\Gamma_r := \mathbb{E}\left\|\bar{x}_{(r+1)\tau+1} - x^\star\right\|^2 + \left(1 - 8\eta L^2 N/\mu\right)\mathbb{E}\left\|\mathbb{1}\bar{x}_{(r-1)\tau+1} - \boldsymbol{x}_{(r-1)\tau}\right\|_F^2$.*

*Proof of Theorem 4:*

By Lemmas 5 and 6, we have two recursions – the average iterate one:

$$\mathbb{E}\left\|\bar{x}_{(r+1)\tau+1} - x^\star\right\|^2 \leq \left(1 - \frac{\eta\mu N}{2}\right)\mathbb{E}\left\|\bar{x}_{r\tau+1} - x^\star\right\|^2 - 2\eta N(1 - 2\eta NL)\left(F(\bar{x}_{r\tau+1}) - F(x^\star)\right)$$

$$+ \frac{16\eta L^2 N}{\mu}\mathbb{E}\left\|\mathbb{1}\bar{x}_{r\tau+1} - \boldsymbol{x}_{r\tau}\right\|^2 + 2\eta^2\sigma^2 \tag{141}$$

and the consensus one:

$$\mathbb{E}\left\|\mathbb{1}\bar{x}_{r\tau+1} - \boldsymbol{x}_{r\tau}\right\|_F^2$$

$$\leq \left(1 - \frac{q_{\min}}{3}\right)\left\|\mathbb{1}\bar{x}_{(r-1)\tau+1} - \boldsymbol{x}_{(r-1)\tau}\right\|_F^2 + \frac{8\eta^2 N^2}{q_{\min}}\left\|\nabla F(\bar{x}_{r\tau+1})\right\|^2 + \frac{8\eta^2 N^2}{q_{\min}}\sigma^2$$

$$\leq \left(1 - \frac{q_{\min}}{3}\right)\left\|\mathbb{1}\bar{x}_{(r-1)\tau+1} - \boldsymbol{x}_{(r-1)\tau}\right\|_F^2 + \frac{16\eta^2 N^2 L}{q_{\min}}\left(F(\bar{x}_{r\tau+1}) - F(x^\star)\right) + \frac{8\eta^2 N^2}{q_{\min}}\sigma^2$$

To lighten the notation, we let $A_{r+1} = \mathbb{E}\left\|\bar{x}_{(r+1)\tau+1} - x^\star\right\|^2$ and $B_r = \mathbb{E}\left\|\mathbb{1}\bar{x}_{r\tau+1} - \boldsymbol{x}_{r\tau}\right\|_F^2$. Therefore, we have

$$A_{r+1} \leq \left(1 - \frac{\eta\mu N}{2}\right)A_r - 2\eta N(1 - 2\eta NL)\left(F(\bar{x}_{r\tau+1}) - F(x^\star)\right) + \frac{16\eta L^2 N}{\mu}B_r + 2\eta^2\sigma^2$$

$$B_r \leq \left(1 - \frac{q_{\min}}{3}\right)B_{r-1} + \frac{16\eta^2 N^2 L}{q_{\min}}\left(F(\bar{x}_{r\tau+1}) - F(x^\star)\right) + \frac{8\eta^2 N^2}{q_{\min}}\sigma^2$$

After summing up them, the term about $\left(F(\bar{x}_{r\tau+1}) - F(x^\star)\right)$ is negative, so we directly remove it in the upper bound. Then, we get

$$A_{r+1} + \left(1 - \frac{16\eta L^2 N}{\mu}\right)B_r \leq \left(1 - \frac{\eta\mu N}{2}\right)A_r + \left(1 - \frac{q_{\min}}{3}\right)B_{r-1} + \left(2\eta^2 + \frac{8\eta^2 N^2}{q_{\min}}\right)\sigma^2$$

$$\leq \left(1 - \frac{\eta\mu N}{2}\right)\left(A_r + \left(1 - \frac{16\eta L^2 N}{\mu}\right)B_{r-1}\right) + \left(2\eta^2 + \frac{8\eta^2 N^2}{q_{\min}}\right)\sigma^2, \tag{142}$$

where (142) holds when $\eta \leq \frac{2q_{\min}}{3N\left(\frac{16L^2}{\mu}+\frac{\mu}{2}\right)}$. Denoting $\Gamma_r := A_r + \left(1 - \frac{16\eta L^2 N}{\mu}\right)B_{r-1}$, (142) will be

$$\Gamma_{r+1} \leq \left(1 - \frac{\eta\mu N}{2}\right)\Gamma_r + \left(2 + \frac{8N^2}{q_{\min}}\right)\eta^2\sigma^2 \leq \left(1 - \frac{\eta\mu N}{2}\right)^R \Gamma_0 + \frac{4(q_{\min}+N^2)}{\mu N q_{\min}}\eta\sigma^2$$

$\square$

Hence, we conclude that `FOCUS` achieves the linear convergence rate of $(1 - \eta\mu N/2)$ to $O(\eta)$-level neighborhood of the optimal solution.

### F.4 Proof of the Convergence of `SG-FOCUS` (Non-Convexity with PL Assumption)

**Theorem 5** (Convergence of `SG-FOCUS` for Nonconvex Functions with the PL Condition). *Under assumptions 1, 2, 4, 6, if the learning rate $\eta \leq \min\left\{\frac{q_{\min}}{3\beta N(4-q_{\min})}, \frac{q_{\min}}{16L^2}, \frac{q_{\min}^{3/2}}{4L\sqrt{6N}}, \frac{q_{\min}^{3/2}}{8\sqrt{2}L(\tau-1)}\right\}$,*

$$\Omega_R \leq (1 - \eta\beta N)^R \Omega_0 + \left(\frac{L}{2} + 32(\tau-1)^2 L^2 + \frac{8}{q_{\min}}\right)\frac{\eta}{\beta}\sigma^2,$$

where $\Omega_r := F(\bar{x}_{r\tau+1}) - F^\star + (1 - 4\eta L^2)\,\mathbb{E}\,\|\mathbb{1}\bar{x}_{(r-1)\tau+1} - \boldsymbol{x}_{(r-1)\tau}\|^2$ and $F^\star$ is the optimal value.

*Proof of Theorem 5:*

Using the $L$-Lipschitz condition, we have

$$\mathbb{E}\left[F(\bar{x}_{(r+1)\tau+1})\right]$$

$$\leq F(\bar{x}_{(r+1)\tau}) - \eta\mathbb{E}\left\langle\nabla F(\bar{x}_{(r+1)\tau}), \mathbb{1}^\mathsf{T}\nabla\boldsymbol{f}(\boldsymbol{x}_{(r+1)\tau};\xi_{(r+1)\tau})\right\rangle + \frac{\eta^2 L}{2}\mathbb{E}\left\|\mathbb{1}^\mathsf{T}\nabla\boldsymbol{f}(\boldsymbol{x}_{(r+1)\tau};\xi_{(r+1)\tau})\right\|^2$$

$$\leq F(\bar{x}_{(r+1)\tau}) - \eta\left\langle\nabla F(\bar{x}_{(r+1)\tau}), \mathbb{1}^\mathsf{T}\nabla\boldsymbol{f}(\boldsymbol{x}_{(r+1)\tau})\right\rangle + \frac{\eta^2 L}{2}\left\|\mathbb{1}^\mathsf{T}\nabla\boldsymbol{f}(\boldsymbol{x}_{(r+1)\tau})\right\|^2 + \frac{\eta^2 LN}{2}\sigma^2$$

$$= F(\bar{x}_{r\tau+1}) - \eta\left\langle\nabla F(\bar{x}_{r\tau+1}), \mathbb{1}^\mathsf{T}\nabla\boldsymbol{f}(\boldsymbol{x}_{(r+1)\tau})\right\rangle + \frac{\eta^2 L}{2}\left\|\mathbb{1}^\mathsf{T}\nabla\boldsymbol{f}(\boldsymbol{x}_{(r+1)\tau})\right\|^2 + \frac{\eta^2 LN}{2}\sigma^2$$

$$\leq F(\bar{x}_{r\tau+1}) - \eta\left\langle\nabla F(\bar{x}_{r\tau+1}), \mathbb{1}^\mathsf{T}\nabla\boldsymbol{f}(\boldsymbol{x}_{(r+1)\tau})\right\rangle + \eta^2 L\left\|N\nabla F(\bar{x}_{r\tau+1}) - \mathbb{1}^\mathsf{T}\nabla\boldsymbol{f}(\boldsymbol{x}_{(r+1)\tau})\right\|^2$$

$$\qquad + \eta^2 N^2 L\|\nabla F(\bar{x}_{r\tau+1})\|^2 + \frac{\eta^2 LN}{2}\sigma^2,$$

where the last inequality uses Jensen's inequality. Using the parallelogram identity to address the cross term, we have

$$-\eta\left\langle\nabla F(\bar{x}_{r\tau+1}), \mathbb{1}^\mathsf{T}\nabla\boldsymbol{f}(\boldsymbol{x}_{(r+1)\tau})\right\rangle$$

$$= -\frac{\eta}{N}\left\langle N\nabla F(\bar{x}_{r\tau+1}), \mathbb{1}^\mathsf{T}\nabla\boldsymbol{f}(\boldsymbol{x}_{(r+1)\tau})\right\rangle$$

$$= \frac{\eta}{2N}\left\|\mathbb{1}^\mathsf{T}\nabla\boldsymbol{f}(\mathbb{1}\bar{x}_{r\tau+1}) - \mathbb{1}^\mathsf{T}\nabla\boldsymbol{f}(\boldsymbol{x}_{(r+1)\tau})\right\|^2 - \frac{\eta N}{2}\|\nabla F(\bar{x}_{r\tau+1})\|^2 - \frac{\eta}{2N}\left\|\mathbb{1}^\mathsf{T}\nabla\boldsymbol{f}(\boldsymbol{x}_{(r+1)\tau})\right\|^2$$

Substituting back, we have

$$\mathbb{E}\left[F(\bar{x}_{(r+1)\tau+1})\right]$$

$$\leq F(\bar{x}_{r\tau+1}) + \frac{\eta}{2N}\left\|\mathbb{1}\nabla\boldsymbol{f}(\mathbb{1}\bar{x}_{r\tau+1}) - \mathbb{1}^\mathsf{T}\nabla\boldsymbol{f}(\boldsymbol{x}_{(r+1)\tau})\right\|^2 - \frac{\eta N}{2}\|\nabla F(\bar{x}_{r\tau+1})\|^2 - \frac{\eta}{2N}\left\|\mathbb{1}^\mathsf{T}\nabla\boldsymbol{f}(\boldsymbol{x}_{(r+1)\tau})\right\|^2$$

$$\qquad + \eta^2 L\left\|N\nabla F(\bar{x}_{r\tau+1}) - \mathbb{1}^\mathsf{T}\nabla\boldsymbol{f}(\boldsymbol{x}_{(r+1)\tau})\right\|^2 + \eta^2 N^2 L\|\nabla F(\bar{x}_{r\tau+1})\|^2 + \frac{\eta^2 LN}{2}\sigma^2$$

$$\leq F(\bar{x}_{r\tau+1}) + \left(\frac{\eta}{2N} + \eta^2 L\right)\left\|\mathbb{1}\nabla\boldsymbol{f}(\mathbb{1}\bar{x}_{r\tau+1}) - \mathbb{1}^\mathsf{T}\nabla\boldsymbol{f}(\boldsymbol{x}_{(r+1)\tau})\right\|^2$$

$$\qquad - \left(\frac{\eta N}{2} - \eta^2 N^2 L\right)\|\nabla F(\bar{x}_{r\tau+1})\|^2 + \frac{\eta^2 LN}{2}\sigma^2$$

$$\leq F(\bar{x}_{r\tau+1}) + \frac{\eta}{N}\left\|\mathbb{1}\nabla\boldsymbol{f}(\mathbb{1}\bar{x}_{r\tau+1}) - \mathbb{1}^\mathsf{T}\nabla\boldsymbol{f}(\boldsymbol{x}_{(r+1)\tau})\right\|^2 - \eta N\|\nabla F(\bar{x}_{r\tau+1})\|^2 + \frac{\eta^2 LN}{2}\sigma^2$$

$$\leq F(\bar{x}_{r\tau+1}) + \frac{\eta}{N}\left\|\mathbb{1}\nabla\boldsymbol{f}(\mathbb{1}\bar{x}_{r\tau+1}) - \mathbb{1}^\mathsf{T}\nabla\boldsymbol{f}(\boldsymbol{x}_{(r+1)\tau})\right\|^2 + \eta\beta N\left(F^\star - F(\bar{x}_{r\tau+1})\right)$$

$$\qquad - \frac{\eta N}{2}\|\nabla F(\bar{x}_{r\tau+1})\|^2 + \frac{\eta^2 LN}{2}\sigma^2,$$

where the third inequality follows $\eta \leq \min\{\frac{1}{4LN}, \frac{1}{2(\tau-1)LN}\}$. Next, we minus $F^\star$ on the both sides to obtain:

$$F(\bar{x}_{(r+1)\tau+1}) - F^\star \leq (1 - \eta\beta N)\left(F(\bar{x}_{r\tau+1}) - F^\star\right) + \frac{\eta}{N}\left\|\mathbb{1}\nabla\boldsymbol{f}(\mathbb{1}\bar{x}_{r\tau+1}) - \mathbb{1}^\mathsf{T}\nabla\boldsymbol{f}(\boldsymbol{x}_{(r+1)\tau})\right\|^2$$

$$- \frac{\eta N}{2}\|\nabla F(\bar{x}_{r\tau+1})\|^2 + \frac{\eta^2 LN}{2}\sigma^2 \qquad (143)$$

Recalling the previous result (136) and putting it into (143), we have:

$$F(\bar{x}_{(r+1)\tau+1}) - F^\star \leq (1 - \eta\beta N)\left(F(\bar{x}_{r\tau+1}) - F^\star\right) + 4\eta L^2\|\mathbb{1}\bar{x}_{r\tau+1} - \boldsymbol{x}_{r\tau}\|^2 - \frac{\eta N}{2}\|\nabla F(\bar{x}_{r\tau+1})\|^2$$

$$+ \left(\frac{L}{2} + 32(\tau-1)^2 L^2 N\right)\eta^2\sigma^2$$

Recalling the Lemma 6, we know

$$\mathbb{E}\left\|\mathbb{1}\bar{x}_{r\tau+1} - \boldsymbol{x}_{r\tau}\right\|_F^2 \le \left(1 - \frac{q_{\min}}{3}\right)\left\|\mathbb{1}\bar{x}_{(r-1)\tau+1} - \boldsymbol{x}_{(r-1)\tau}\right\|_F^2 + \frac{8\eta^2 N^2}{q_{\min}}\left\|\nabla F(\bar{x}_{r\tau+1})\right\|^2 + \frac{8\eta^2 N}{q_{\min}}\sigma^2$$

With a condition $\eta \le \min\{\frac{3q_{\min}}{32N}, \frac{q_{\min}}{3\beta N(4-q_{\min})}, \frac{q_{\min}}{16L^2}\}$, we denote $A_r = F(\bar{x}_{r\tau+1}) - F^\star$ and $B_r = \mathbb{E}\left\|\mathbb{1}\bar{x}_{r\tau+1} - \boldsymbol{x}_{r\tau}\right\|^2$, and then we have

$$A_{r+1} + \left(1 - 4\eta L^2\right)B_r$$

$$\le (1 - \eta\beta N)A_r + \left(1 - \frac{q_{\min}}{3}\right)B_{r-1} + \left(\frac{LN}{2} + 32(\tau-1)^2 L^2 N + \frac{8N}{q_{\min}}\right)\eta^2\sigma^2$$

$$\le (1 - \eta\beta N)\left(A_r + \left(1 - 4\eta L^2\right)B_{r-1}\right) + \left(\frac{LN}{2} + 32(\tau-1)^2 L^2 N + \frac{8N}{q_{\min}}\right)\eta^2\sigma^2 \quad (144)$$

Denoting $\Omega_r := A_r + \left(1 - 4\eta L^2\right)B_{r-1}$, (144) will be

$$\Omega_{r+1} \le (1 - \eta\beta N)\Omega_r + \left(\frac{LN}{2} + 32(\tau-1)^2 L^2 N + \frac{8N}{q_{\min}}\right)\eta^2\sigma^2$$

$$\le (1 - \eta\beta N)^R \Omega_0 + \left(\frac{L}{2} + 32(\tau-1)^2 L^2 + \frac{8}{q_{\min}}\right)\frac{\eta}{\beta}\sigma^2$$

$\square$

Therefore, we conclude that `SG-FOCUS` achieves the linear convergence rate of $(1 - \eta\beta N)$ to $O(\eta)$-level neighborhood of the optimal solution in the nonconvex case with PL condition.

### F.5   Proof of the Convergence of `SG-FOCUS` (General Non-Convexity)

**Theorem 6** (Convergence of `SG-FOCUS` for General Nonconvex Functions). *Under assumptions 1, 2 and 6, if the learning rate* $\eta \le \min\{\frac{q_{\min}^{3/2}}{4L\sqrt{6N}}, \frac{q_{\min}^{3/2}}{8\sqrt{2}L(\tau-1)}, \frac{1}{4LN}, \frac{1}{4L(\tau-1)}\}$,

$$\frac{1}{R}\sum_{r=0}^{R-1}\mathbb{E}\left\|\nabla F(\bar{x}_{r\tau+1})\right\|^2 \le \frac{8\left(F(\bar{x}_1) - F(\bar{x}^\star)\right)}{\eta N R} + \left(2LN + \frac{8N^2}{(N - 8L^2)q_{\min}}\right)\eta^2\sigma^2$$

*Proof of Theorem 6:*

We begin with $L-$Lipschitz condition:

$$\mathbb{E}\left[F(\bar{x}_{(r+1)\tau+1})\right]$$

$$\le F(\bar{x}_{r\tau+1}) - \eta\mathbb{E}\left\langle\nabla F(\bar{x}_{r\tau+1}), \mathbb{1}^\mathsf{T}\nabla\boldsymbol{f}(\boldsymbol{x}_{(r+1)\tau}; \xi_{(r+1)\tau})\right\rangle + \frac{\eta^2 L}{2}\mathbb{E}\left\|\mathbb{1}^\mathsf{T}\nabla\boldsymbol{f}(\boldsymbol{x}_{(r+1)\tau}; \xi_{(r+1)\tau})\right\|^2$$

$$\le F(\bar{x}_{r\tau+1}) - \eta\left\langle\nabla F(\bar{x}_{r\tau+1}), \mathbb{1}^\mathsf{T}\nabla\boldsymbol{f}(\boldsymbol{x}_{(r+1)\tau})\right\rangle + \frac{\eta^2 L}{2}\left\|\mathbb{1}^\mathsf{T}\nabla\boldsymbol{f}(\boldsymbol{x}_{(r+1)\tau})\right\|^2 + \frac{\eta^2 LN}{2}\sigma^2$$

$$\le F(\bar{x}_{r\tau+1}) - \eta\left\langle\nabla F(\bar{x}_{r\tau+1}), \mathbb{1}^\mathsf{T}\nabla\boldsymbol{f}(\boldsymbol{x}_{(r+1)\tau})\right\rangle + \eta^2 L\left\|N\nabla F(\bar{x}_{r\tau+1}) - \mathbb{1}^\mathsf{T}\nabla\boldsymbol{f}(\boldsymbol{x}_{(r+1)\tau})\right\|^2$$

$$+ \eta^2 N^2 L\|\nabla F(\bar{x}_{r\tau+1})\|^2 + \frac{\eta^2 LN}{2}\sigma^2, \quad (145)$$

where the last inequality utilizes the Jensen's inequality. Then, we deal with the cross term:

$$-\eta\left\langle\nabla F(\bar{x}_{r\tau+1}), \mathbb{1}^\mathsf{T}\nabla\boldsymbol{f}(\boldsymbol{x}_{(r+1)\tau})\right\rangle$$

$$= -\eta N\|\nabla F(\bar{x}_{r\tau+1})\|^2 + \eta\left\langle\nabla F(\bar{x}_{r\tau+1}), N\nabla F(\bar{x}_{r\tau+1}) - \mathbb{1}^\mathsf{T}\nabla\boldsymbol{f}(\boldsymbol{x}_{(r+1)\tau})\right\rangle$$

$$\le -\frac{\eta N}{2}\|\nabla F(\bar{x}_{r\tau+1})\|^2 + \frac{\eta}{2N}\left\|\mathbb{1}\nabla\boldsymbol{f}(\mathbb{1}\bar{x}_{r\tau+1}) - \mathbb{1}^\mathsf{T}\nabla\boldsymbol{f}(\boldsymbol{x}_{(r+1)\tau})\right\|^2 \quad (146)$$

Substituting (146) back to (145), we have

$$\frac{\eta N}{2}(1 - 2\eta NL)\mathbb{E}\left\|\nabla F(\bar{x}_{r\tau+1})\right\|^2 \le \mathbb{E}\left[F(\bar{x}_{r\tau+1}) - F(\bar{x}_{(r+1)\tau+1})\right] + \frac{\eta^2 LN}{2}\sigma^2$$

$$+ \left( \frac{\eta}{2N} + \eta^2 L \right) \mathbb{E} \left\| \mathbb{1}^{\mathsf{T}} \nabla \boldsymbol{f}(\bar{x}_{r\tau+1}) - \mathbb{1}^{\mathsf{T}} \nabla \boldsymbol{f}(\boldsymbol{x}_{(r+1)\tau}) \right\|^2$$

To bound the term $\left\| \mathbb{1}^{\mathsf{T}} \nabla \boldsymbol{f}(\bar{x}_{r\tau+1}) - \mathbb{1}^{\mathsf{T}} \nabla \boldsymbol{f}(\boldsymbol{x}_{(r+1)\tau}) \right\|^2$, we can directly use the result (136), so we will obtain:

$$\frac{\eta N}{2}(1 - 2\eta NL)\mathbb{E}\|\nabla F(\bar{x}_{r\tau+1})\|^2 \leq F(\bar{x}_{r\tau+1}) - F(\bar{x}_{(r+1)\tau+1}) + 2\left( \frac{LN}{2} + \frac{\eta}{2N} + \eta^2 L \right) \eta^2 \sigma^2$$
$$+ \left( \frac{\eta}{2N} + \eta^2 L \right) \left( 2L^2 N + 8\eta^2(\tau-1)^2 L^4 N \right) \mathbb{E} \|\mathbb{1}\bar{x}_{r\tau+1} - \boldsymbol{x}_{r\tau}\|^2$$

When $\eta \leq \min\{\frac{1}{4LN}, \frac{1}{4L(\tau-1)}\}$, we can simplify it further as follows:

$$\mathbb{E}\|\nabla F(\bar{x}_{r\tau+1})\|^2 \leq \frac{4}{\eta N}\left( F(\bar{x}_{r\tau+1}) - F(\bar{x}_{(r+1)\tau+1}) \right) + \frac{8L^2}{N}\mathbb{E}\|\mathbb{1}\bar{x}_{r\tau+1} - \boldsymbol{x}_{r\tau}\|^2 + 2LN\eta^2\sigma^2$$

Recalling the Lemma 6, we know

$$\mathbb{E}\|\mathbb{1}\bar{x}_{r\tau+1} - \boldsymbol{x}_{r\tau}\|_F^2 \leq \left(1 - \frac{q_{\min}}{3}\right)\|\mathbb{1}\bar{x}_{(r-1)\tau+1} - \boldsymbol{x}_{(r-1)\tau}\|_F^2 + \frac{8\eta^2 N^2}{q_{\min}}\|\nabla F(\bar{x}_{r\tau+1})\|^2 + \frac{8\eta^2 N}{q_{\min}}\sigma^2$$

When $\eta \leq \frac{3}{4\sqrt{2}N}$, we have

$$\frac{1}{R}\sum_{r=0}^{R-1}\mathbb{E}\|\nabla F(\bar{x}_{r\tau+1})\|^2$$
$$\leq \frac{8\left(F(\bar{x}_1) - F(\bar{x}^\star)\right)}{\eta NR} + \frac{16L^2}{NR}\left(1 - \frac{q_{\min}}{3}\right)^R \|\mathbb{1}\bar{x}_1 - \boldsymbol{x}_0\|_F^2 + \left(2LN + \frac{8N^2}{(N - 8L^2)q_{\min}}\right)\eta^2\sigma^2$$

If the initial models of clients and the server are the same, then $\|\mathbb{1}\bar{x}_1 - \boldsymbol{x}_0\|_F^2 = 0$. The final convergence rate is simply

$$\frac{1}{R}\sum_{r=0}^{R-1}\mathbb{E}\|\nabla F(\bar{x}_{r\tau+1})\|^2 \leq \frac{8\left(F(\bar{x}_1) - F(\bar{x}^\star)\right)}{\eta NR} + \left(2LN + \frac{8N^2}{(N - 8L^2)q_{\min}}\right)\eta^2\sigma^2$$

$\square$

# G Supplementary Experiments for `SG-FOCUS`

## G.1 Experiment Setup

To examine `SG-FOCUS`'s performance under arbitrary client participation and the highly non-iid conditions, we compare `SG-FOCUS` with FedAU [Wang and Ji, 2024] and SCAFFOLD [Karimireddy et al., 2020] on the image classification task by using the CIFAR10 [Krizhevsky et al., 2009] dataset. The model we used is a three-layer convolutional neural network. As for our baselines, FedAU is a typical FL algorithm designed to tackle unknown client participation, and SCAFFOLD is a classic FL algorithm designed to deal with data heterogeneity. Therefore, it is reasonable to select these two FL algorithms as our baselines.

**The source code for the implementation is provided in the attached supplementary material.** Our hyperparameter settings are: learning rate $\eta = 2e{-}3$, local update $\tau = 3$, the number of clients $N = 32$, total communication rounds $r = 10000$. To ensure totally arbitrary client participation, we do not restrict the number of participating clients in each round. To simulate highly non-iid data distribution, we use Dirichlet distribution and use $\alpha = 0.05$ to model that. Note that a smaller $\alpha$ represents higher data heterogeneity. We illustrate the highly heterogeneous data distribution we used across 32 clients in our experiments in Figure 8. The size of the bubble reflects the number of data points of the particular class used in that client. From Figure 8, we can roughly observe that the data of each client mainly covers 1-2 classes, so the degree of non-iid data is quite high.

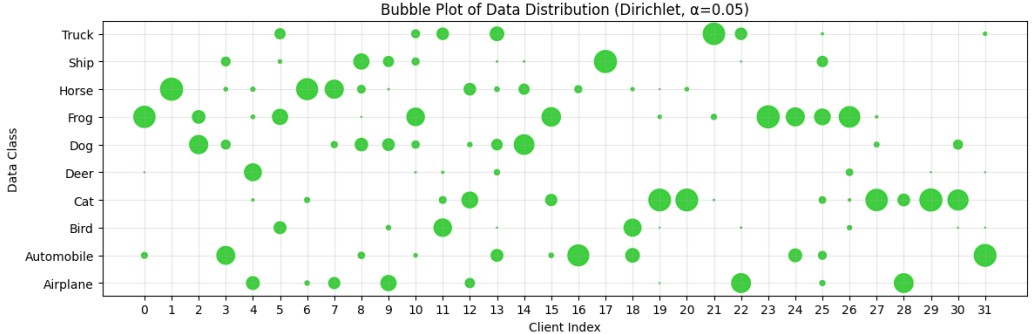

Figure 8: Dirichlet Data Distribution ($\alpha = 0.05$)

## G.2   Experiment Results of SG-FOCUS

From Figure 9, we observe that under highly heterogeneous data distribution ($\alpha = 0.05$) and arbitrary client participation/sampling, our SG-FOCUS shows the best performance on convergence speed and test accuracy, compared to FedAU, MIFA, and SCAFFOLD. These results echo the view in our main paper: SG-FOCUS can jointly handle both data heterogeneity and arbitrary client participation.

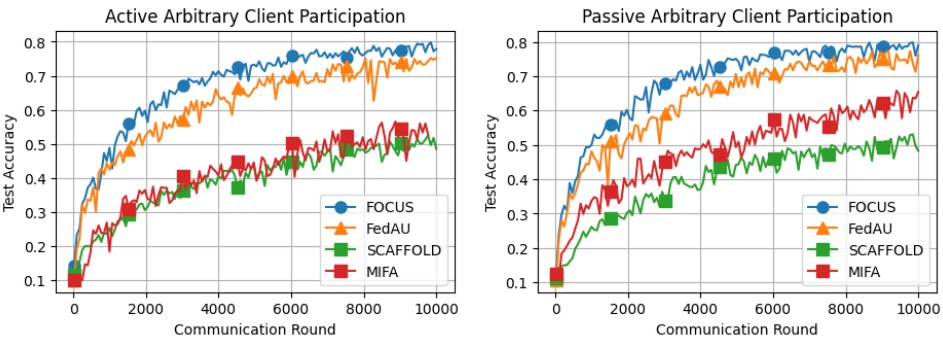

Figure 9: Performance Comparison of SG-FOCUS and Baselines on CIFAR10 Dataset under Arbitrary Client Participation and High Data Heterogeneity.

