# OpenReview forum: "Exact and Linear Convergence for Federated Learning under Arbitrary Client Participation is Attainable"
_NeurIPS.cc/2025/Conference — NeurIPS 2025 poster_

### Official Review · Reviewer_DDXi · 2025-06-21

**Clarity:** 3
**Significance:** 3
**Originality:** 4
**Rating:** 6
**Confidence:** 5

**Summary:**

This paper addresses a fundamental problem in Federated Learning (FL): achieving exact convergence under arbitrary client participation and data heterogeneity without resorting to diminishing learning rates. The authors propose a new algorithm, FOCUS, inspired by decentralized optimization techniques, especially push-pull gradient tracking. The paper makes strong theoretical contributions; in particular, it models client participation using stochastic matrices and establishes sound convergence guarantees, with no bounded heterogeneity or participation bias assumptions. I believe this paper provides new insights into federated learning algorithms and has the potential to inspire further research in the field.

**Questions:**

This part collects some minor concerns on the paper, mainly in presentation and writing.

1. In line 6, the article states that "we introduce the concept of stochastic matrix ...". This wording may create the impression that the concept of stochastic matrix is being introduced by this work.

2. In line 52, the authors claim that FOCUS exhibits linear convergence for both **convex** and non-convex scenarios. I suggest to replace _convex_ with _strongly convex_.

3. In lines 54-55, the authors claim that SG-FOCUS demonstrates **faster** convergence and **higher** accuracy. It is recommended to clarify the other side of the comparison. It is weird that a stochastic-gradient variant converges faster than its deterministic counterpart.

4. In (2c), the authors may need to double check whether $x_{\tau+1,i}^{(r)}$ should be $x_{\tau,i}^{(r)}$.

5. In lines 74-75, the authors state that "Yet, this approach incurs increased communication costs, doubling them
due to the transmission of a control variate with the same dimensionality as the model parameters." However, [1, Sec. 4.2] shows that SCAFFOLD can be implemented with a single communicated variable. So this claim should be revisited and clarified.

6. In Assumption 1, "which occurs with **a** unknown probability" should be "which occurs with **an** unknown probability".

7. In the paragraph in lines 152-158, the authors use both $\mathbb 0$ and $\mathbf 0$. Are they the same?

8. In line 184, "equivalency" should be "equivalence".

[1] X. Huang, P. Li, X. Li. Stochastic controlled averaging for federated learning with communication compression. ICLR 2024.

**Ethical Concerns:**

["NO or VERY MINOR ethics concerns only"]

**Final Justification:**

The authors have addressed all my concerns and I remain very positive on the article. To reiterate, I believe the key idea is innovative and departs meaningfully from prior work.

**Limitations:**

yes

**Quality:**

4

**Strengths And Weaknesses:**

**Strength**

To me, the idea in the paper is innovative and departs meaningfully from prior work. I believe the key contribution is a novel viewpoint to understand federated learning algorithms. Previous work often states that optimization methods in FL are special cases with decentralized algorithms with the communication network being the fully-connected graph, or equivalently, $W = (1/N) \mathbf{1} \mathbf{1}^T$. This paper departs from the mainstream and introduces a novel analytical framework grounded in the push-pull strategy in decentralized optimization.

**Areas to improve**

1. Is it possible to compare FOCUS with SCAFFOLD? I understand that they are just different. But I wonder whether the authors have any intuition behind the distinction of update rules.

2. Do the authors have an intuition why pushing gradient $\boldsymbol{y}$ requires column style rather than row style?  In other words, why does the update of $y_{r+1}$ requires the **sum** of $y_{\tau, i}^{(r)}$ rather than the **average**?

---

> ### Author Rebuttal · Authors · 2025-07-28
>
> We sincerely thank the reviewer for their positive evaluation and encouraging feedback. We also appreciate the thoughtful comments and helpful corrections, which have allowed us to further improve the manuscript. Our detailed responses to each point are provided below.
>
> ## Two main questions:
>
> > Area to improve 1:  Is it possible to compare FOCUS with SCAFFOLD?
>
> This is an excellent and thought-provoking question. Actually, it prompts us to ask a broader, more fundamental question: **What is the connection between the gradient tracking technique in FOCUS and the variance reduction technique in SCAFFOLD?**
>
> Pondering on this, in conjunction with helpful suggestions from other reviewers to compare with MIFA, has led us to some new insights. We get a preliminary conclusion: **in a simplified distributed computation scenario, the server-side update rules of SCAFFOLD, MIFA, and our proposed FOCUS can all be interpreted as distinct variants of the distributed SAGA algorithm.**
>
> To illustrate this, we analyze a simplified scenario. We consider only a single local update per round and focus on the server-side update mechanism. In FL setting , the noise from stochastic gradient estimation is equivalent to the noise introduced by only sampling a subset of clients \$S\$. The global gradient and the average of sampled gradients has the following relation:
>
> $$
> \nabla f(x) = \frac{1}{N}\sum_{i=0}^{N-1} \nabla f_i(x) = \frac{1}{|S|}\sum_{i\in S} \nabla f_i(x) + {\rm stochastic\ noise}
> $$
>
> To achieve exact convergence, an algorithm's update direction must asymptotically approach the true global gradient. But the client sampling restriction implies that we can only access the part of local gradient: $\{\nabla f_i(x)\}_{i\in S}$. The main idea of SAGA algorithm is that accomplishes this by maintaining an auxiliary memory of the most recent gradient computed for each client. With some slightly modifications, we adopt the following notation $g_i = \nabla f_i (x'_i)$, where $x'_i$ is the last access time of computing the gradient, or equivalently the last participation time in FL setting. The SAGA update is
>
> $$
> \hat{g} = \frac{1}{|S|}\sum_{i\in S} (\nabla f_i(x) - g_i) + \frac{1}{N}\sum_{i=0}^{N-1} g_i
> $$
>
> While gradient tracking (used in FOCUS) and variance reduction (used in SCAFFOLD/MIFA) are generally distinct techniques, they become analogous in our specific federated learning setting. The core idea on the server side is the same: the tracking variable $y_r$ is used to asymptotically track the true global gradient.
>
> To see this clearly, consider the update rule for the tracker $y_r$ in FOCUS (simplified for this analysis):
> $$
> y_r = y_{r-1} + \frac{1}{|S|}\sum_{i\in S} (\nabla f_i(x_{r,i}) - \nabla f_i(x_{r',i}))
> $$
> By comparing these two update equations, the parallel becomes evident. The historical tracker $y_r$ in FOCUS plays the same role as the full historical gradient average $\frac{1}{N}\sum_{i=0}^{N-1} g_i$ in SAGA. The incremental update to the tracker in FOCUS is analogous to the correction term in the SAGA estimator. This observation reinforces our discussion in Section 5.2 regarding delayed gradient interpolation, showing that our tracking mechanism functions as a SAGA-like scheme in this context.
>
>
> SCAFFOLD and MIFA also fit this SAGA-like framework. They both use control variates (global $c$ and local $c_i$) to correct for the client sampling noise. As noted in the SCAFFOLD paper, their approach is SAGA-like in that it tracks this correction term incrementally, differing from SVRG, which requires periodic full gradient computations. The primary architectural difference between them is that SCAFFOLD maintains its control variates on the client-side, while MIFA stores them on the server-side.
>
> However, this analogy becomes less direct on the client-side, especially when multiple local updates are performed, as this introduces client drift dynamics not captured by the basic SAGA model. We will add a new section to the appendix to formalize this discussion and elaborate on these connections. We thank the reviewers again for stimulating this valuable thought and are happy to discuss more on this.
>
>
> > Area to improve 2:  why pushing gradient $\mathbf{y}$ requires column style rather than row style?
>
> This is another excellent and important question. It directly touches upon one of the central messages of our paper: that an unexplored family of Federated Learning algorithms might be systematically designed using column-stochastic updates. .
>
> First, we'd like to clarify that **an averaging-based approach is possible, but only under the specific condition that the number of participating clients, when  $|S_r|$ is fixed for all $r$.** Because we can rescale $y' = \frac{1}{S} y$ for all $y$, where $S \equiv |S_{r}|,\forall r$, the algorithm will be equivalent upto some scaling factor on step size. Conversely, and more importantly, this implies that when the number of participating clients varies over time—a common scenario in practical federated learning—a simple averaging-style algorithm is no longer equivalent and cannot replicate our method's dynamics.
>
> A key intuition behind our design comes from the distinct properties of column and row stochastic matrices. **The column stochastic matrix, C, provides a "mass conservation" property.** Imagine a column vector $y$ where each element $y_{i}$ represent some mass of the $i-$th object. In the operation $y^+ = Cy$  ,the total sum of these masses, $\sum_{i} y_{i}$ is preserved (to see that, left multiplying $\mathbb{1}^T$). In our algorithm, $y_{i}$ is analogous to the local gradient information at client $i$. Using a column stochastic matrix ensures that the sum of these local quantities, which represents the global gradient (server-side), is not arbitrarily altered during aggregation.
>
> In contrast, a row stochastic matrix, $R$, drives the system toward consensus. Under mild conditions on $R$, repeated the process that $y^+ = Ry$, i.e. $y^{\infty} = \lim_{n\to\infty} R^n y$. causes all elements of the vector to converge to a single value. However, **this consensus value is a weighted average of the initial elements.** In the context of our optimization problem, converging to a weighted average of gradients would introduce a bias. This bias would prevent the algorithm from converging to the true minimum of the original problem, thereby destroying the exactness and linear convergence properties we aim to achieve.
>
> We hope this explanation sheds more light on our algorithm design and are happy to engage in further discussion on this topic.
>
> ## Other Minor Questions:
>
> We sincerely thank the reviewer for the careful proofreading and for identifying typos and ambiguities in the writing. We agree with all the points raised in questions 2, 4, 6, 7, and 8 and will incorporate these corrections into the revised manuscript. To keep our response concise, we provide detailed answers below for the remaining questions.
>
> > Q1:  we introduce the concept of stochastic matrix ..."
>
>  We appreciate the thorough thought of the reviewer. "First, we review the concept of stochastic matrix ..." should be a better wording.
>
> > Q3: ...  claim that SG-FOCUS demonstrates faster convergence and higher accuracy,
>
> Thanks for pointing this out! We agree that we should modify this contribution point as follows: "We also introduce a stochastic gradient variant, SG-FOCUS, which demonstrates faster convergence and higher accuracy than the other baselines (e.g., FedAU, SCAFFOLD), both theoretically and empirically."
>
> >  Q5: .. SCAFFOLD can be implemented with a single communicated variable...
>
> We thank the reviewer for this precise reference and for pointing out the need for clarification. Our initial statement was imprecise, and we will revise it. Just one point to clarify that in [1, Sec 4.2] the uplink (client-to-server) communication in SCAFFOLD can be optimized to a single variable. However, the downlink (server-to-client) communication still requires transmitting two variables: the global model and the server control variate (per Algorithm 4, line 5 in [1]).

---

### Official Review · Reviewer_PLZ6 · 2025-06-29

**Clarity:** 3
**Significance:** 3
**Originality:** 3
**Rating:** 5
**Confidence:** 4

**Summary:**

This paper propose FOCUS via push-pull strategy to address the arbitrary participation problem in federated learning.

**Questions:**

**Questions**:

1. It is unclear whether $x$ (or $y$) is a row or column vector. See $x$ in Equation (1) and $\boldsymbol{x}$ in Line 152; see $\boldsymbol{x}$ between Lines 444 and 445 and Line 446.

2. Lines 549-550: Why we say $\boldsymbol{y}_k$ has one more inner iteration?

3. Equation (67) and (68): Why is only $C_{k,all}$ defined ($R_{k,all}$ is not defined)?

4. Line 607:

   > the client's $y_{r\tau+1,i} = 0$ for any clients.

   Is this right?

5. Line 610:

   > the first equation means the difference between the model before pulling ...

   Does "the model" represents "the clients' model"?

6. Line 667:

   > (102) holds when $\eta \leq \frac{2q_\min}{3N \left( \frac{8L^2}{\mu}+\frac{\mu}{2}\right)}$.

   Does this cause an additional constraint $\eta \lesssim \frac{\mu q_min}{NL^2}$ for the strongly convex case in Theorem 1?

**Suggestions**:

1. Figures: It would be better to use the PDF format for the figures.

2. Theorem 1. It seems that the convergence of FOCUS is affected by $p_\min$. The convergence of FOCUS can be slow when $p_\min$ is very small (e.g., $p_\min = \frac{1}{R}$). I think this limitation can be added to Section 5.3.

3. Missing references. The following paper gave a similar conclusion to Wang and Ji [2023].

   Federated Learning Under Heterogeneous and Correlated Client Availability, IEEE/ACM Transaction on Networking, 2024.

4. Table 1:

   > There is no convergence proof of SCAFFOLD under arbitary client participation scenario.

   The following paper can be relevant.

   Federated Learning under Periodic Client Participation and Heterogeneous Data A New Communication-Efficient Algorithm and Analysis, NeurIPS, 2024.

5. Line 74, Section 2:

   > Yet, this approach incurs increased communication costs, doubling them due to the transmission of a control variate with the same dimensionality as the model parameters.

   This sentence can be misleading. See Appendix A.1 in the following paper for the efficient implementation of SCAFFOLD.

   TCT Convexifying Federated Learning using Bootstrapped Neural Tangent Kernels, NeurIPS, 2022.

6. Section 3.1: Could you kindly give detailed explanation (or proofs) for the conclusions in the special cases. For example, explain why $q_i \approx p_i/(\sum_j p_j)$ when $p_i$ are close to each other in Case 2, and explain why $p_i = 1-(1-q_i)^m$ when it is sampled with replacement in Case 3. I think this would make the cases easier to understand for the readers.

7. Section 3.1: Could you kindly give some practical FL scenarios (or give some simple and illustrative examples) for these cases, just like Appendix B.2 in the following paper?

   A Unified Analysis of Federated Learning with Arbitrary Client Participation, NeurIPS, 2022.

8. Lines 272-273.

   > To see that, leveraging the tracking property of the variable $y_k$ and special construction of matrix $C_k$, we can establish that the server's $y_{r+1} = \sum_{i=1}^N \nabla f_i(x_{k+1},i)$.

   Could you kindly add the proof for this?

9. Lines 275-276:

   > FOCUS effectively transforms arbitrary participation probabilities into an arbitrary delay in gradient updates.

   This interpretation is interesting. This sounds like MIFA in the following paper. Could you kindly add some comparison with respect to the convergence rates.

   Fast Federated Learning in the Presence of Arbitrary Device Unavailability, NeurIPS, 2021.

10. References. There are some papers which are cited as preprint, but have been published. For example, Wang and Ji [2023] has been published in ICLR, Xiang et al. [2024] has been published in NeurIPS.

11. Figure 9: Please add the results of FedAvg.

If the problems (weaknesses, suggestions, questions) I raised above are wrong, please free free to point them out.

**Ethical Concerns:**

["NO or VERY MINOR ethics concerns only"]

**Final Justification:**

I agree with other reviewers that this paper is over the bar.

The main concern in my initial review is that the main contents of this paper do not depend on the matrices, and the motivation of this paper is not so clear. The authors claimed that the framework (associated with the matrices) is also a contribution of this paper. This justification seems reasonable for me.

Thus, I decide to raise the rating to 5. Given the high overall evaluation of this paper, my expectations for it are correspondingly higher.

I hope that the authors discuss more about their framework: why the framework is important, how the framework help design new algorithms, which scenario the framework applies to. For the current version, the motivation for this framework and matrices is not quite clear. In addition, the authors can consider whether the framework can include the existing works for FL with arbitrary client participation, like FedAU [Wang and Ji, 2023], FedAWE [Xiang et al., 2024], MIFA [Gu et al., 2021]. If not, the authors can provide the reasons of the failure.

Transforming a simple federated learning problem into a complex decentralized federated learning problem still feels unnatural to me. More discussion is still required for this transformation. The authors could consider whether there is a scenario in which such a transformation becomes indispensable.

**Limitations:**

yes

**Quality:**

3

**Strengths And Weaknesses:**

**Strengths**:

1. This paper finds the relation between distributed optimization over directed graphs and federated learning with arbitrary participation.
2. This paper proposes FOCUS via push-pull strategy, which shows faster convergence than FedAvg.
3. The theory seems solid and rigorous.
4. The paper is well-written, well-organized.

**Weaknesses**:

**The motivation for constructing the matrices**. The constructions of the assign matrix ($R$), average matrices ($A$) and other matrices ($C$ and $W$) are interesting. However, the motivation for constructing these matrices **in this paper** is not strong or convincing. In my opinion, all the content can be presented clearly even without these constructions: (i) The model of arbitrary client participation relies on Assumption 1, instead of these constructions. (ii) The convergence analysis of FedAvg has been established in Wang and Ji [2022] without the constructed matrices. (iii) FOCUS can also be interpreted without these constructions (see Section 4.2.2). (iv) After checking the proofs of Theorem 1, I think the proofs can also be written without these constructions.

Using the matrices can increase the workload of reading this paper. For example, when checking the proofs, I need to revisit the definitions of some matrices (e.g., Appendices B, C.1). Some definitions of them are not straightforward, e.g., $\nabla \boldsymbol{f}(\boldsymbol{x}\_k) = [\nabla f_0(x\_{k,0}),\nabla f_1(x\_{k,1}),\ldots,\nabla f_N(x\_{k,N})]^T$ and $\nabla \boldsymbol{f}(1\bar x\_k) = [\nabla f_0(\bar x\_k),\nabla f_1(\bar x\_k),\ldots,\nabla f_N(\bar x\_k)]^T$ in Appendix B.

One example where the use of matrix notation is necessary: The construction of the mixing matrix in decentralized federated learning (or decentralized SGD) is necessary, as the assumptions relies on the mixing matrix. See Koloskova et al. [2020].

---

> ### Author Rebuttal · Authors · 2025-07-28
>
> We are sincerely grateful to the reviewer for providing such detailed and practical suggestions. Your thoughtful feedback has been quite helpful and has provided us with a clear path for improving our manuscript. We truly appreciate the time and effort you dedicated to the review. Our detailed responses to each point are provided below.
>
> ## Weakness:
>
> > 1. The motivation for constructing the matrices..
>
> **Our response:** We sincerely appreciate the reviewer's thoughtful suggestions on the paper's presentation. While we acknowledge that the suggested structure offers a valid alternative, we believe our current organization is essential for conveying the paper's core message for the following reasons:
>
> **Beyond presenting FOCUS as a standalone algorithm, our primary contribution is the introduction of a general and systematic framework for designing federated learning algorithms based on principled optimization rather than heuristics.**
>
> In this context, stochastic matrices are not merely a notational choice but the foundational modeling tool that makes this framework possible. They are crucial for several reasons:
>
> - They allow us to reformulate the communication process as an explicit mathematical constraint. This step is what enables the shift from heuristic design to formal optimization; without it, deriving an algorithm like FOCUS would be nearly impossible.
>     - For example, Sec 4.2.2 will be unmotivated without introducing column stochastic matrix. See eq. 18 and 19. Taking sum and reseting the variable $y=0$ are not arbitrary heuristics but principled steps justified by the column stochastic matrix.
> - This matrix-based formulation leads to more concise and elegant proofs.
> - Most importantly, it establishes a versatile framework that can serve as a foundation for developing other novel FL algorithms simply by employing different primal-dual solvers.
>
> Our manuscript is therefore structured to walk the reader through this exact methodology:
>
> 1. First, we use stochastic matrices to model communication patterns as explicit constraints.
> 2. Next, we reformulate the decentralized objective as a constrained optimization problem.
> 3. Finally, we apply a known optimization technique (a primal-dual method) to derive the final algorithm.
>
> We chose this explicit presentation because we believe the framework itself is as significant a contribution as the FOCUS algorithm—a point the other reviewers have kindly acknowledged. **We are confident that this transparent, first-principles approach provides a valuable new perspective and a generalizable tool for the federated learning community.**
>
> > 2. Using the matrices can increase the workload of reading this paper.
>
> **Our response:** We understand the reviewer's concern. Our matrix-based notation is indeed less common in federated learning literature, and we recognize it may present an initial hurdle. However, we chose it deliberately for several key advantages:
>
> - Conciseness: It allows us to avoid cumbersome summation notation ($\sum$) and express complex decentralized operations compactly. When combined with tools like the Frobenius norm, it makes the proofs more streamlined.
> - Geometric Intuition: We believe that reasoning about the null-space of matrices provides a more intuitive geometric perspective than manipulating long chains of element-wise inequalities.
>
> In the revision, we will add a dedicated subsection explaining our notational choices. To briefly illustrate why this notation is natural, recall the global objective $f(X) = \sum_i f_i(x_{i})$, taking the gradient with respect to each element we have $\nabla_{x_i} f(x) = \nabla f_i(X)$,
> let $\nabla f(X)= {\rm vstack}[\nabla_{x_0}, \cdots, \nabla_{x_N}] f(X)$. For $\mathbb{1}^T \bar{x}_k$, notice $f(\mathbb{1}^T \bar{x}_k) = \sum_i f_i(\bar{x}_k)$ so that the notation is consistent. We believe these notations, once grasped, offers a more elegant way to analyze the algorithm.
>
> **Due to space limitation, for question 1 and 5,  suggestion 1, 3, 5, 7, 10, and 11, we omit detailed response. We agree with these and will update the paper to reflect it.**
>
> ## Questions:
>
> > 2. Lines 549-550: Why we say $\mathbf{y}_k$ has one more inner iteration?
>
> This is a subtle but crucial point in our analysis, as it distinguishes the error introduced by local updates from the error due to data heterogeneity (Note $\tau$ versus $\tau-1$). Notice eq (38) and eq(39), under the same iteration $k$, $x_{k+1}=W_{k}y_{k+1}$ that an extra $W_k$ is applied to get $x_{k+1}$. When $W_{k}$ is not identical matrix(inner iteration/local update), $x_{k+1}$ will become some common value while $y_{k+1}$ is still some value varying among different agents.
>
> > 3. Equation (67) and (68): Why is only $C_{k, all}$ defined ($R_{k, all}$ is not defined)?
>
> It is the special feature of FL. The downlink communication (modeled by $C_k$) is pulling model, i.e., the server's model is overriding the client's model at the begin of iteration. Hence, we can virtually think the client is always pulling the server's model at all iterations no matter the client is active or not. Meanwhile, the uplink communication (modeled by $R_k$) doesn't have such properties. If we replace the $R_{k}$ by $R_{k, all}$, it change the behavior of algorithm, i.e., we average more client's model than necessary. You can refer the appendix line 488 and 492 and figure 6-7 in the appendix that we have some discussion there.
>
> > 4. Line 607: the client's $y_{r \tau+1, i}=0$ for any clients.
>
> Yes, it is because of the line 6 in algorithm 1. This very special behavior is induced by the properties column matrix $C_k$. When the client $i$ is sampled, it will reset the $y_k=0$ because all elements in the $i$-th row of $C_k$ are zero. Also see equation (17)-(19) and equation (69) for the virtual participation form. Lastly, combining the initial condition, we can conclude $y_{r \tau+1, i}=0$ for any clients.
>
> > 6. Line 667: .. Does this cause an additional stepsize constraint for the strongly convex case in Theorem 1?
>
> Thanks for the careful proof reading! We indeed missed this extra condition.
>
> ## Suggestions:
>
> > 2: FOCUS is affected by $p_\min$ or $q_\min$.
>
> On one side, it is intuitive that for a very sparsely participated client, the algorithm performance is deteriorated. On the other side, this bound might be further improved. Please refer our response to Reviewer AKZy, Weakness 3 & Q2 for more details. (Sorry we don't have enough space to expand this point here).
>
> > 4: convergence proof of SCAFFOLD under arbitrary participation
>
> Thanks for providing this reference! It is a very relevant one that we should include it. Based on our reading of this paper, it is still not the same scenario as we listed in our table. First, it provides the convergence theorem for Amplified SCAFFOLD, which requires the server to store the gradient information for all client at the server. Second, the proof is provided based on the cyclic particiation or regular participation. In Section 6, the limitation stated, "Our framework requires that all clients have an equal chance of participation across well-defined windows of time that are known to the algorithm
> implementer, which may not always hold." In our arbitrary participation setup, we do not require this equal participation assumption.
>
> > 6: give detailed explanation (or proofs) for the conclusions in the special cases.
>
> Special Case 2: Active Arbitrary Participation -- If $p_i$ is close to each other, we can denote it as $p_i = p + \epsilon_i$, where $\epsilon_i$ is a very small number.
>
> Special Case 3 Passive Arbitrary Participation -- we sample the $m$ client with replacement. If a client $i$ is not participated, it means it is not selected by all $m$ times, i.e., $(1-q_i)^m$. Hence, the participation probability $p_i$ is $1-(1-q_i)^m$.
>
>
> > 8: Add proof about Lines 272-273.
>
> We are happy to provide it. First recall the column stochastic matrix definition that $\mathbb{1}^T C = \mathbb{1}^T$. Left multiplying $\mathbb{1}^T$ to equation (12), we have (Because of display system, we use capital instead of boldsymbol representing the vstacked vectors)
>
> $$
> \mathbb{1}^T Y_{k+1} =  \mathbb{1}^T C(Y_k + \nabla f(X_{k+1}) - \nabla f(X_{k})) = \mathbb{1}^TY_{k} + \mathbb{1}^T\nabla f(X_{k+1}) - \mathbb{1}^T \nabla f(X_{k})
> $$
>
> Recalling the initial condition $\mathbb{1}^TY_{0} = \mathbb{1}^T \nabla f(X_{0})$, we can conclude $\mathbb{1}^TY_{k} = \mathbb{1}^T \nabla f(x_{k})$ for any $k$ by induction. Lastly, noticing all client's $y$ is reset to $0$ at end of local update, hence, the server's value $y_r = \mathbb{1}^T Y_{k}$.
>
> > 9: Add comparison with MIFA.
>
> We thank the reviewer for bringing the MIFA paper to our attention; it is indeed a relevant and interesting comparison.
>
> The core idea of MIFA is to correct gradient bias using memorized client updates, a technique inspired by the SAGA algorithm. This approach achieves convergence rates of $O(1/T)$ for (general) convex case and $O(1/\sqrt{T})$ for non-convex case. While the mechanism of using memorized updates is conceptually similar to ours, FOCUS achieves significantly stronger theoretical guarantees, including a linear convergence rate of $O(\exp^{-T})$ for strongly convex and $O(1/T)$ for nonconvex.
>
> The key difference—and the reason for this performance gap—stems from the design philosophy.
> - MIFA is presented as a heuristic combination of FedAvg and SAGA. This design effectively mitigates the effects of data heterogeneity but does not fundamentally resolve the client drift introduced by local client updates.
> - FOCUS, in contrast, is not a heuristic algorithm. It is systematically derived from a formal optimization formulation where communication and local updates are modeled as constraints.
>
> **This principled, optimization-based derivation is precisely why FOCUS can jointly resolve, arbitrary client participation, data heterogeneity, and client drift, leading to its superior convergence guarantees.**

---

> ### Comment · Reviewer_PLZ6 · 2025-08-05
> **Further suggestions**
>
> The rebuttal addressed my concerns. I will adjust my rating accordingly (in Reviewer-AC Discussions), after reading the comments of other reviewers.
>
> Some suggestions:
>
> 1. The authors need to revise the paper as they have promised.
>
> 2. For my weakness 1. The motivation behind the design of the matrices appears reasonable to me. That is, this paper aims to provide a general framework, beyond the FOCUS algorithm. The authors can talk more about the benefits of their general framework for designing FL algorithms in the next version. In the current version, only FOCUS and FedAvg are included in the framework. Is it possible to include other existing algorithms? This will convince me of the benefits of this framework. It is not necessary to answer this question in this rebuttal. However, I hope the authors can discuss it in the next version. The authors can also discuss the implications of this framework for future algorithm design.
>
> 3. > For example, Sec 4.2.2 will be unmotivated without introducing column stochastic matrix.
>
>    For my weakness 1. In addition, this example seems also interesting. The authors can also discuss more designing details that inspired by the framework for the algorithms.
>
> 4. The authors can discuss more about the push-pull algorithms in the decentralized algorithms in the existing works, as pointed out by Reviewer yWqo. The authors can discuss the relation and difference between them in the next version.

---

> ### Author Response · Authors · 2025-08-05
>
> We thanks again for reviewer's continuous suggestion to improve the quality of the manuscript. We will incorporate the suggestions, including the decentralized matrix-vector notations, the framework idea, and more push-pull intuition, in the revision accordingly.
>
> > Is it possible to include other existing algorithms?
>
> Actually, using our framework to design other novel algorithms is a promising direction that we are actively considering for future work. But we are happy to share some preliminary thoughts.
>
> To provide a concrete example, as we also discussed in our response to Reviewer yWqo (question 3), let's consider the push-sum (or Gradient-Push) algorithm. In the matrix-vector notation, the algorithm can be expressed as follows:
> \begin{align}
> w_{k+1} =& C_k w_{k}  \\\\
> z_{k+1} =& C_k(z_{k} - \eta \nabla F(x_{k}))\\\\
> x_{k+1} =& z_{k+1} / w_{k+1}
> \end{align}
> where $w_k \in \mathbb{R}^{N \times 1}$ is a vector that tracks **a scalar weight** for each client, and $x_k \in \mathbb{R}^{N \times d}$ is the final matrix of model parameters. The update at the last step is an element-wise division, where each client's $d$-dimensional model (a row in an intermediate state matrix $z_t$) is normalized by its corresponding scalar weight from $w_{k+1}$. For complete algorithmic details, please see reference [R].
>
> Next, following this same idea, we can model the client sampling process by using a specific column stochastic matrix, $C_k$. Substituting this matrix into the general form yields the concrete algorithm. However, there is one mild caveat for push-sum style algorithms: the normalization vector, $w_{k}$, must not contain any zero elements to avoid division by zero. For example, consider a system with 3 clients where only clients 2 and 3 are sampled. The column stochastic matrix at the iteration for the client pulling server will be like this:
> $$
> C_k = \begin{bmatrix}
> 0.4 & 0 & 0 & 0 \\\\
> 0 & 1 & 0 & 0 \\\\
> 0.3 & 0 & 1 & 0 \\\\
> 0.3 & 0 & 0 & 1
> \end{bmatrix}
> $$
> while the matrix at the iteration that the clients push the local updated value to the server will be like
> $$
> C_k = \begin{bmatrix}
> 1 & 0 & 0.5 & 0.5 \\\\
> 0 & 1 & 0 & 0 \\\\
> 0 & 0 & 0.5 & 0 \\\\
> 0 & 0 & 0 & 0.5
> \end{bmatrix}
> $$
>
> (Note other choices for the non-zero weights are possible)
>
> A key consequence is that **the algorithm then automatically compensates for varying participation levels through the weight vector $w_k$.** This introduces another novel method for correcting participation bias; instead of relying on the commonly used weighted averaging of models, this approach uses a direct element-wise division by the adaptive weights. Furthermore, this push-sum technique can be combined with other gradient tracking or variance reduction methods to potentially improve convergence.
>
> A full exploration of these algorithms is beyond the scope of this rebuttal, but we believe this highlights the rich design space available for this framework idea.
>
>
>
> [R] Nedić, Angelia, and Alex Olshevsky. _"Stochastic gradient-push for strongly convex functions on time-varying directed graphs."_ IEEE Transactions on Automatic Control 61.12 (2016): 3936-3947.

---

### Official Review · Reviewer_AKZy · 2025-07-01

**Clarity:** 4
**Significance:** 3
**Originality:** 4
**Rating:** 5
**Confidence:** 4

**Summary:**

This paper studies the problem of partial participation in federated learning, in the particular context where clients participate with (fixed) arbitrary probabilities.
First, an interpretation of the properties of FedAvg-like algorithm as a decentralized algorithm with time-varying graphs is proposed, allowing to express the updates through a sequence of stochastic matrix products.
Based on this interpretation, a novel method, coined FOCUS, is proposed, which is built on a primal-dual approach, and recovers gradient-tracking-type ideas.
The proposed method is shown to achieve good empirical performance in comparison with existing methods.

**Questions:**

1. What would the convergence rate look like if removing the artifical dependence in $N$ in the rate? Is it possible to fully state communication and sample complexity results, with full dependency on the constants and on the probability of communication?
2. In particular, $q_i$ seem to appear in Theorem 1 only through its minimal value $\min_i q_i$. Is it possible to obtain something better, like on the average of the $q_i$'s, or is it in some sens optimal?
3. Although the matrix interpretation of the algorithm helps in the design of the new method FOCUS, it seems that it results in unusual conditions on the step size ($\eta \le \mu^2 / L$). Is it because the matrix method is limited? Or because this is not the best matrix decomposition possible? Could it be improved or is there another more profound reason for these conditions to show up?

**Ethical Concerns:**

["NO or VERY MINOR ethics concerns only"]

**Final Justification:**

I was already positive about this paper, and remain positive about it after the authors' response.

I think the way they describe the algortihms using product of stochastic matrices is interesting to the NeurIPS community, which to me justifies acceptance of the paper.

I would appreciate it if the authors could comment on the following two points in the revised manuscript:
- I am a bit surprised by the presentation of their convergence results, which show an explicit dependence on the number of clients $N$ in the convergence rate, which is unusual in FL: this is in fact an artefact of the presentation, since they also decrease the step size with this same value, which may be a bit misleading.
- The stochastic setting is only presented in appendix, while I believe it is important enough to be put in the main text, taking advantage of the additional page for the camera-ready.

**Limitations:**

Yes limitations have been properly discussed.

**Paper Formatting Concerns:**

No issues

**Quality:**

3

**Strengths And Weaknesses:**

**Strengths**
1. The interpertation of FedAvg with partial participation as a decentralized algorithm with time-varying graph is interesting, and allows for representation of algorithms as a product of stochastic matrices, which is often a fruitful approach. This is directly applied to the development of a novel FL method that outperforms many of the existing ones, which is very interesting.
2. The propsed method is the first to achieve linear convergence under arbitrary participation, with without relying on tricks like almost-surely bounded time between successive participation of clients.
3. Theoretical convergence rates are provided, together with numerical evaluation that shows the superiority of the proposed methods on some problems.


**Weaknesses**
1. The convergence rates rely on conditions on the step size like $\eta \le \mu^2 / L$, which are way worse than the typical $\eta \le 1/L$. Moreover, they require the step size to scale with the smallest participation rate $q_i$, which is unknwon and may prevent the algorithm from converging.
2. Convergence rates are somewhat misleading, as they seem to claim that the convergence is faster as the number of clients grow, but at the same time the step size is small than $1/N$. Contrarily to what is written in Theorem 1, there is thus no acceleration with the number of clients $N$ in the convergence rate (which is to be expected in FL).
3. Overall, it is not clear how the convergence speed is affected by the probability of communication, although one could expect this to play a major role in the algorithm's properties.
4. The method is only applied to deterministic gradients, although it borrows analysis techniques that are typical from stochastic approximation (using stochastic matrices).


Typos:
- In Equation (3), the left-hand term should not be the square of the norm, but rather the norm.
- In Equation (18), it seems it should be $\Leftarrow$ since it involves communication.

---

> ### Author Rebuttal · Authors · 2025-07-28
>
> We are very grateful for your thorough review and insightful comments. We have carefully considered all your suggestions and believe it can helped us strengthen the manuscript. Our point-by-point responses to your feedback are as follows.
>
> > Weaknesses 1 & Q1. The convergence rates rely on conditions on the step size like $\eta \leq \mu^2/L$, which are way worse than the typical $\eta \leq 1/L$. Moreover, they require the step size to scale with the smallest participation rate $q_i$, which is unknwon and may prevent the algorithm from converging.
>
> We thank the reviewer for this insightful question. We believe it refers to the first term in our step-size condition for the strongly convex case, which is on the order of $\mu/L^2$ instead of $\mu^2/L$.
>
> We first want to clarify this term and its context. Using the condition number $\kappa =L/\mu$, this term can be written as $1/(\kappa L)$. We acknowledge that this bound is not as tight as the optimal rate for standard Gradient Descent, which is $O(2/(\mu+L))$. Our bound matches the optimal one only in the specific quadratic case where $\mu=L$ and is sub-optimal otherwise.
>
> The source of this sub-optimality is not inherent to our matrix-based method, but rather stems from the analytical tools used in the proof. Here is the key reason:
>
> In standard GD, **the tight bound is often derived using the co-coercivity property, which elegantly captures the joint relationship between the smoothness constant $L$ and the strong convexity constant $\mu$. However, due to the increased complexity of the FOCUS algorithm, directly applying the co-coercivity property is not straightforward.** Instead, to handle the complex cross-terms in our analysis (see Equation 81), we use Young's inequality. While effective, this inequality is looser and separates the effects of $\mu$ and $L$, which introduces the slack that leads to the sub-optimal bound.
>
> > Weakness 2 & Question 3. Convergence rates are somewhat misleading, as they seem to claim that the convergence is faster as the number of clients grow, but at the same time the step size is small than $1/N$. Contrarily to what is written in Theorem 1, there is thus no acceleration with the number of clients $N$ in the convergence rate (which is to be expected in FL).
>
> We thank the reviewer for this insightful observation. The reviewer correctly notes that in our deterministic analysis, the coeffient $(1-\eta O(N))$ term in the convergence term is offset by the $O(1/N)$ dependency in the step-size constraint.
>
> **It is actually expected and impossible to improve this rate.** Before we give a detailed reason. Let's give an example to illustrate it. Suppose $f(x) = \frac{1}{N}\sum_i f_i(x_i) =  \frac{1}{N}\sum_{i}\|x\|^2 \equiv \|x\|^2$. Clearly, no first-order FL method can exhibit a convergence rate that accelerates with $N$ for this problem, because adding more identical clients doesn't make the problem fundamentally easier to solve. FOCUS, being an exact algorithm in its primary analysis, correctly reflects this reality.
>
> **This apparent contradiction with the common sense of linear speedup in FL arises because FOCUS is an exact algorithm.** The typical "linear speedup" in federated learning is a feature of stochastic settings. It's a direct result of variance reduction; by averaging stochastic gradients from $N$ clients, the variance of the gradient estimate is reduced, leading to a diminishing error term, often on the order of $O(\eta^2\sigma^2/N)$. However, as an exact algorithm, FOCUS has no such stochastic noise term to average out. Its convergence rate is determined by the deterministic properties of the objective function. As the previous toy example illustrates, simply adding more clients to a deterministic problem does not make it fundamentally easier or faster to solve or improve the rate of descent.
>
>
> **Crucially, our framework is fully consistent with the standard notion of linear speedup when stochastic noise is introduced.**
> As demonstrated in our Theorem 5, the analysis for the SG-FOCUS applied to general non-convex problems with stochastic gradients—reveals the expected linear speedup with respect to the number of agents, $N$. The part of our error bound dependent on variance scales as $O(N)\eta^2\sigma$. By selecting the step-size appropriately, with $\eta \propto 1/N$, this term reduces to the familiar $O(\sigma^2/N)$ rate.
>
>
> > Weakness 3 & Q2. Overall, it is not clear how the convergence speed is affected by the probability of communication, although one could expect this to play a major role in the algorithm's properties.
>
> This is an excellent and important question. We agree with the reviewer's sharp intuition that the bound involving $q_{\min}$ is likely not tight, and a more refined analysis would be a valuable contribution. Given that the primary message of this paper is the algorithm's design and the existing convergence proof is already quite involved, we believe a deep investigation into tightening this bound is best suited for future work.
>
> However, we are happy to share our perspective on a potential path forward. The $q_{\min}$ term is introduced in our consensus analysis (Lemma 4, specifically Eq. 91) to establish a clear recursion for the total (unweighted) consensus error. A promising approach to achieve a tighter bound would be to define a weighted consensus error term, analogous to the technique used in the FedAvg analysis (referenced in Eq. 29). This would likely involve adapting the method from our proof (between Eqs. 54 and 55), where the vector q is handled collectively via Jensen's inequality, thus avoiding the simplification to its minimum element.
>
>
> > Weakness 4. The method is only applied to deterministic gradients, although it borrows analysis techniques that are typical from stochastic approximation (using stochastic matrices).
>
> We thank the reviewer for this point. We believe it only refers to our analysis of FOCUS in the deterministic setting, and we'd like to clarify that **we also provide a full analysis for the stochastic gradient setting. Our proposed SG-FOCUS variant is designed specifically for this case.** Due to page limitations, the detailed algorithm and its convergence analysis were placed in Appendix E and F. We included pointers to this material in the main text, both in our list of contributions and in Section 5.2. When there are more space for the revised version, we will move the key results of SG-FOCUS into the main body of the paper to improve its visibility.
>
> > Typos:
> (1) In Equation (3), the left-hand term should not be the square of the norm, but rather the norm.
> (2) In Equation (18), it seems it should be $\Leftarrow$ since it involves communication.
>
> We are grateful to the reviewer for your meticulous reading and for catching these typos. We will correct them in the revised manuscript.

---

> > ### Comment · Reviewer_AKZy · 2025-08-03
> >
> > I would like to thank the authors for addressing my concerns, and I remain positive about this paper.
> >
> > I know what the "linear speed-up" is. I was just surprising to see rates that depend explicitely in $N$, which is generally not the case: in fact, they do not depend on it, since the step size is scaled down in $1/N$. I think the presentation of Theorem 1 us thus misleading, since it seems to imply that convergence is faster when N increases, which is not the case, and can indeed not be improved.
> >
> > Nonetheless, I still think that the matrix point of view is still interesting enough to justify acceptance of the paper.
> > However, I think that the paper would be much improved if the authors could fix this to avoid the misleading presentation of their Theorem 1, and discuss the stochastic case in the main text.

---

> ### Author Response · Authors · 2025-08-03
>
> We thank the reviewer for your continued positive feedback and constructive suggestions for improving our paper! We will include a discussion to discuss the rate in terms of number $N$ and the key results of SG-FOCUS into the main body when space allowed.
>
> To clarify this unusual the $N$ factor: it's a direct consequence of our framework's use of a column stochastic matrix. This makes the aggregation step for our tracking variable ($y$) a summation, in contrast to the averaging seen in typical federated learning algorithms.

---

### Official Review · Reviewer_yWqo · 2025-07-03

**Clarity:** 3
**Significance:** 4
**Originality:** 3
**Rating:** 5
**Confidence:** 4

**Summary:**

This paper tackles the arbitrary client participation problem in federated learning, where clients may join or leave at will during FL training. To address this, the authors propose a new analysis framework from the time-varying graph perspective. The theoretical analysis shows that exact and linear convergence for federated learning is achievable.

**Questions:**

1. Can the authors elaborate more on the intuition behind exact convergence gradient tracking?
2. Can the authors provide more experiments
 * under more arbitrary client participation settings?
 * on MIFA and FedVARP algorithms?
3. Are there any connections with the push-sum algorithm in the fully decentralized learning literature, for example, in [R3]?

[R3] Olshevsky, A., Paschalidis, I. C., & Spiridonoff, A. (2018, June). Fully asynchronous push-sum with growing intercommunication intervals. In 2018 Annual American Control Conference (ACC) (pp. 591-596). IEEE.

**Ethical Concerns:**

["NO or VERY MINOR ethics concerns only"]

**Final Justification:**

The authors have addressed all of my concerns. Since this is a solid theoretical paper, the extra experiments will be a bonus to their contributions as the current experiments can already illustrate their points. I believe the authors would include the results in their final revision for a better numerical understanding of their method.

**Limitations:**

yes

**Quality:**

3

**Strengths And Weaknesses:**

### Strengths
1. The paper is well written. I enjoy reading the paper.
2. The proposed approach is novel and outperforms the state-of-the-art.
3. I appreciate the beauty of applying insights from stochastic matrix theory to the analysis of the federated learning algorithm.

### Weaknesses
1. FOCUS combines insights from a fully decentralized algorithm and a federated learning algorithm with a server. In particular, it is partially inspired by the gradient tracking algorithm. It is a pity that the authors did not provide much intuition to help the readers understand.
2.  In the experiment section, only one case of arbitrary participation is provided.
3. To deal with client participation, we also have algorithms based on variance reduction techniques, including MIFA [R1] and FedVARP [R2].

**References:**
[R1] Gu, X., Huang, K., Zhang, J., & Huang, L. (2021). Fast federated learning in the presence of arbitrary device unavailability. Advances in Neural Information Processing Systems, 34, 12052-12064.

[R2] Jhunjhunwala, D., Sharma, P., Nagarkatti, A., & Joshi, G. (2022, August). Fedvarp: Tackling the variance due to partial client participation in federated learning. In Uncertainty in Artificial Intelligence (pp. 906-916). PMLR.

---

> ### Author Rebuttal · Authors · 2025-07-28
>
> We sincerely thank the reviewer for the thoughtful comments and meaningful questions. This valuable feedback has allowed us to improve the manuscript. Our detailed responses to each point are provided below.
>
> > Weakness 1 & Q1. The intuition behind exact convergence gradient tracking?
>
> **Our response:** Thank you for raising this important question. We are happy to elaborate here and will add a detailed explanation to the appendix to make the manuscript more self-contained.
>
> The intuition behind push-pull algorithms can be understood through three core concepts: the property of the column stochastic matrix ($C$), the property of the row stochastic matrix ($R$), and the gradient tracking mechanism.
>
> - **Column Stochastic Matrix ($C$): Mass Conservation**. Imagine a vector $y$ where each element $y_{i}$ represent some mass of the $i-$th object. In the operation $y^+ = Cy$ , the total sum of these masses, $\sum_{i} y_{i}$ is preserved. (The average is preserved as well). This is because of the column stochatic definition, $\mathbb{1}^T C=\mathbb{1}^T$, so the sum remains unchanged:
> $$
> \bar{y}^+ := \mathbb{1}^T y^+ = \mathbb{1}^T C y = \mathbb{1}^Ty := \bar{y}
> $$
> In our algorithm, $y_{i}$ is analogous to the local gradient information at client $i$. Using a column stochastic matrix ensures that the sum of these local quantities, which represents the global gradient (server-side), is not arbitrarily altered during aggregation for the next tracking one.
>
> - **Row Stochastic Matrix ($R$): Reaching Consensus**. A row stochastic matrix, $R$, drives the system toward consensus. Under mild conditions on $R$, repeating the process that $y^+ = Ry$, i.e. $y^{\infty} = \lim_{n\to\infty} R^n y$, causes all elements of the vector to converge to a single, common value. This is also commonly referred as the reaching the consensus. This property is crucial for the model parameters, $x$, as we require all clients to eventually agree on the same model.
>
> - **The Tracking Trick: Telescoping Sum**. The tracking mechanism functions like a telescoping sum to ensure sum of the gradient estimator is unbiased to the sum of desired global gradient. Consider a simplified update for the tracking variable $y$
> $$
>  y^+ = C(y + \nabla f(x) - \nabla f(x'))
> $$
> Left multiplying $ \mathbb{1}^T$, we get:
> $$
>  \mathbb{1}^T y^+ = \mathbb{1}^Ty + \mathbb{1}^T\nabla f(x) - \mathbb{1}^T \nabla f(x')
> $$
> If we initialize the tracker such that its sum matches the sum of the gradients $\mathbb{1}^Ty = \mathbb{1}^T \nabla f(x')$, the update ensures that the tracker's sum will perfectly match the sum of the new gradients $\mathbb{1}^T y^+ = \mathbb{1}^T\nabla f(x)$. By induction, the tracker sum correctly follows the true global gradient sum at every step.
>
> By adding this detailed breakdown to the appendix, we hope that the explanation in the main text (lines 192-204) will be crystal clear, allowing readers to understand how the push-pull algorithm elegantly satisfies both the consensus and gradient tracking requirements.
>
> Lastly, we want to highlight that the gradient tracking technique used in our work is just one possible method to achieve exact convergence within our proposed framework. One of the key contribution and the core idea of this paper is to present a general methodology that 1. We use stochastic matrices as a tool to model the decentralized learning process. 2.We reformulate the original unconstrained problem (1) into the equivalent constrained problems (9) and (10). Once the problem is expressed in this constrained form, a variety of optimization algorithms can be applied. **Crucially, from pure optimization perspective, this formulation implies that we need the primal-dual method instead of primal-only method to achieve the exact, linear convergence in FL problem.** Our work with the push-pull algorithm is one such example. Other promising approaches could also be adapted within this framework, including methods like push-sum (from reference [R3], which the reviewer helpfully provided) and algorithms like EXTRA and Exact Diffusion. **Ultimately, we believe that this framework-based methodology, rather than relying on algorithm-specific heuristics, is what opens the door to a broader class of efficient and provably exact convergence FL algorithms.**
>
> > Weakness 2. In the experiment section, only one case of arbitrary participation is provided.
>
> **Our response:** We thank the reviewer for highlighting the experiments, but **we would like to respectfully point out that the appendix already contains experiments on both active arbitrary client participation (Case 2) and passive client participation (Case 3).**
>
> We agree that more extensive experiments are valuable and will add a broader suite of results for different participation scenarios to the revised manuscript. While we cannot include new figures in this rebuttal, we have already conducted these experiments and can summarize our key observations
>
> - Low Participation Probability: When some clients participate very infrequently, the convergence rate is, as expected, slower. The loss curve sometimes exhibits temporary plateaus for several rounds before resuming its downward trend. This confirms the algorithm's robustness, as it still converges correctly even with sparse client contributions.
>
> - High Participation Probability: Conversely, as the client participation probability increases, the convergence curve becomes smoother and more closely approaches the linear convergence rate shown in the full-participation case (Figure 1).
>
>
> > Weakness 3 & Question 2. To deal with client participation, we also have algorithms based on variance reduction techniques, including MIFA [R1] and FedVARP [R2].
>
> **Our response:** Thank you for raising this important point. We were aware of many variance reduction (VR) methods developed for federated learning, including MIFA [R1], FedVARP [R2], SCAFFOLD, Anarchic FL, and FedBuff.
>
> However, our work differs from these methods in design philosophy, problem setting, and theoretical foundation.
> Existing VR-based algorithms rely on heuristic construction of variance-reducing control variates.
> In contrast, **our method (FOCUS) is systematically derived by reformulating the federated learning objective as a constrained optimization problem and applying a principled primal-dual analysis.**
>
> As a result, prior VR methods require strong assumptions on client participation, such as bounded delay in MIFA (Assumption 4 in [R1]) or uniform client sampling in FedVARP (line 3 in algorithm [R2]), which are not realistic in practical federated systems where client participation is dynamic and unknown. In contrast, our method is explicitly designed to operate under arbitrary client participation, a significantly more challenging and general setting not addressed by the aforementioned works.
>
> More importantly, achieving exact convergence under arbitrary client participation remains an open challenge. Prior works [Wang and Ji, 2022; Wang et al., 2020] have shown that such participation introduces an inherent bias, leading models to converge to stationary points of a distorted, participation-weighted objective rather than the true global one. To the best of our knowledge, our work is the first to provide a rigorous framework that enables exact convergence in this challenging setting. This is achieved through a novel constrained optimization problem reformulation with a principled primal-dual analysis, new analytical framework via stochastic matrix analysis, and new algorithmic insights into mitigating client participation bias. it would be nearly impossible to arrive at FOCUS through heuristic design.
>
> In summary, while there is a rich body of work on VR methods in federated learning, **our approach introduces a new formulation, methodology, and theory tailored for the realistic and challenging setting of arbitrary client participation,** which together constitute a significant and novel contribution.
>
> In our paper, we selected SCAFFOLD as a representative VR method for detailed comparison. Following the reviewer’s suggestion, we have conducted additional experiments with MIFA and FedVARP. The results show performance comparable to SCAFFOLD but consistently underperformed compared to our FOCUS algorithm. This is expected, given their shared reliance on heuristically constructed variance control variates. We will include these results in the revision.
>
> > Question 3: Are there any connections with the push-sum algorithm in the fully decentralized learning literature, for example, in [R3]?
>
> **Our response:** Thank you for providing this excellent reference. The answer is yes -- both shared the some common traits.
>
> First, to clarify the terminology, the push-sum technique is designed to achieve an unbiased consensus using only a column stochastic matrix, avoiding the more restrictive requirement of a doubly stochastic matrix. When this is combined with a gradient oracle, it results in algorithms like (Sub)gradient-Push.
>
> The similarities with FOCUS are: both rely on a column stochastic matrix and a gradient tracking mechanism to ensure the gradient estimator remains unbiased.
>
> **The difference lies in the mechanism used to achieve model consensus:**
>
> - Our push-pull method uses a row stochastic matrix ($R$) to explicitly "pull" the model parameters ($x$) towards a common average.
>
> - The push-sum method, in contrast, maintains an additional iterated scalar that propagates alongside the model parameters using the same column stochastic matrix. It then achieves consensus by normalizing the model state with this scalar (i.e., by calculating the ratio $z=x/y$, as shown in Eq. 7 of [R3]).
>
> Ultimately, both push-pull and push-sum are valid paths to exact convergence. This suggests that a novel FL algorithm could be developed by applying the push-sum technique within the optimization framework presented in our paper, highlighting a promising direction for future work.

---

### Decision · Program_Chairs · 2025-09-17

**Decision:**

Accept (poster)

**Comment:**

The paper presents a new algorithm for Federated Learning (FL) under arbitrary client participation and data heterogeneity. Past algorithms with guarantees in this setting suffer from slow convergence due to a decaying learning rate. The authors propose a novel algorithm, termed FOCUS, with improved convergence guarantees. Strengths of the paper include both the new algorithm but also a new proof technique, that approaches the analysis of the algorithm's performance through a decentralized algorithm over time-varying graph, and allows for the representation of FOCUS but possibly other algorithms as products of stochastic matrices. Several reviewers found this connection/point of view interesting and insightful, and worth reporting to the NeurIPS community. Reviewers also found the theoretical contribution to be solid, meaningfully departing from prior art.

Reviewers have asked for several potential improvements that could improve the camera ready version of the paper:

1. Additional experiments including more varied participation settings as well as comparisons to competitors such as MIFA and FedVARP.

2.   One reviewer remained concerned about the artificiality of the dependence of the convergence guarantee on the number of clients, and remains convinced that it is an artifact of the decreasing  (as $1/N$) step size, rather than the linear speadup that the authors allude to in their response. This should be clarified, ideally via contrasting to how bounds are presented in other papers, ensuring a correct "apples-to-apples" comparison.

3. Several reviewers requested that the stochastic setting, being a main technical contribution, to not be relegated in the appendix, but it has a more prominent position in the main text. Perhaps this can be done by taking advantage of the additional page for the camera-ready version of the paper.

4. The same is true about the overall proof-by-stochastic matrices framework: one reviewer wanted to see a discussion on why this framework is important, how it informed the design of SGFOCUS/FOCUS, and whether it has other applications. Could it, for example be applied to other FL algorithms with arbitrary client participation, like FedAU [Wang and Ji, 2023], FedAWE [Xiang et al., 2024], MIFA [Gu et al., 2021]. If not, it would be great if the paper indicated why/what is the key challenge/point of failure.

5. Finally, several of the smaller issues (typos, questions regarding intuition) raised and addressed during the rebuttal/discussion phase should also be addressed in the final version.